# Upper tropospheric ice sensitivity to sulfate geoengineering

Daniele Visioni[1,2], Giovanni Pitari[1], Glauco di Genova[2], Simone Tilmes[3], and Irene Cionni[4]

[1]Department of Physical and Chemical Sciences, Universitá dell'Aquila, 67100 L'Aquila, Italy
[2]CETEMPS, Universitá dell'Aquila, 67100 L'Aquila, Italy
[3]National Center for Atmospheric Research, Boulder, CO 80305, USA
[4]ENEA, Ente per le Nuove Tecnologie, l'Energia e l'Ambiente, 00123 Rome, Italy

**Abstract.** Aside from the direct surface cooling that sulfate geoengineering (SG) would produce, the investigation of the possible side effects of this method is still ongoing, such as, for instance, on upper tropospheric cirrus cloudiness. The goal of the present study is to better understand the SG thermo-dynamical effects on the freezing mechanisms leading to ice particle formation.This is done by comparing the SG model simulations against a Representative Concentration Pathway 4.5 (RCP4.5) reference case. In one case, the aerosol-driven surface cooling is included and coupled to the stratospheric warming resulting from the aerosol absorption of terrestrial and solar near infrared radiation. In a second SG perturbed case, the surface temperatures are kept unchanged with respect to the reference RCP4.5 case. Surface cooling and lower stratospheric warming, together, tend to stabilize the atmosphere, thus decreasing the turbulence and updraft velocities (-10% in our modelling study). The net effect is an induced cirrus thinning, which may then produce a significant indirect negative radiative forcing (RF). This would go in the same direction as the direct effect of solar radiation scattering by the aerosols, thus influencing the amount of sulfur needed to counteract the positive RF due to the greenhouse gases. In our study, given an 8 Tg-$SO_2$/yr equatorial injection into the lower stratosphere, an all-sky net tropopause RF of -1.46 W/m$^2$ is calculated, of which -0.3 W/m$^2$ (20%) is from the indirect effect on cirrus thinning (6% reduction in ice optical depth). When the surface cooling is ignored, the ice optical depth reduction is lowered to 3%, with an all-sky net tropopause RF of -1.4 W/m$^2$, of which -0.14 W/m$^2$ (10%) is from cirrus thinning. Relative to the clear-sky net tropopause RF due to the SG aerosols (-2.1 W/m$^2$), the cumulative effect of the background clouds and cirrus thinning accounts for +0.6 W/m$^2$, due to the partial compensation of large positive shortwave (+1.6 W/m$^2$) and negative longwave adjustments (-1.0 W/m$^2$). When the surface cooling is ignored, the net cloud adjustment becomes +0.8 W/m$^2$, with the shortwave contribution (+1.5 W/m$^2$) almost twice as much as that of the longwave (-0.7 W/m$^2$). This highlights the importance of including all of the dynamical feedbacks of the SG aerosols.

## 1 Introduction

Sulfate geoengineering (SG) is one of the methods that have been proposed in the scientific community (Budyko (1974); Crutzen (2006); Niemeier and Tilmes (2017)) to cool our planet for a limited amount of time, in response to the warming caused by the increasing greenhouse gases of anthropogenic origin. SG proposes the injection of $SO_2$ into the tropical lower stratosphere in order to produce an optically active cloud of $H_2SO_4$-$H_2O$ supercooled liquid aerosols that would reflect part

of the incoming solar radiation back to space. These aerosols, however, would at the same time warm the lower stratosphere by a few degrees. The idea stems from the cooling effect of past explosive volcanic eruptions in the tropical region (the last being Pinatubo in 1991). These major eruptions injected large amounts of $SO_2$ into the lower stratosphere and increased the planetary albedo. The resulting cooling effect has been clearly observed (Robock (2000)), although its magnitude is still being discussed (Canty et al. (2013)).

In the case of past volcanic eruptions, both the direct and indirect effects of episodic large injections of sulfur into the stratosphere have been observed and documented; this is obviously not possible for planned sustained sulfur injections in the SG experiments. Because of this, the scientific community mainly relies on simulations using climate models and comparisons of the results among them, such as, for instance, under the GeoMIP project (Kravitz et al. (2011); Kravitz et al. (2013)). Different injection scenarios have been proposed and adopted in modelling experiments, the most used being the one with a constant sulfur injection rate at the Equator for a certain number of years to understand the climate response to such an atmospheric perturbation. Simulations have also been performed to identify the optimal magnitude and location of the stratospheric sulfur injection and to obtain the highest ratio between the radiative forcing (RF) and the injection magnitude (Niemeier and Schmidt (2017); Tilmes et al. (2017); Kleinschmitt et al. (2017)).

Amongst various side effects of SG, those with non-negligible impacts on the RF were analysed and summarized in Visioni et al. (2017a). These are related to an enhancement of stratospheric ozone destruction (Tilmes et al. (2008); Pitari et al. (2014); Xia et al. (2017)), an increase in the concentration and lifetime of methane (Visioni et al. (2017b)), an increase of stratospheric water vapour due to a tropical tropopause layer (TTL) warming (Pitari et al. (2014)) and, most importantly, to a change in the probability of the formation of cirrus ice particles in the upper troposphere (UT) (Kuebbeler et al. (2012)). Regarding this latter effect, some studies have appeared in the recent literature that propose ways in which SG could affect the UT cirrus ice number density and optical depth. We will discuss them below and try to expand some aspects further in the present work.

In an unperturbed atmosphere, the formation of UT ice particles may take place either by homogeneous or heterogeneous freezing (Karcher and Lohmann (2002); Hendricks et al. (2011)), with the former process normally dominating over the latter, at least in model simulations (Storelvmo and Herger (2014); Gasparini and Lohmann (2016); Gasparini et al. (2017); Barahona et al. (2017)). Cziczo et al. (2013), however, reported that, in some areas, in-situ measurements show that heterogeneous freezing dominates over homogeneous freezing. Homogeneous freezing takes place when the ice saturation ratio is relatively high (typically above ∼1.5), local temperatures are below the threshold for atmospheric ice particle formation (∼238 K) and supercooled solution droplets are present, namely, sulfate aerosols or sulfate-coated aerosols. Supersaturation conditions are maintained by intense vertical motions controlling the adiabatic cooling rate and bringing water vapour from the lower to the upper troposphere. Ice crystals formed in this way both reflect part of the incoming solar radiation (negative RF) and trap part of the outgoing planetary radiation, contributing to the greenhouse effect (positive RF). The sign of the combined effects could not easily be determined in a variety of atmospheric conditions. Normally, it has been shown that the net UT ice contribution to the RF is positive (Chen et al. (2000); Fusina et al. (2007); Gasparini et al. (2017)). This is, however, a rather delicate balance and strongly depends on the humidity, cloud cover and optical properties (Mitchell et al. (2008)), so that a robust atmospheric

perturbation, such as the one that the SG could produce, may significantly affect it.

The perturbation to the UT ice could be twofold. On one hand, Cirisan et al. (2013) studied how the $H_2SO_4$-$H_2O$ droplets resulting from the sulfur injection would interact with cirrus clouds, both microphysically and radiatively. An upper tropospheric increase of the sulfate aerosol number concentration is expected under the SG conditions due to gravitational sedimentation and the large-scale transport of the particles below the tropopause from the lower stratosphere (LS). However, sulfuric acid liquid supercooled droplets cannot act as ice nuclei (IN) for heterogeneous freezing. At the same time, the background number concentration of the UT aerosols acting as nuclei for homogeneous freezing is already much higher with respect to the ice particle number density. For this reason, a negligible increase of the active IN population would be found in the UT (mainly due to a shift in the distribution of sulfate particles towards radii where homogeneous freezing is more favorable); the same would hold true for the positive RF associated with a possible increase of ice particles from this effect, as Cirisan et al. (2013) concluded in their study.

Kuebbeler et al. (2012), on the other hand, analysed the effects produced by dynamical changes due to the modification of the tropospheric thermal gradient produced by stratospheric geoengineering aerosols. In particular, the LS warming, caused by increasing heating rates in the optically thick sulfate cloud, tends to decrease the tropospheric lapse rate. A subsequent decrease in the available turbulent kinetic energy (TKE) would follow and translate in a slowing down of the updraft and of the adiabatic cooling rate, thus reducing the probability for sufficiently high supersaturation values capable of producing ice crystals formation via homogeneous freezing. Their study found a resulting large net RF reduction in magnitude with respect to clear-sky conditions, where only the direct aerosol forcing is considered (-0.93 W/m$^2$ against -1.53 W/m$^2$). They concluded that this forcing reduction results not only from the mere (passive) presence of background clouds that affect the atmospheric radiative transfer but also from the cirrus cloud thinning produced by the SG aerosols. This may obviously have clear implications regarding the potential of the SG to counterbalance global warming.

The aforementioned study, however, lacked an important part of the possible dynamical feedback of the SG, that is, the changes in sea surface temperatures (SSTs) that would result from the decreased incoming solar radiation. The goal of the present study is to study the impact on cirrus ice particles formed via homogeneous freezing of a stratospheric sulfate injection and to understand how both the local stratospheric warming and the surface and tropospheric cooling can affect this process; to do this, we will use the composition-climate coupled model developed at the University of L'Aquila (ULAQ-CCM). We performed an SG simulation with an 8 Tg-$SO_2$/yr injection, using surface temperatures (Ts) calculated in the atmosphere-ocean coupled model CCSM-CAM4 (Community Climate System Model - Community Atmospheric Model version 4), operated with the same sulfur injection (thus resulting in a general surface cooling, with respect to atmospheric unperturbed conditions). This perturbed experiment (named G4, according to the convention of Kravitz et al. (2011), regardless of the time constant magnitude of the injection) is compared against a baseline simulation without SG and using a background anthropogenic emission scenario corresponding to the Representative Concentration Pathway 4.5 (RCP4.5) (Taylor et al. (2012)) (named Base case in our study). To properly compare our results with those of Kuebbeler et al. (2012), a third simulation was performed with the

same geoengineering sulfur injection of G4 but with the surface temperatures fixed at the Base case values (named G4K).

The effects of the SG Ts changes on the lower stratospheric dynamics were already discussed in Visioni et al. (2017b); this time, we focus on their impact in the upper troposphere. Unlike other side effects of sulfur injection into the stratosphere, a comparison between the effects of a volcanic eruption and the SG on cirrus ice is difficult to draw. This is mainly because in a volcanic eruption episode (contrary to SG), a large amount of solid ash particles is injected into the lower stratosphere together with $SO_2$. Part of these particles, after settling down below the tropopause, may contribute to increasing the number density of IN available for heterogeneous freezing in the UT. This could help to explain some observed increases in UT ice particles after the Pinatubo eruption (Sassen et al. (1995)). More recently, Friberg et al. (2005) showed that cirrus cloud reflectance and optical depth are reduced in the Northern Hemisphere in periods with more pronounced volcanic activity. However, other studies such as Meyer et al. (2015) dispute this effect, and no conclusive answer can be given.

Understanding the RF contribution of the UT ice perturbation in a SG scenario is particularly crucial if the scientific community wants to design experiments whose goal is to meet a given climate target, as proposed in Kravitz et al. (2017) and MacMartin et al. (2017).

This paper is structured in three subsequent sections plus the conclusions: in Section 2, we describe the CCSM-CAM4 and ULAQ-CCM models and the setup of the numerical experiments. Furthermore, in Section 2 we try to evaluate the ULAQ-CCM skill in simulating the formation of the cirrus ice clouds, using re-analysis and satellite data. In Section 3, we discuss the model-calculated changes in the thermo-dynamical properties of the atmosphere and in cirrus cloudiness (size distribution, extinction, optical depth, number concentration) produced by the SG, and finally, we show how these perturbations translate into tropopause radiative forcing terms.

## 2 Model descriptions and setup of numerical experiments

### 2.1 CCSM-CAM4

The Community Climate System Model - Community Atmospheric Model version 4 (CCSM-CAM4) is an atmosphere-ocean coupled model that was used in this experiment to calculate the evolution of Ts for both the Base case (RCP4.5 scenario) and a geoengineering case with the same sulfur injection as the ULAQ-CCM model, described in Tilmes et al. (2015). For these simulations, the model was run without interactive chemistry. The resolution of the model is $1.9° \times 2.5°$ with 26 vertical levels and the top of the model is at 3 hPa. The model has been fully described in Neale et al. (2013) and Tilmes et al. (2016) and has been shown to compare well against observations in the stratosphere in Lamarque et al. (2012). Ice clouds are diagnosed from a purely relative humidity-based formulation (Neale et al. (2013)). The results of an 8 Tg-$SO_2$/yr injection on surface temperatures and the effects of the inclusion of the perturbed Ts in the ULAQ-CCM model have been already discussed in

Visioni et al. (2017b).

## 2.2 ULAQ-CCM

The University of L'Aquila composition-climate coupled model was described in its first version in Pitari et al. (2002); sub-sequent model versions were documented in modelling intercomparison campaigns (Eyring et al. (2006); Morgenstern et al. (2010); Morgenstern et al. (2017)). Model updates in the horizontal and vertical resolution, photolysis cross sections, the treatment of Schumann-Runge bands and radiative transfer code were described and tested in Pitari et al. (2014) and Chipperfield et al. (2014). The shortwave radiative module has been documented and tested for tropospheric aerosols in Randles et al. (2013) and for volcanic stratospheric aerosols in Pitari et al. (2016a). It makes use of a two-stream delta-Eddington approximation and is on-line in the model for photolysis, solar heating rates and radiative flux calculations. A companion broadband, k-distribution longwave radiative module is used for the heating rate and radiative flux calculations in the planetary infrared spectrum (Chou (2001)).

A critical atmospheric region in the SG studies is the upper troposphere and lower stratosphere (UTLS). An extensive model evaluation based on specific physical and chemical aspects was made in Gettelman et al. (2010) and Hegglin et al. (2010)). Subsequent model improvements in this region were discussed in Pitari et al. (2016b). The treatment of Ts, and their importance for the lower stratospheric dynamics and species transport under a geoengineering scenario, has been discussed in Visioni et al. (2017b). Another very important aspect to be taken into account for large-scale species transport in the lower stratosphere is the role of the quasi-biennial oscillation (QBO) in SG studies. It has been discussed from different points of view in some recent studies (Aquila et al. (2014); Niemeier and Schmidt (2017); Visioni et al. (2018)). A nudging procedure for the QBO is adopted in the ULAQ-CCM, based on an observed historical data series of equatorial mean zonal winds (Morgenstern et al. (2017)).

For the sake of completeness, we discuss in the following two sub-headings some of the model features, in particular, those relevant for stratospheric sulfate aerosols and upper tropospheric cirrus ice particle formation.

**Table 1.** Summary of ULAQ-CCM features and numerical experiments for the present study.

| Years of simulation | 1960-2015 | 2020-2069 |
|---|---|---|
| Type of simulation | Reference | Base (RCP4.5) + G4 + G4K |
| Ensemble size | 2 | 1 +2 + 2 |
| Horizontal and vertical resolution | \multicolumn | |
| Chemistry | On-line (strat & trop) | |
| Dynamics | Calculated[1] | Calculated[2] |
| QBO | Nudged (from eqt. wind obs.) | Nudged (iteration of observed cycles of eqt. winds) |
| Altitude of equatorual injection of SO$_2$ in G4 (8 Tg-SO$_2$/yr) | - | 18-25 km (Gaussian Distribution) |

The "Horizontal and vertical resolution" row spans: 5° × 6°, L126 log-pressure / top: 0.04 hPa

[1] Sea surface temperatures from observations; on-line explicitly calculated land temperatures.

[2] Surface temperatures from CCSM-CAM4 (land, ocean, sea ice coverage), separately for Base and G4 (Visioni et al. (2017b)); Base values also used for G4K. Indirect effects of SG aerosols on surface temperatures are calculated on-line in the ULAQ-CCM radiative module (due to UT ice, GHGs and SO$_4$ imbalance relative to CCSM-CAM4); see text in section 2.3.

### 2.2.1 Stratospheric sulfate aerosols

In SG experiments G4 and G4K, SO$_2$ is injected at the Equator (0° longitude) throughout the altitude range 18-25 km with a Gaussian distribution centred at 21.5 km. The OH oxidation of SO$_2$ starts the production of supercooled H$_2$O-H$_2$SO$_4$ particles, whose size distribution is calculated in an aerosol microphysics module with a sectional approach, starting from gas-particle interaction processes (nucleation, H$_2$SO$_4$ condensation and H$_2$O growth) and then including aerosol particle coagulation. Removal processes are included via gravitational settling across the tropopause and evaporation in the upper stratosphere (Visioni et al. (2018)).

In the troposphere, the ULAQ-CCM includes sulfate production from the dimethyl sulfide (DMS) and SO$_2$ emissions, with gas phase and aqueous/ice SO$_2$ oxidation (by OH and H$_2$O$_2$, O$_3$, respectively) to produce SO$_4$ (Feichter et al. (1996); Clegg and Abbatt (2001)). The tropospheric and stratospheric SOx budget in the ULAQ-CCM (for unperturbed background conditions) was recently discussed in Pitari et al. (2016c), with a focus on the role of non-explosive volcanic sulfur emissions, and in Visioni et al. (2018), in connection with the SG.

Aerosol extinction, optical thickness, single scattering albedo and surface area density are calculated on-line at all model grid-points every hour. This allows the interactive calculation of up/down diffuse radiation and absorption of solar near-infrared and planetary radiation by SG aerosols, with explicit full coupling of the aerosol, chemistry and radiation modules in the ULAQ-CCM. This justifies the 'composition-climate' name for this coupled model, which is more general than the usual 'chemistry-climate' model name.

The ULAQ-CCM ability to produce the correct confinement of sulfate aerosols in the tropical stratosphere has already been documented in the literature, with a comparison against SAGE II data following the Pinatubo eruption or looking at the SG

conditions (see Pitari et al. (2014); Pitari et al. (2016a); Visioni et al. (2017b)).

### 2.2.2 Upper tropospheric ice

The formation of UT ice particles may take place via heterogeneous and homogeneous freezing mechanisms. In the latter
case, the ULAQ-CCM adopts the approach initially described in Karcher and Lohmann (2002), which assumes ice crystals
formed only via homogeneous freezing of solution droplets as a function of local UT temperatures and updraft velocities, also
including the effects of a variable aerosol size distribution. These updraft velocities are obtained as the sum of a dominant
term related to the TKE and a much smaller contribution from the large-scale tropospheric circulation (Lohmann and Karcher
(2002)). Typical vertical velocity net values are on the order of 10-20 cm/s (see Section 3.1) and allow the formation of thin
cirrus.

For the ice supersaturation ratio, we adopt a simplified probabilistic approach, starting from the knowledge of climatolog-
ical frequencies of the UT relative humidity ($RH_{ICE}$), from which a mean value and a standard deviation can be calculated,
assuming a normal distribution. Local ice super-saturation conditions ($RH_{ICE}>100\%$) are a result of turbulent ascent and can
be found in the UT, in the vertical layer below the tropopause (where turbulent updraft conditions may be found) and above
an altitude where T < 238 K (i.e., the assumed threshold for the spontaneous freezing of solution droplets). Here, the condi-
tions for ice formation are met and we may calculate the probability that $RH_{ICE}>1.5$ ($P_{HOM}$). This represent the assumed
threshold for homogeneous freezing to be activated (in our model this threshold does not depend on local temperature or wa-
ter activity conditions), which is considerably higher with respect to the threshold for heterogeneous freezing to take place
($RH_{ICE}>\sim1.3$) (Hendricks et al. (2011)). This represents the probability that an ice particle could be formed via heteroge-
neous freezing ($P_{HET}$) on a pre-existing population of ice condensation nuclei ($N_{IN}$), typically mineral dust or BC particles
transported from the surface.

The size distribution and number density $n_{HET}$ of ice particles formed via heterogeneous freezing is calculated starting from
the formulation of Hendricks et al. (2011) using the ULAQ microphysical scheme adopted for polar stratospheric ice particle
formation (Pitari et al. (2002)). $N_{IN}$ is the sum of grid-point model-predicted concentrations of mineral dust and black carbon
aerosols ($N_{DU}$ and $N_{BC}$, respectively) and is used as the population of available condensation nuclei, with $P_{HET}$ being the
probability that $RH_{ICE} > 1.3$ at any model grid point. The problem in this case is on the actual availability of solid ice nuclei.
A low fraction of activated IN is suggested in the literature ($f_{DU}=1\%$ for mineral dust and $f_{BC}=0.25\%$ for BC) because the
large majority of IN will rapidly be coated by sulfate (Hendricks et al. (2011)). The number density $n_{HET}$ is then obtained as:

$$n_{HET} = (f_{BC}N_{BC} + f_{DU}N_{DU})P_{HET} \tag{1}$$

The specification of the active ice fraction for both mineral dust and BC represents the major source of uncertainty for UT
ice particle formation via heterogeneous freezing. With the above assumptions, in the ULAQ-CCM homogeneous freezing

normally dominates ice particle formation, with respect to the heterogeneous freezing mechanism. However, this may not be considered a general conclusion, assumed to be valid in all thermodynamics conditions and any local atmospheric composition, as it has been shown for instance in Cziczo et al. (2013), where a predominance of heterogeneous freezing over homogeneous may be found. In general, which freezing mechanism dominates in the atmosphere is still very uncertain.

The calculated mass mixing ratio of ice formed in the ULAQ-CCM through both freezing mechanisms is shown in Fig. 1ac for two pressure layers, 150-200 hPa and 350-400 hPa, where the ice formation is greater in the tropics and mid-high latitudes, respectively. These calculations are compared against the MERRA-2 (Bosilovich et al. (2017); Gelaro et al. (2017)) and ERA5 reanalyses (Stephens et al. (2000)), all averaged over the same decade (2003-2012). For the upper layer (150-200

10   hPa), we also show in Fig. S1 the MLS satellite retrieval (Wu et al. (2008)), which compares very closely to the ERA5 reanalysis. Tropical ice formation shows a strong land-ocean asymmetry due to significantly higher $P_{HOM}$ and $P_{HET}$ values over land. For both pressure layers, the magnitude and spatial distribution of the ice mass mixing ratio are comparable between the ULAQ-CCM and MERRA. Regarding the datasets used to compare against our model results, note that there is a large spread amongst retrievals (such as MODIS or CALIPSO) and amongst reanalyses (Zhang et al. (2010); Duncan and Eriksson (2018)).

In particular, MERRA-2 appears to be in the lower end of the spectrum in regard to some quantities, such as ice water path. Considering that the dataset only considers non precipitating ice (Duncan and Eriksson (2018)), this quantity might be however closer to the one simulated in our model, and thus allow for a more correct comparison.

While the probability of homogeneous ice formation is defined as above, the number density and size of the ice particles

formed this way is determined by the local temperatures and vertical velocities, in addition to the competing ice formation mechanism via heterogeneous freezing. The lower the temperature, the faster the nucleation rate; thus, more ice crystals can be formed. On the other hand, higher vertical velocities increase the saturation ratio, leading to more ice crystals formed before water deposition on ice crystals reduces supersaturation below the threshold. The spatial distribution of the cirrus ice optical depth (OD) in the model is calculated as:

$$\tau_{ice} = \Delta z \sum_i \sum_j Q_{ext} \pi r_{ij}^2 n_{ij}(r) \qquad (2)$$

where the extinction efficiency coefficient $Q_{ext} \sim 2$ at all visible wavelengths for ice particle sizes on the order of 5-50 $\mu$m; i is an index for the vertical layers, and the sum is over all the vertical layers in the UT; j is an index for the particle size bins, and the sum is over the whole size distribution; $r_{ij}$ is the particle radius at the i-th layer and j-th bin; and $n_{ij}$ is the corresponding ice number density.

Equation 2 can easily be applied to the model, and the results are shown in Fig. 2a. An evaluation can be made again using the ice mixing ratio from MERRA-2 and ERA5 (shown in Fig. 1bc for two specific pressure layers), together with the ULAQ-

## Ice mass mixing ratio (mg/kg)

**Figure 1.** Lat/lon maps of the ice mass mixing ratio (mg/kg-air) for pressure layers representative of tropical (panels a,c,e) and extratropical (panels b,d,f) upper troposphere. Panels (a,b) are for the ULAQ-CCM; panels (c,d) are for MERRA-2 data (Bosilovich et al. (2017)); panels (e,f) are for ERA5 data (Stephens et al. (2000)). Time average is on years 2003-2012.

CCM values of the ice particle effective radius. With these two quantities we have indirectly derived $\tau_{ice}$ at every horizontal grid point in Eq. (2), using the hydrostatic equation:

$$\tau_{ice} = Q_{ext} \frac{3}{2} \frac{\Delta p}{g} \frac{1}{\rho_{ice}} \sum_i \frac{\chi_i}{r_i} \tag{3}$$

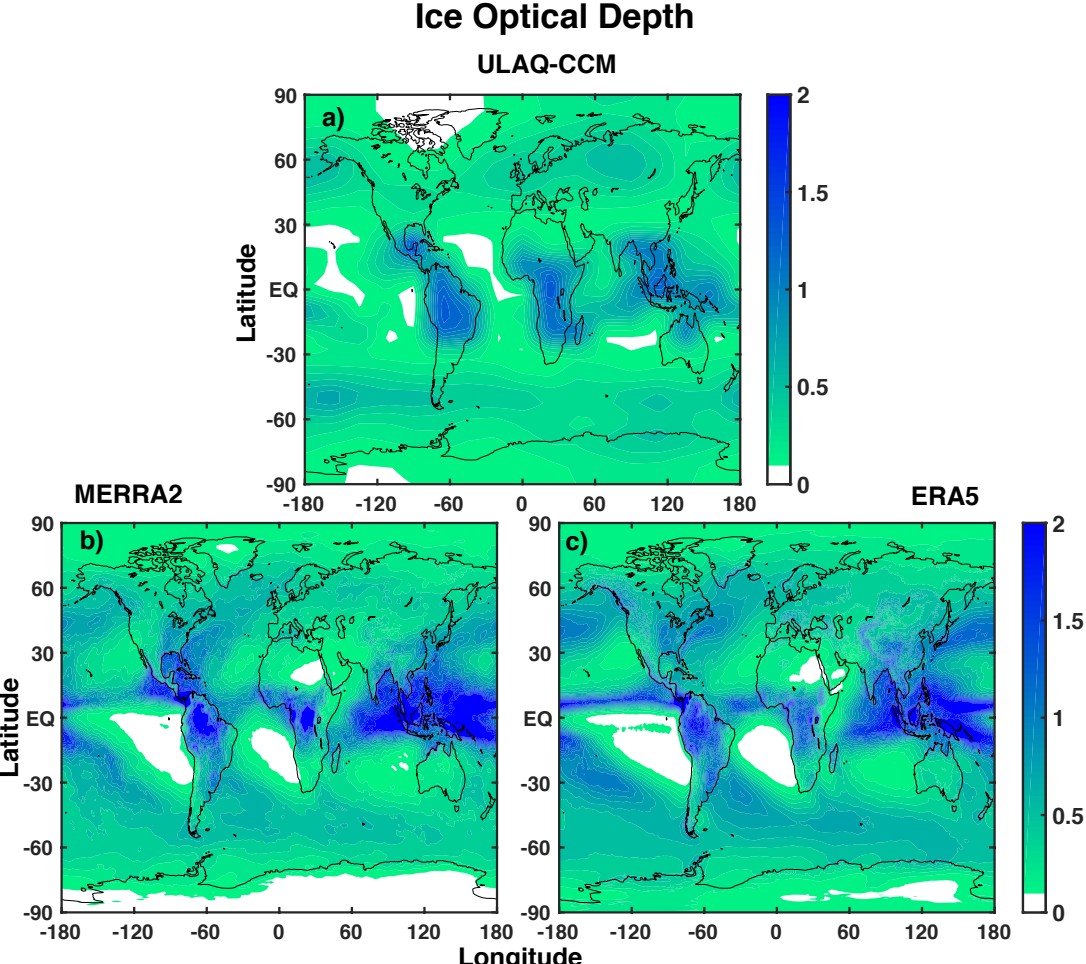

**Figure 2.** Ice optical depth at $\lambda$=0.55 $\mu$m, from ULAQ-CCM calculations (a) and from the MERRA-2 (b) and ERA5 (c) ice mass mixing ratio (100 <p< 450 hPa), with ULAQ-CCM particle effective radius. Time is averaged over the years 2003-2012.

where the sum is, again, over all the vertical layers (constant $\Delta$p=50 hPa), g is the acceleration of gravity, $\rho_{ice}$ is the ice bulk density, $r_i$ is the ULAQ-CCM effective radius at the $i^{th}$ layer; and $\chi_i$ is the MERRA-2 and ERA5 ice mass mixing ratio at the $i^{th}$ layer. Doing so, we obtain the optical depth values in Fig. 2bc. The ODs are comparable in terms of spatial distribution, with the highest values in the tropics over land. The absolute values in the ULAQ-CCM, however, are significantly smaller over the tropics. The reason is that updraft velocities result in a relatively narrow interval (w<30 cm/s) when calculated only as a function of TKE (as in the ULAQ-CCM), while thick cirrus formation takes place from strong (and less frequent) convective events (w<100 cm/s). This detrained ice originating in deep convection is not included in our model formulation.

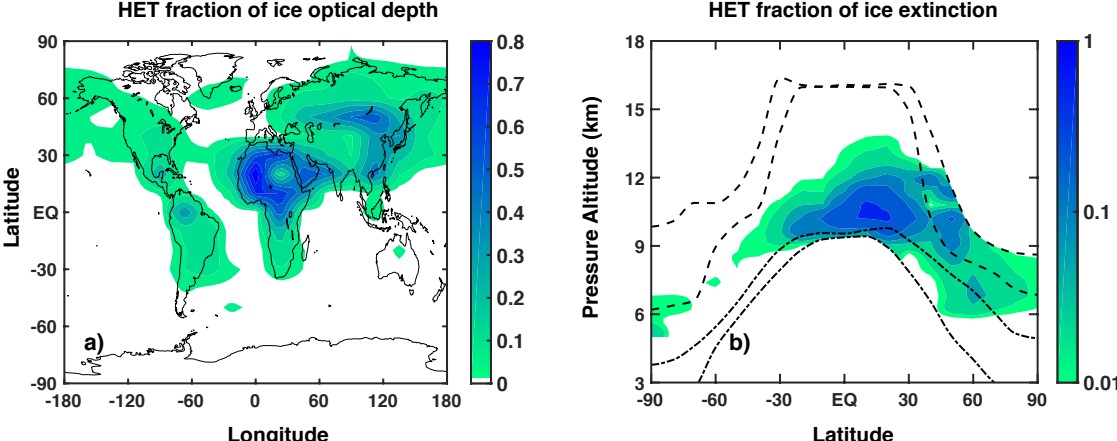

**Figure 3.** Fraction of total ice formed through heterogeneous freezing in ULAQ-CCM averaged over the years 2003-2012, as a function of latitude and longitude for the total optical depth (a) and as a function of altitude and latitude for the zonally averaged extinction (b). In panel (b), the colour scale is logarithmic, starting at 0.01 (i.e., 1% of total ice extinction) up to 1 (100%). The dashed lines show the mean tropopause height with seasonal variability (where seasonal variability is defined as $\pm 1\ \sigma$ of the average height). The dash-dotted lines show the mean height (with seasonal variability) at which T=238 K (freezing is allowed for colder temperatures).

In Fig. 3, we show the model-predicted fraction of ice formed through heterogeneous freezing in terms of optical depth (Fig.3a) and zonally averaged extinction (Fig.3b). In both panels, we see that a large part of the ice particles formed through heterogeneous freezing is located in the tropical band at lower altitudes, where a higher concentration of mineral dust and BC ice nuclei can be transported from the surface. In those regions, the fraction of ice formed this way can be as much as 80% of
the total.

In Fig. 4ab we show the model calculated vertical profiles of ice particle number density averaged over the tropics (Fig. 4a) and the extratropics (Fig. 4b), with superimposed the time variability produced by changing conditions of vertical velocity, temperature and $P_{HOM}$, $P_{HET}$. The ice number density maxima are located at rather different altitudes in the two latitude bands,
close to 13 km in the tropics and 8 km elsewhere. This is clearly expected from the latitudinal variability of the tropopause height.

With a procedure similar to the one described above for the ice OD, we may derive a first order approximation of the ice number density from the MERRA-2 and ERA5 ice mass mixing ratio and ULAQ-CCM radii. Similar to Eq. 3, for the ice number density $n_i$ at each vertical layer we obtain the following expression:

$$n_i = \frac{3}{4\pi} \frac{1}{\rho_{ice}} \frac{1}{r^3} \chi_i \qquad (4)$$

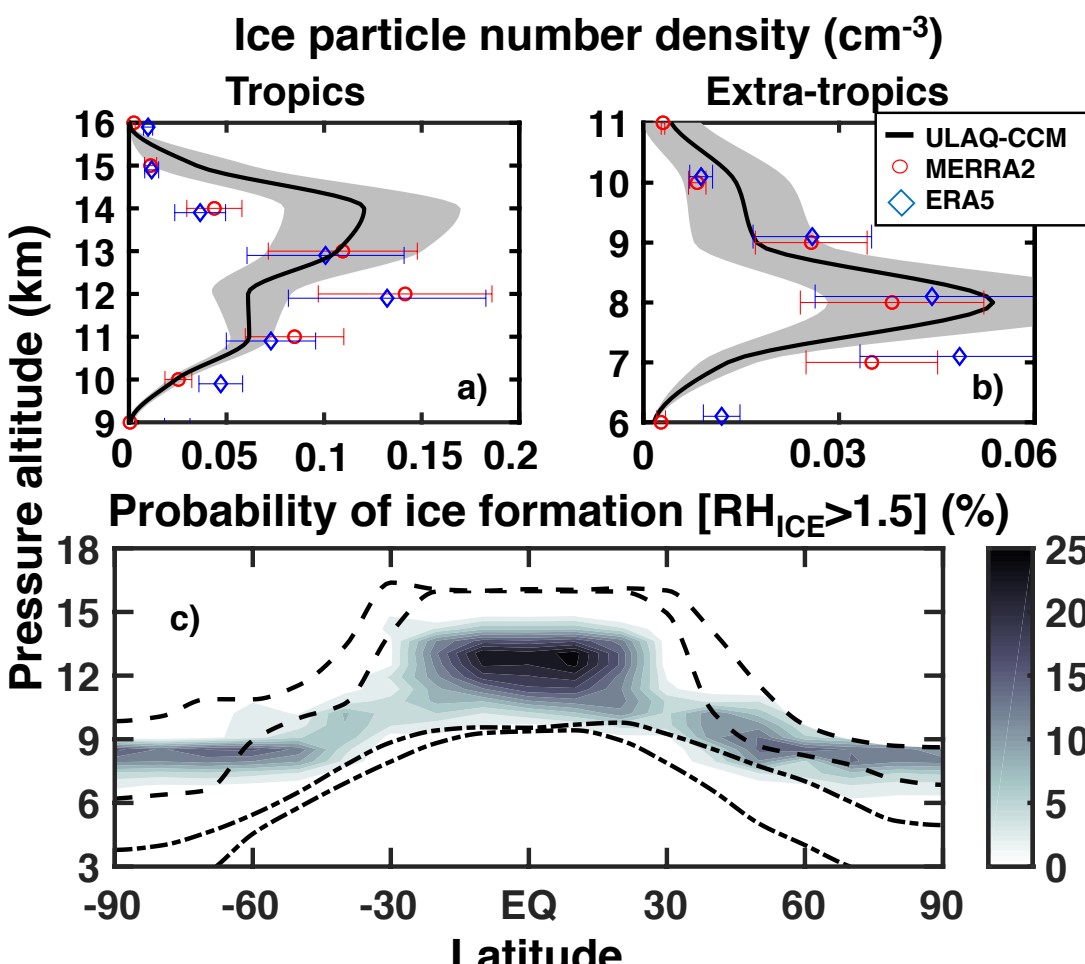

**Figure 4.** Average upper tropospheric profiles of ice particle number density (cm$^{-3}$), for the tropics (25S-25N) and extratropics (35S-90S, 35N-90N), in panels (a) and (b), respectively. Time is averaged over the years 2003-2012. Shaded areas represent $\pm 1\sigma$ for the ensemble over the 10-year period. The red circles show indirectly derived values from the MERRA-2 and ERA5 ice mass mixing ratio and ULAQ-CCM effective radius (see text). Panel (c) shows the zonally averaged probability of ice formation via homogeneous freezing (percent), as a function of altitude and latitude. The dashed lines show the mean tropopause height (with seasonal variability). The dash-dotted lines show the mean height at which T=238 K (with seasonal variability).

The results from Eq. 4 (circles in Fig. 4ab) show that while the model and the indirectly derived values from the reanalyses agree in terms of the general vertical distribution and localization of the maxima in the extratropics, the ULAQ-CCM tends, however, to have smaller number densities in the tropics in the 10-13 km layer. Again, this should not surprise in light of the fact that we are focusing on a specific type of cirrus cloud particles.

5   Figure 4c shows the model-calculated values of $P_{HOM}$, as a 2D zonally averaged distribution. Using these $P_{HOM}$ values, it

is possible to scale a $n_i$ value measured in the mid-latitude airborne campaign of Strom et al. (1997) during a young cirrus formation, to derive an average climatological value to be considered consistent with our modelling approach. They measured a mid-latitude ice concentration value n=0.3 cm$^{-3}$ in a young cirrus cloud at T=220 K and p=320 hPa. If we scale this result with our corresponding $P_{HOM}$=12$\pm$3%, a 'climatological-mean' value n=0.025$\pm$0.005 cm$^{-3}$ is obtained, close to our model

5   prediction value of 0.031$\pm$ 0.008 cm$^{-3}$ (Fig. 4b).

**Table 2.** Summary of globally and time-averaged sulfate aerosol and cirrus ice particle related quantities, as calculated in the ULAQ-CCM and compared with available satellite and reanalyses data. Sulfate aerosols: sectional approach (Pitari et al. (2002); Pitari et al. (2014)). Cirrus ice particles: parameterization for homogenous (HOM) and heterogeneous (HET) freezing are summarized in the text and based on the formulation of Karcher and Lohmann (2002) (HOM), but including the effects of the aerosol size distribution, and Hendricks et al. (2011) (HET); a probabilistic approach is adopted for the ice supersaturation ratio. Standard deviations are calculated over the time series of globally averaged monthly mean values. On the global average, our model predicts a 90% fraction of the ice optical depth formed via homogeneous freezing.

| | |
|---|---|
| Stratospheric sulfate optical depth [post-Pinatubo conditions] [reference (September 1991 - August 1992)] | $0.11 \pm 0.02$ (ULAQ-CCM) $0.13 \pm 0.02$ (SAGE II) $0.13 \pm 0.02$ (AVHRR) |
| Sulfate $r_{eff}$ ($\mu$m) (30-100 hPa, 25S-25N) [post-Pinatubo conditions] [reference (September 1991 - August 1992)] | $0.54 \pm 0.06$ (ULAQ-CCM) $0.58 \pm 0.06$ (SAGE II) |
| Sulfate $r_{eff}$ ($\mu$m) (30-100 hPa, 25S-25N) [volcanic unperturbed conditions] [reference (1999 - 2000)] | $0.19 \pm 0.02$ (ULAQ-CCM) $0.22 \pm 0.02$ (SAGE II) |
| Ice mass mixing ratio (mg/kg) (150-200 hPa) [reference (2003 - 2012)] | $3.3 \pm 0.2$ (ULAQ-CCM) (HOM) $0.1 \pm 0.1$ (ULAQ-CCM) (HET) $3.5 \pm 0.4$ (MERRA-2) $3.2 \pm 0.4$ (ERA5) |
| Ice mass mixing ratio (mg/kg) (200-300 hPa) [reference (2003 - 2012)] | $3.8 \pm 0.5$ (ULAQ-CCM) (HOM) $0.6 \pm 0.2$ (ULAQ-CCM) (HET) $5.5 \pm 0.8$ (MERRA-2) $5.7 \pm 0.9$ (ERA5) |
| Ice mass mixing ratio (mg/kg) (350-400 hPa) [reference (2003 - 2012)] | $2.4 \pm 0.4$ (ULAQ-CCM) (HOM) $0.1 \pm 0.1$ (ULAQ-CCM) (HET) $2.6 \pm 0.5$ (MERRA-2) $2.7 \pm 0.7$ (ERA5) |
| Tropospheric ice $r_{eff}$ ($\mu$m) [reference (2003 - 2012)] | $31.3 \pm 3.1$ (ULAQ-CCM) (HOM) $34.6 \pm 3.8$ (ULAQ-CCM) (HET) $33.4 \pm 2.1$ (MODIS) |
| Tropospheric ice optical depth [reference (2003 - 2012)] | $0.37 \pm 0.03$ (ULAQ-CCM) (HOM) $0.04 \pm 0.01$ (ULAQ-CCM) (HET) $0.62 \pm 0.04$ (MERRA-2) $0.65 \pm 0.06$ (ERA5) |

Relevant aerosol and ice quantities calculated in the ULAQ-CCM are summarized in Table 2 in comparison with available satellite observations. The first two rows in Table 2 compare the ULAQ-CCM results for stratospheric sulfate optical depth (OD) and the tropical effective radius ($r_{eff}$) against SAGE-II and AVHRR satellite observations (Thomason et al. (1997); Long and Stowe (1994)), under post-Pinatubo conditions (Pitari et al. (2016a)). This is done to highlight the realistic repre-
sentation of the gas-particle conversion and aerosol microphysics processes in the model, along with the aerosol large-scale transport in the lower stratosphere in case of a major tropical volcanic eruption, which may be used as a proxy for SG with an equatorial SO$_2$ injection. A comparison of the aerosol effective radii under volcanic and background conditions (see rows 2 and 3 in Table 2) clearly shows the effects of the sulfuric acid condensation on the size extension of the aerosol accumulation mode and how this is represented in the model.

The bottom 5 rows in Table 2 compare the global budget calculations for tropospheric ice particles with values obtained from the MERRA-2 and ERA5 reanalyses (ice mass mixing ratio) and ULAQ-CCM effective radius (compared in row 7 with the ice effective radius as retrieved by MODIS). The simultaneous use of these two products (reanalysis values for ice mass mixing ratio and model calculated radius) allows an indirect calculation of the ice optical depth (row 8 of Table 2), as previously discussed. The ULAQ-CCM OD underestimation is mostly related to the ice mass mixing ratio lower values in the largest portion
of the upper troposphere (see row 5 of Table 2 ) and may be, in part, explained with the inclusion of a relatively narrow interval for updraft velocities (w<30 cm/s).
The values are given separately for the ice formed through homogeneous and heterogeneous freezing.

### 2.3   Setup of the numerical experiments and role of perturbed SSTs

The use of a composition-climate coupled model, such as the ULAQ-CCM model, offers multiple advantages in this type of study: (a) the on-line inclusion of interaction between aerosol and ice particles microphysics with chemistry, radiation, climate, dynamics and transport; (b) the stratosphere-troposphere explicit interactions for the large-scale transport of gas and aerosol species (the model adopted high vertical resolution is important across the tropopause region); (c) the sufficiently detailed chemistry both in the stratosphere and troposphere, with a robust design for heterogeneous chemical reactions on sulfuric acid
aerosols, polar stratospheric cloud particles, and upper tropospheric ice and liquid water cloud particles. This allows us to account for the atmospheric circulation changes produced by sulfate geoengineering. The ULAQ-CCM model has many times proven to be capable of producing sound physical and chemical responses to both sulfate geoengineering (Pitari et al. (2014); Visioni et al. (2017b)) and for large explosive volcanic eruptions (Pitari (1993); Pitari et al. (2016b); Pitari et al. (2016a)).

In addition to a reference historical model experiment (1960-2015), we performed three sets of SG simulations: a baseline (Base) unperturbed case and two geoengineering experiments (G4 and G4K), both run with an injection of 8 Tg-SO$_2$/yr into the equatorial stratosphere between 18 and 25 km of altitude, as described in Kravitz et al. (2011) for the GeoMIP G4 experiment with a sustained fixed injection of sulfur dioxide (5 Tg-SO$_2$/yr in that case: while we use 8 Tg-SO$_2$/yr, all other prescriptions such as height and latitude of the injections are the same as in the above mentioned paper). These numerical experiments were

all run between years 2020 and 2069, with analyses focusing on the 2030 to 2069 period; all take place under the same RCP4.5 reference scenario for well-mixed greenhouse gases. The ULAQ-CCM is not an atmosphere-ocean coupled model and uses externally provided surface temperatures as prescribed boundary condition for the dynamical module. These surface temperatures are taken from the CCSM-CAM4 model, which was run under the same RCP4.5 and G4 conditions (8 Tg-SO$_2$/yr fixed

injection into the equatorial lower stratosphere). In this way our main experiment G4 may account for the Ts response to SG (Fig. 5). We acknowledge that this procedure may only be valid as a first order approximation, considering that CCSM-CAM4 has not been run with a coupled chemistry and a much simpler cirrus parametrization that produces negligible changes in the geoengineering experiment (Neale et al. (2013)). However, we believe it to be still a consistent one, considering that the main effect produced by the sulfate injection is the direct aerosol effect (Visioni et al. (2017a)) and that the prescribed stratospheric

aerosol field in the SG simulation in CCSM-CAM4 (Tilmes et al. (2015)) is comparable to the one produced by the sulfate injection in ULAQ-CCM. With this in mind, in the next paragraphs, we first discuss the Ts perturbation and its significance for this study and then the approach adopted for minimizing the inconsistency introduced in ULAQ-CCM with the use of Ts from a different model.

A strong inter-hemispheric asymmetry in the Ts changes produced by SG is evident in Fig. 5 (see also the annually and zonally averaged values in Fig. 6a), with a negative anomaly in the Arctic region that is approximately 1 K larger than that of the high southern latitudes. The SG cooling impact on the Arctic sea ice is such that larger negative surface temperature anomalies are favoured in the Northern Hemisphere high latitudes for several months during the year, from the fall to spring months (see Fig. 5a, Fig. 5b, Fig. 5d), thus increasing atmospheric stabilization with respect to the Southern Hemisphere. Note,

however, that the dynamical effects of this enhanced atmospheric stability in the SG conditions (decreasing wave activity and turbulence) may be partially counterbalanced by the increased longitudinal variability of the induced cooling, mostly connected with positive surface temperature anomalies in the subpolar North Atlantic. These positive temperature anomalies in the North Atlantic sub-Arctic are a direct consequence of the increasing amount of polar sea ice in the SG conditions, with the southward transport of colder and saltier ocean waters in the sub-Arctic, with respect to the RCP4.5 Base conditions (Tilmes et al. (2009)).

In this way, the North Atlantic subpolar downwelling of these cold surface waters to the deep ocean is favoured with respect to the Base conditions, thus producing positive anomalies in sea surface temperatures.

Although not statistically significant, the SG-induced warming on the Antarctic continent during wintertime (Fig. 5c) is a direct consequence of the geoengineering aerosol positive radiative forcing in the planetary longwave, which represents the net

forcing at these high latitudes in the absence of sunlight. This radiative feature will be further discussed in Section 3. All these high-latitude positive temperature anomalies directly reflect in the large variability of the zonally averaged surface temperature changes presented in Fig. 6a.

To correct for the potentially significant model inconsistency introduced by the use of surface temperatures taken from a

different model, the following procedure has been adopted. The ULAQ-CCM radiative-climate module has been modified for

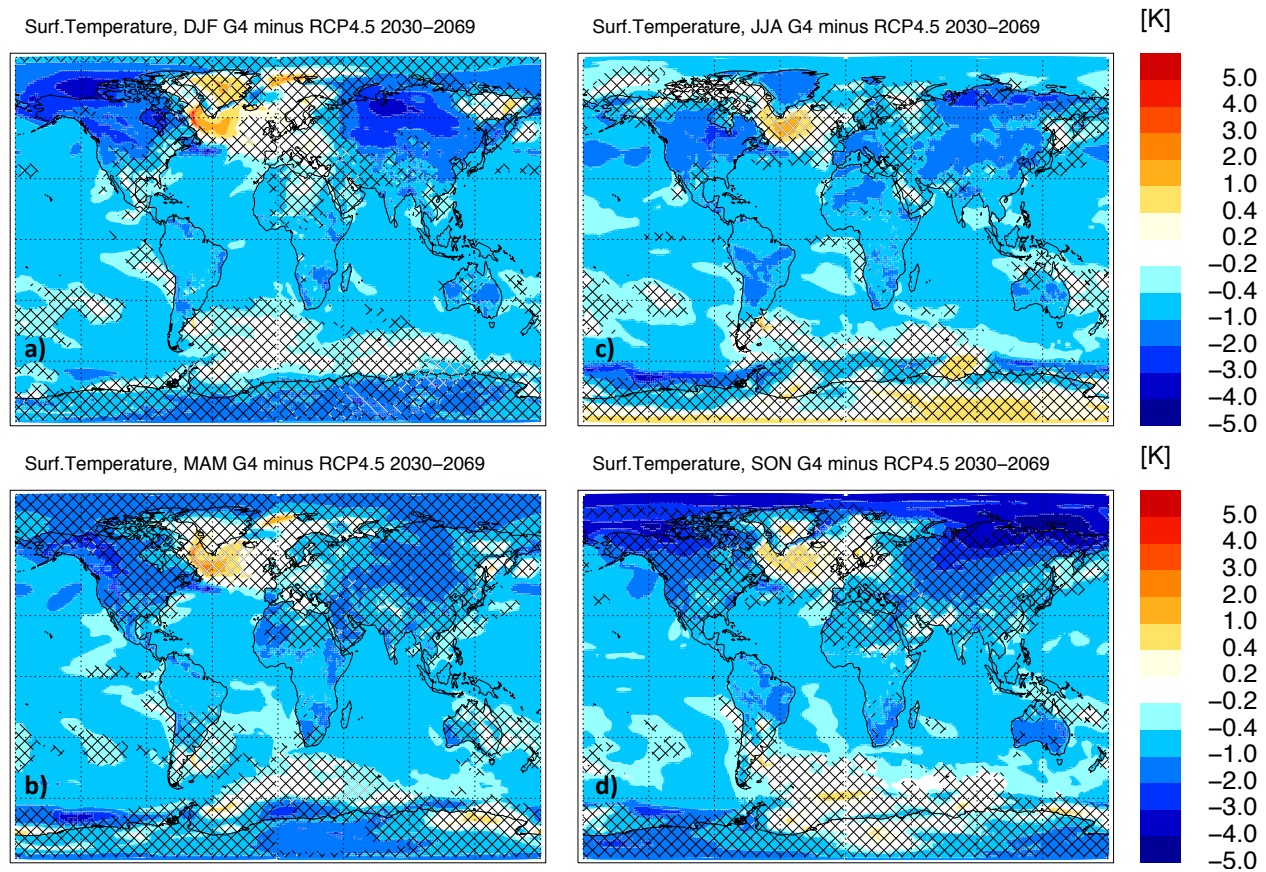

**Figure 5.** Seasonally averaged surface temperature anomalies G4-RCP4.5 (K) from the atmosphere-ocean coupled model CCSM-CAM4 (time average 2030-2069). The shaded areas are not statistically significant within $\pm 1\sigma$. Panels (a-d) refer to: December-January-February (a); March-April-May (b); June-July-August (c); September-October-November (d).

calculating on-line (in a fully coupled approach) the Ts perturbation produced by the radiative flux changes due to the stratospheric sulfate aerosol imbalance with respect to the CCSM-CAM4 distribution in the G4 case. In addition, we also include in the radiative balance the SG-driven indirect perturbation of greenhouse gases ($O_3$, $H_2O$, $CH_4$ and $CO_2$ from the changing methane oxidation), as well as of upper tropospheric ice particles. This on-line calculated Ts perturbation is then added to the externally provided Ts field from CCSM-CAM4 for the G4 experiment. Table S1, Fig. S2 and Fig. 6 document these radiative flux changes and their impact on the calculated Ts.

Surface temperature changes due to the above discussed indirect SG effects are calculated from the instantaneous perturbation of radiative fluxes, which is of course an exact procedure over continents and polar ice caps, whereas is only approximate over the oceans. On the other hand, as explained above and clearly visible in Table S1, Fig. S1 and Fig. 6, the radiative pertur-

bation additive to the dominant one (i.e., the one produced by stratospheric sulfate aerosols in the CCSM-CAM4 simulation) is normally small, both globally and locally (notice the different color scale between Fig. 6b and Fig. 5). Only the ice induced changes of Ts may be comparable in magnitude to those from the stratospheric aerosols, but limited to tropical continental surfaces, where UT ice may have significant optical depth values. On the other hand, the SST calculated changes due to chemistry and ice indirect effects of SG are usually smaller, so that the impact of our approximation may be expected to be negligible.

Together with the G4 simulation, a sensitivity case (G4K) was run, with surface temperatures fixed at the RCP4.5 Base values. Here, the experimental approach is similar to that of Kuebbeler et al. (2012) who ran a G4 simulation with a 5 Tg-$SO_2$/yr injection and prescribed sea surface temperatures and sea ice from the RCP4.5 Base case. This is done not only to highlight the role of the tropospheric temperature perturbations in cirrus ice formation (given a certain vertical velocity change) but mostly to calculate the updraft sensitivity to different conditions of tropospheric stabilization introduced by the stratospheric sulfate aerosol injection.

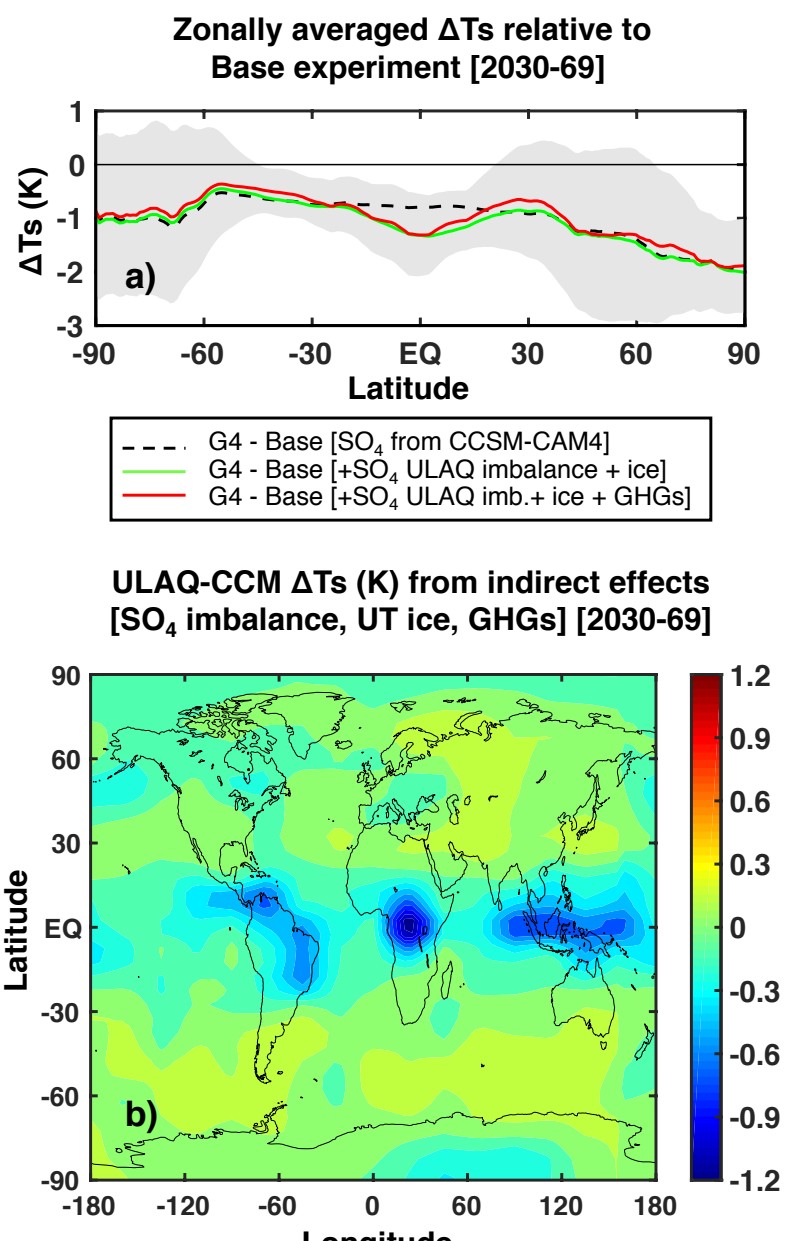

**Figure 6.** Panel (a): zonally averaged Ts anomalies G4-RCP4.5 (K), under different conditions for the G4 perturbed case (time average 2030-69): from the atmosphere-ocean coupled model CCSM-CAM4 (black dashed line); as above, but adding the Ts anomalies in the ULAQ-CCM with on-line coupling of cirrus ice changes and the $SO_4$ imbalance between CCSM-CAM4 and ULAQ-CCM (green line); as above, but adding also the Ts anomalies in the ULAQ-CCM with on-line coupling of GHG changes (red line) (see text and legend). The shaded area represents $\pm 1$ $\sigma$ of the zonally averaged Ts over the 40-year period. b) Lat-lon distribution of the Ts anomalies (K) calculated on-line in the ULAQ-CCM model considering cirrus ice changes, the $SO_4$ imbalance between CCSM-CAM4 and ULAQ-CCM and GHG changes (time average 2030-2069).

## 3 Model response to sulfate geoengineering

In this section, we will show the ULAQ-CCM response to the stratospheric sulfate injection. Some of the perturbations have already been discussed in previous works, in particular regarding stratospheric dynamics changes (Pitari et al. (2014); Visioni et al. (2017b)). Here, we will focus on the thermo-dynamical changes in the upper troposphere and, consequently, on changes in the formation of cirrus ice clouds.

### 3.1 Thermo-dynamical changes in the troposphere

Figure 7 shows the differences in temperature and updraft in G4 and G4K with respect to the Base case. In G4, we observe a tropospheric cooling of $\simeq$1-2 K in the ice formation region throughout all latitudes, while the warming due to the sulfate aerosol absorption of shortwave and longwave radiation is confined above the tropopause (Fig. 7a). When surface temperatures are kept fixed at the RCP4.5 baseline values with the SG perturbation (G4K case), the upper troposphere and lower stratosphere temperature anomalies look very different (Fig. 7b). The tropospheric cooling is absent and the stratospheric warming produced by absorption of longwave planetary and near-infrared solar radiation is more uniformly spread across the lower stratosphere, with some penetration also in the UT ($\simeq$0-1 K). The latter is due to the sulfate aerosol cross-tropopause fluxes that are due to the large-scale transport (at mid-latitudes) and gravitational sedimentation (mostly relevant in the tropical region).

The updrafts responsible for the upper tropospheric ice particle formation result from the sum of a rather small large-scale vertical velocity contribution (on the order of 1-2 cm/s) and a dominant part due to motions associated with synoptic scale disturbances and gravity waves (on the order of 10-20 cm/s); the latter is calculated as a function of the TKE (Lohmann and Karcher (2002)) with the exact formulation reported in Eq. 5:

$$w_{TOT} = w_{LS} + 0.7\sqrt{TKE} \tag{5}$$

The vertical velocity is reduced in G4 with respect to the Base case by $\simeq$1-2 cm/s in the whole UT (Fig. 7c) (on the order of -10%, as visible in Fig. 8), due to the atmospheric stabilization caused by a reduction in the temperature vertical gradient.

Fig. 9a shows the average tropical vertical profiles of the $SO_4$ mixing ratio (in the particulate phase), for both the Base and SG experiments (with an 8 Tg-$SO_2$ injection). The changes in zonally averaged net heating rates, temperatures and zonal winds are also shown in Fig. 9, panels (b), (c) and (d), respectively. They help explain how the SG sulfate perturbation may act as driver for dynamical changes in the UT, with significant effects on ice particle formation.

In Fig. 9a, it is interesting to note a somewhat smaller tropical aerosol confinement in the G4K case. This is consistent with the findings of Visioni et al. (2017b): the aerosol-driven surface cooling in G4 (contrary to G4K) favours a decreased wave activity and a consequent decrease in poleward mass fluxes from the tropical reservoir, for both gas and aerosol species. On the other hand, the increased $H_2SO_4$ tropical amount available for aerosol formation tends to produce larger particles with smaller equivalent optical thickness (see Niemeier and Schmidt (2017); Visioni et al. (2018)). In light of this, smaller stratospheric heating rate anomalies are calculated in G4 with respect to G4K (Fig. 9b): in the latter case, we then expect an enhanced temperature increase in the tropical lower stratosphere (Fig. 9c), coupled to a slight tropospheric warming due to the SG aerosol

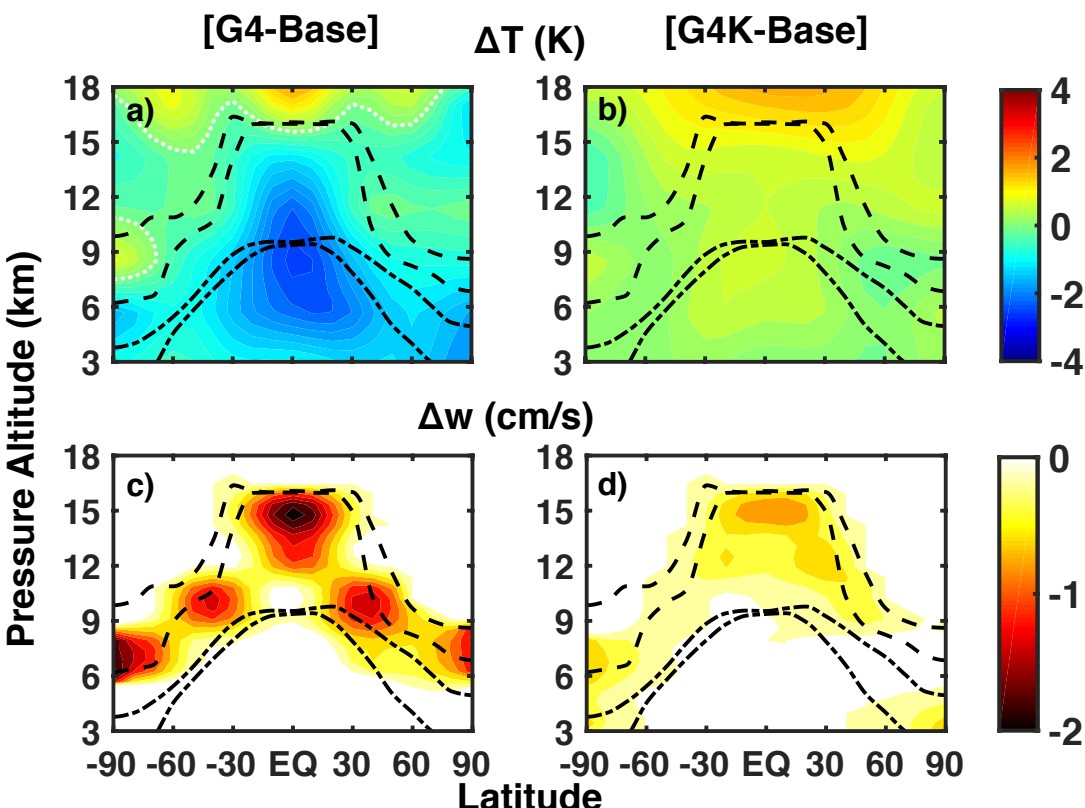

**Figure 7.** Zonally and time-averaged changes of temperature (panels a,b) and vertical velocity (panels c,d) in experiments G4 (panels a,c) and G4K (panels b,d) with respect to the Base case (years 2030-69). The dashed lines show the mean tropopause height (with seasonal variability). The dash-dotted lines show the mean height at whichT=238 K (with seasonal variability). The dotted white lines in panel a) highlights where $\Delta T=0$ K.

sedimentation below the tropopause. The latter, on the other hand, results to be greatly overbalanced by mid-upper tropospheric cooling in G4, due to less intense latent heat exchange resulting from the aerosol-driven Ts decrease (contrary to G4K). As a result, the G4 atmosphere is more efficiently stabilized with respect to G4K, and the positive/negative anomalies of T/u shears in the UT (Fig. 9cd) favour a decrease of the TKE (and updraft velocities) in G4 with respect to G4K (Fig. 7cd).

5  All features of the SW and LW heating rate anomalies in Fig. 9b can be fully explained taking into account the aerosol-$O_3$ coupled effects (Pitari et al. (2014)). The sign of tropical ozone changes under the SG conditions depends on altitude. The $O_3$ decreases below ~25 km and increases above this height; this helps explain the positive/negative heating anomalies in SW and LW components above 25-km altitude.

10  The SG induced reduction of updraft velocities is significantly smaller in the G4K case ($\simeq$0.5 cm/s, on the order of -3% the baseline values), as clearly visible in Fig. 7d. This will represent the major change in our approach to studying the UT ice

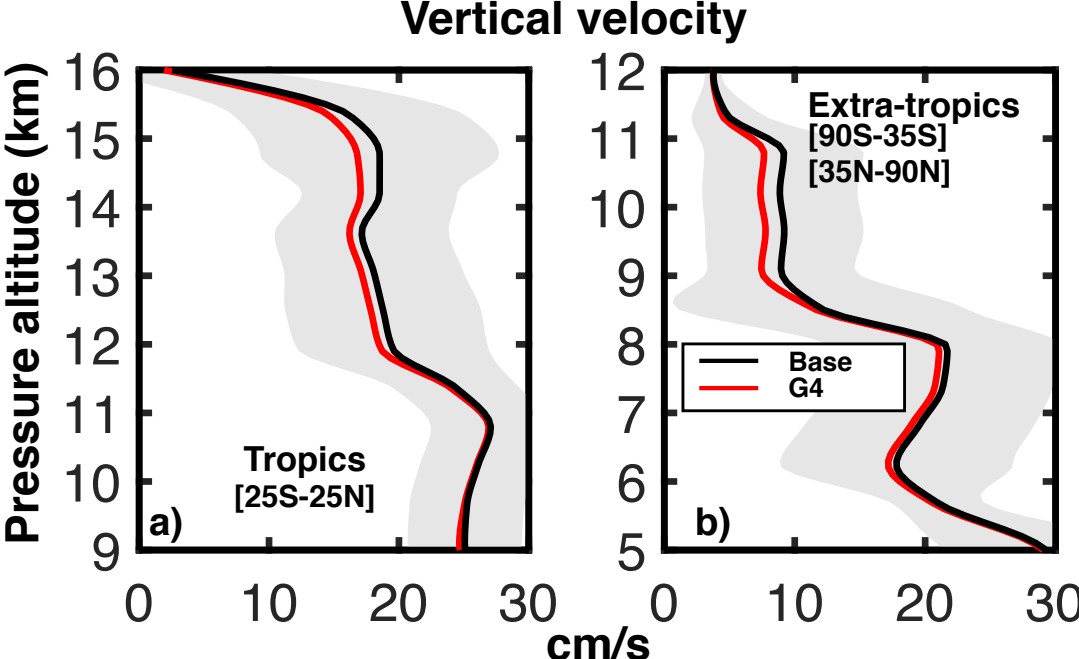

**Figure 8.** Average upper tropospheric profiles of the vertical velocity (cm/s) in G4 and Base experiments (years 2030-69). Panels (a) and (b) are for the tropics and extratropics, respectively (see legends). The vertical velocity w is obtained as the sum of the large-scale value and that calculated as a function of the TKE (see Lohmann and Karcher (2002) and Eq. 5), which essentially accounts for the synoptic scale and gravity wave motions. The shaded areas of the same color represent $\pm\,1\sigma$ for the ensemble over the 40-year period 2030 to 69.

sensitivity to SG with respect to the one adopted in Kuebbeler et al. (2012). According to our calculations, when taking into account both the main radiative effects of geoengineering stratospheric aerosols (i.e., lower stratospheric heating on one hand, surface and tropospheric cooling on the other hand), the resulting impact on tropospheric turbulence and updraft is significantly enhanced with respect to the case in which only the stratospheric warming is considered. A noticeable difference in the G4K
5   w-anomalies with respect to those of G4 is at low altitudes over the polar regions, where the G4K negative values are larger than in G4. This may be largely explained by the increasing longitudinal variability of surface temperatures in the G4 case, mainly in the sub-Arctic region (see previous discussion relative to Fig. 5).

The tropical and extratropical average profiles of the updraft velocity are shown in Fig. 8 for both the Base and G4 conditions. The G4K curve (not shown) is intermediate between the previous two. The pronounced variability of the vertical velocity is
10   expected as a consequence of time, latitude and longitude fluctuations of the TKE. This will produce a significant dispersion of the ice particle size distribution (see ahead in Section 3.2).

**Figure 9.** Average tropical vertical profiles (25S-25N, years 2030-69) of the SO$_4$ volume mixing ratio for G4, G4K and Base experiments (ppbv, panel a); G4-Base changes of net, shortwave and longwave heating rates (K/day, panel b) (LW is calculated with temperature fixed at Base values) (net heating rate changes are also shown for G4K-Base, with the blue line); G4-Base and G4K-Base temperature changes (K, panel c); G4-Base and G4K-Base changes of mean zonal winds (m/s, panel d). The shaded areas of the same colour represent $\pm 1 \sigma$ for the ensemble over the 40-year period 2030 to 69.

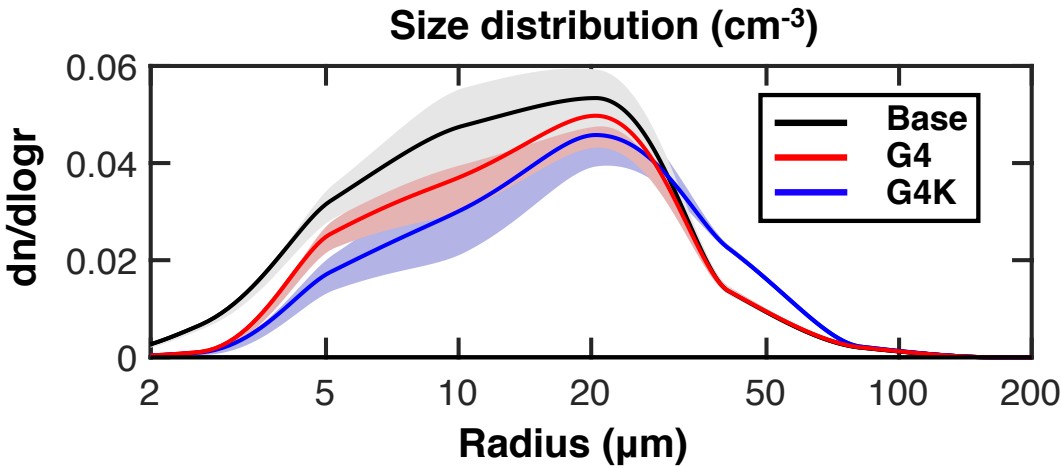

**Figure 10.** Globally and time-averaged number density values of ice crystals as a function of particle radius (dn/dlogr, cm$^{-3}$) (years 2030-69). Shaded areas of the same colour represent $\pm 1\ \sigma$ for the ensemble over the 40-year period 2030-69. The calculated global mean values of the ice particle effective radius are as follows: Base $\rightarrow$ 31.2$\pm$3.2 $\mu$m; G4 $\rightarrow$ 33.5$\pm$3.6 $\mu$m; G4K $\rightarrow$ 36.8$\pm$4.1 $\mu$m. The reference MODIS value in Table 2 is 33.4$\pm$2.1 $\mu$m.

## 3.2    Tropospheric ice perturbations due to sulfate geoengineering

In Section 2.2.2, we showed that the ULAQ-CCM parametrization for ice particle formation through both homogeneous and heterogeneous freezing produces a spatial distribution of the UT ice particles reasonably comparable to available data in terms of ice number concentration, OD, mass mixing ratio and effective radius. We now move to analyse the model-calculated SG
perturbation of some of these quantities by comparing the G4 and G4K simulations against the Base case. As we have previously discussed and shown in Fig. 7-9, these perturbations are essentially produced and regulated by decreasing vertical velocities (-1.7 cm/s and -0.8 cm/s, in the tropical region below the tropopause, for G4 and G4K, respectively) and by changing the tropospheric temperatures (-1.2 K and +0.5 K, in the tropical UT region, for G4 and G4K, respectively).

The model-calculated globally and time-averaged size distribution of the ice particles is presented in Fig. 10 for the three experiments, along with their globally averaged effective radius. A significant change in size distribution is highlighted in Fig. 10 in both SG experiments with respect to not only the Base case, but also G4 and G4K. The common feature in both SG cases is the expected decreased particle population over the whole radial spectrum with respect to the Base experiment. This is due to the increased atmospheric stabilization forced by the SG aerosols with reduced updraft velocities and consequent decrease
of the UT ice supersaturation probability.

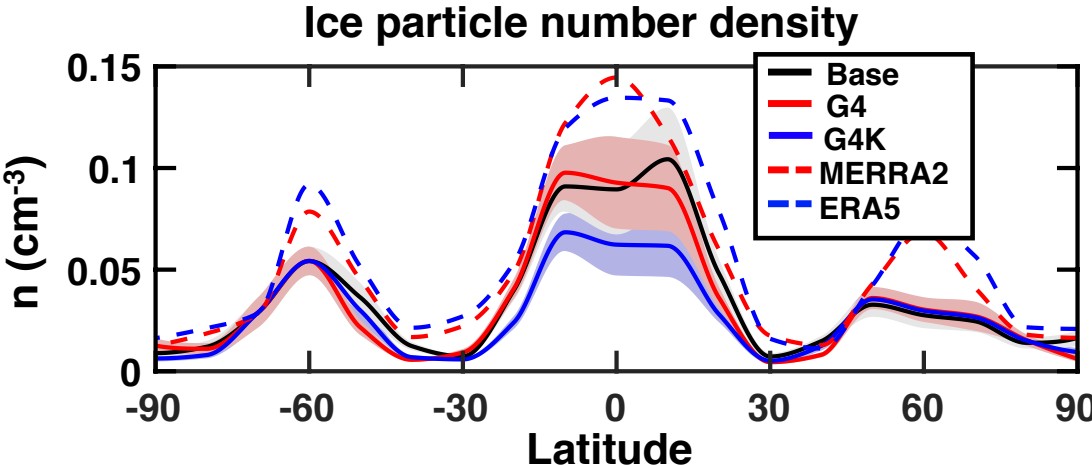

**Figure 11.** Zonally and time-averaged total number density values of ice crystals as a function of latitude (n, cm$^{-3}$) (years 2030-69), as calculated in the ULAQ-CCM (for Base, G4, G4K experiments) and compared with indirectly derived values from the MERRA-2 and ERA5 ice mass mixing ratio and ULAQ-CCM (Eq. 4). Number densities are calculated at pressure layers 150-200 hPa for 25S-25N, 200-250 hPa for 25-35 (N/S), 250-300 hPa for 35-45 (N/S), 300-350 for 45-55 (N/S) and 350-400 for 55-90 (N/S).

The UT temperature anomalies, however, are very different in the two SG experiments with respect to the Base case (see Fig. 7). As a consequence of this, the tropospheric cooling produced in G4 by the Ts adjustment to the stratospheric aerosol negative RF favours a number density increase of ice particles with respect to the G4K experiment but is still less than in the Base case (see also Fig. 11), due to the dominant impact of the reduced updraft. Cooler temperatures, in fact, cause a faster nucleation of the ice particles, quickly removing water vapour available for the freezing itself and limiting the condensational growth of ice particles (Kuebbeler et al. (2012); Visioni et al. (2017a)). At the same time, the velocity and temperature negative anomalies partially compensate each other also in the particle size spectrum, with a resulting effective radius in G4 larger with respect to the one in the unperturbed atmosphere (33.5±3.2 $\mu$m and 31.2±3.2 $\mu$m, respectively) but smaller than that in G4K. In this latter case, the UT is slightly warmed up with respect to the Base case (see Fig. 7) so that both the velocity and temperature anomalies tend to increase the particle size (36.8±4.1 $\mu$m). Globally, the ULAQ-CCM baseline values of the effective radius fall well inside the MODIS range of variability (33.4±2.1 $\mu$m).

As visible in Fig. 11 the calculated ice number densities follow the zonal mean behaviour of the MERRA-2 and ERA5 indirectly derived values, with the previously discussed underestimation tendency, mainly in the tropical region (see Fig. 4).

### 3.2.1 Optical depth

The ice extinction anomalies of G4-Base that are calculated in the ULAQ-CCM are negative in the whole UT (Fig. 12ab) due to the decreasing number density of the particles caused by the reduced vertical velocities in the SG dynamical conditions (see

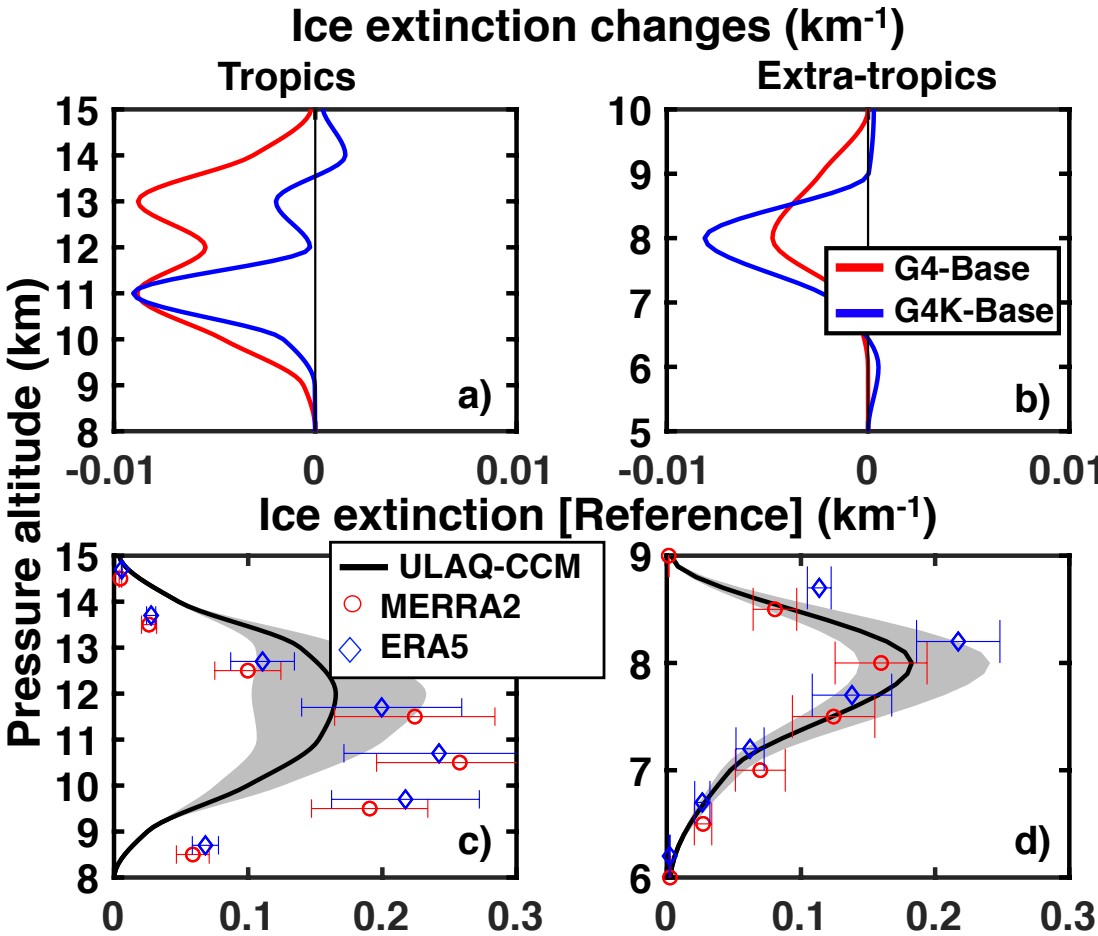

**Figure 12.** Average upper tropospheric profiles of ice particle extinction ($\lambda$=0.55 $\mu$m) (km-1) for the tropics (25S-25N) and extratropics (35S-90S, 35N-90N) in panels (a,c) and (b,d), respectively. Panels (a,b): ice extinction changes for G4-Base (red curves) and G4K-Base (blue curves) (years 2030-69). Panels (c,d): comparison of ULAQ-CCM calculated values of ice extinction with indirectly derived values from the MERRA-2 and ERA5 ice mass mixing ratio and ULAQ-CCM effective radius (red and blue circles) (see text). The time average is over the years 2003-2012. The shaded areas represent $\pm 1\ \sigma$ for the ensemble over the 10-year period.

Fig. 7-8). Although the UT cooling in G4 tends to partially offset the effects of the updraft decrease on the ice particle number density, the overall impact is of a general decrease of the UT ice extinction and is even more pronounced than in G4K where the tropospheric cooling is not taken into account. In the latter case, however, the particle effective radius is larger than in G4, as discussed above for Fig. 10. These size distribution changes affect not only ice extinction, but also the shortwave and longwave radiative responses per unit optical depth (see ahead Section 3.2.2).

Following the procedure described in Section 2.2 (see Eq. 3), an evaluation of the model calculated ice extinction profiles is attempted (Fig. 12cd). This is made using indirectly derived values from the MERRA-2 and ERA5 ice mass mixing ratio and the ULAQ-CCM effective radius, as in Eq. 6 below. Here, $\chi_{ext,i}$ is the ice extinction at the $i^{th}$ vertical layer and $\rho_{atm,i}$ is the atmospheric mass density at the same vertical layer:

$$\chi_{ext,i} = Q_{ext}\frac{3}{2}\frac{\rho_{atm,i}}{\rho_{ice}}\frac{\chi_i}{r} \tag{6}$$

The ULAQ-CCM tropical underestimation of the ice extinction below 13 km is consistent with that of the ice number density and is partly justified by the specific assumptions made on cirrus cloud formation in the model, as pointed out in the discussion of Fig. 4.

The net result on the ice optical depth (i.e., the vertical integral of ice extinction) is shown in Fig. 13. In general, a latitude-dependent OD reduction comparable to that found in Kuebbeler et al. (2012) is present in G4K, while in the G4 case (as expected from the extinction anomalies) a further decrease is calculated mainly in the tropics, even though the UT temperatures are cooler. The effects regarding the temperature and updraft cannot be easily separated, but the colder tropospheric temperatures in G4 with respect to G4K reduce the particle size increase respect to the Base case, producing an additional decrease in the optical depth. The coupled effects of the velocity and temperature anomalies on the ice particle number density and size produce the most relevant impact in our study, pointing out the importance of allowing surface temperatures to respond to the stratospheric aerosol radiative forcing.

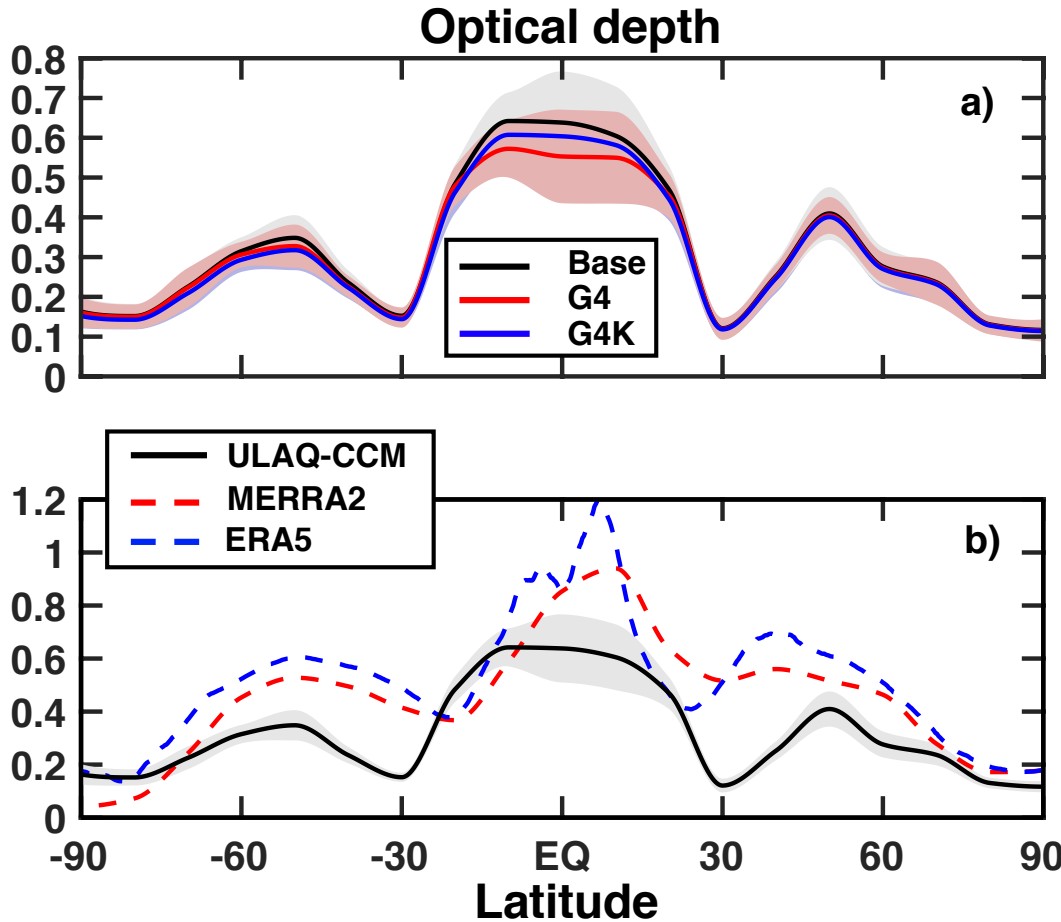

**Figure 13.** Zonally and time-averaged values of the ice optical depth ($\lambda$=0.55 $\mu$m) for the ULAQ-CCM Base, G4 and G4K experiments (solid black, red and blue lines, respectively) (panel a) and Base case comparison with the MERRA-2 and ERA5 indirectly derived values (dashed red and blue line) (panel b). The model results are for years 2030-69; the MERRA-2 and ERA5 data are for years 2003-12. The shaded area represents $\pm 1$ $\sigma$ for the ensemble over the 40-year period 2030-69.

### 3.2.2 Consequences on radiative forcing

The well-tested radiative transfer code on-line in the ULAQ-CCM (Chou (2001); Randles et al. (2013); SPARC (2013)) has been used to calculate the shortwave and longwave components of the tropopause radiative forcing due to SG aerosols (direct forcing) and to UT ice changes (indirect forcing). As discussed so far, the latter are largely produced by the SG-driven dynamical perturbations on the homogeneous freezing process for ice formation. The ice radiative effects have been calculated using up-to-date wavelength-dependent refractive index available in the literature (Warren (1984); Warren and Brandt (2008); Curtis et al. (2005)) and compared against previous results under similar conditions, such as those by Schumann et al. (2012). All the radiative calculations shown in this section have been performed off-line with the same radiative transfer code as the one present on-line in the ULAQ-CCM model, in order separate the effects of the single components analyzed.

The results are shown separately for the G4 and G4K experiments, both with respect to the RCP4.5 Base case. Following the previously discussed thinning of the UT ice clouds, a positive SW RF is calculated because of the decreased scattering of the incoming solar radiation by the ice particles. However, such an effect is largely covered by the negative LW RF due to a lessened capacity of the ice particles to trap outgoing planetary radiation; therefore, the obtained net effect on RF is negative, as shown in Table 3. This indirect negative RF is smaller but still significant when compared to the negative direct net RF due to the SG aerosols (∼30% of it).

It is interesting to note that the shortwave component of the ice RF is indeed smaller than the longwave component, however, not as much as one could expect from the very different normalized RFs (i.e., forcing per unit OD) at a given particle radius. The reason is that both the SW and LW normalized RFs are decreasing with the increasing particle radius, but the relative changes of these normalized RF components are significantly different between the SW and LW. According to our radiative calculations, the SW normalized values decrease (in magnitude) from -12.1 W/m$^2$ to -5.7 W/m$^2$ (-53%) with the ice effective radius increasing from 15 $\mu$m to 40 $\mu$m, whereas the instantaneous LW normalized RF remains quasi-constant on an average value of +53 W/m$^2$, with a smooth 3% decrease over the same radius interval. The resulting SW RF is then controlled not only by the negative OD changes (-0.020 in G4 and -0.012 G4K) but also by the magnitude of the particle radius increase, which is larger in G4K than in G4, both with respect to the Base case (see discussion of Fig. 10).

**Table 3.** Top three rows: globally and time-averaged values of the upper tropospheric ice optical depth changes and RF differences (W/m$^2$) between the SG perturbed experiments and the RCP4.5 Base case due to changes in ice crystal concentration and size. Middle three rows: globally averaged values of stratospheric sulfate aerosol optical depth changes and RF differences (W/m$^2$) defined as above due to changes in aerosol concentration and size. Bottom three rows: total OD and RF changes (i.e., ice + sulfate). All results are for all-sky conditions (i.e., including the presence of background cloudiness) and with an 8 Tg-SO$_2$/yr injection. The RFs are calculated at the tropopause with temperature adjustment. The time average is over the years 2030-69.

| Exp [all sky] | Ice OD change | RF SW | RF LW | RF Net |
|---|---|---|---|---|
| G4-Base | -0.024 ± 0.003 | +0.50 | -0.79 | -0.29 ± 0.04 |
| G4K-Base | -0.012 ± 0.001 | +0.35 | -0.49 | -0.14 ± 0.02 |
| **Exp [all sky]** | **SO$_4$ OD change** | **RF SW** | **RF LW** | **RF Net** |
| G4-Base | +0.079 ± 0.003 | -2.03 | +0.86 | -1.17 ± 0.06 |
| G4K-Base | +0.083 ± 0.003 | -2.14 | +0.90 | -1.24 ± 0.06 |
| **Exp [all sky]** | **Total OD change** | **RF SW** | **RF LW** | **RF Net** |
| G4-Base | -0.024+0.079 | -1.53 | +0.07 | -1.46 ± 0.10 |
| G4K-Base | -0.012+0.083 | -1.79 | +0.33 | -1.38 ± 0.08 |

**Table 4.** Rearrangement of the results presented in Table 3, with the calculated cloud adjustments (bottom three rows) to clear-sky RF components (top three rows). The cloud adjustments for the SW and LW RF contributions are shown separately for the mere presence of background atmospheric clouds (left) and for the cirrus thinning (right): the former is calculated as the difference between the all-sky and clear-sky aerosol RFs, with the all-sky including the background warm clouds and fixed UT ice clouds.

| Exp [clear sky] | RF SW | | RF LW | | RF Net |
|---|---|---|---|---|---|
| G4-Base | -3.13 | | +1.07 | | -2.06 |
| G4K-Base | -3.30 | | +1.14 | | -2.16 |
| **Cloud adjustment** | **RF SW** | | **RF LW** | | **RF Net** |
| G4-Base | +1.10 | +0.50 | -0.21 | -0.79 | +0.60 |
| G4K-Base | +1.16 | +0.35 | -0.24 | -0.49 | +0.78 |

Table 4 presents, in a compact form, the globally and time averaged ULAQ-CCM results for the cloud adjustments of clear-sky RF components due to the SG stratospheric aerosols. The SW and LW cloud adjustments are roughly comparable to the ones calculated in Kuebbeler et al. (2012) (+1.11 W/m$^2$ and -0.51 W/m$^2$, respectively, calculated at the top of atmosphere for an SG experiment with a 5 Tg-SO$_2$/yr injection). These numbers could be compared with those obtained in the ULAQ-CCM G4K case (although for an 8 Tg-SO2/yr injection), i.e., +1.51 W/m$^2$ and -0.73 W/m$^2$ for SW and LW, respectively, with a net value of +0.52 W/m$^2$ against +0.60 W/m$^2$ in Kuebbeler et al. (2012).

In the (more realistic) G4 simulation performed by the ULAQ-CCM model, the SW cloud adjustment is only slightly smaller

than in the G4K, while a significantly larger negative LW component is calculated. This ends up in a net adjustment of +0.52 W/m$^2$ in the G4 against +0.72 W/m$^2$ in the G4K experiment. A latitude-dependent view of these results is presented in Fig. 16. The black solid line shows the net positive adjustment (SW+LW) due to the mere presence of background clouds, which substantially alter the radiative fluxes (see also Kuebbeler et al. (2012); Schulz et al. (2006); Stier et al. (2013)). These clouds are kept fixed in the ULAQ-CCM model, using climatological values, and thus do not present changes under the G4 scenario. An estimate of the all-sky RF contribution due to SG-driven changes of background clouds is beyond the purposes of the present study. According to our model calculations, the negative LW is the dominant component of the cloud adjustment due to cirrus ice thinning, and this is particularly true for the more realistic G4 simulation. In this latter case, significantly larger values of the LW adjustment are found over the tropics with respect to G4K, consistent with the ice extinction profile changes in Fig. 12a.

Further informations on the model calculated RFs are shown in Fig. S3, where we show both the Clear-Sky latitudinal distribution of the sulfate aerosols RF (Fig. S3a) for both G4 and G4K and the LW and SW cloud adjustment due to the presence of background clouds for both G4 and G4K (Fig. S3b).

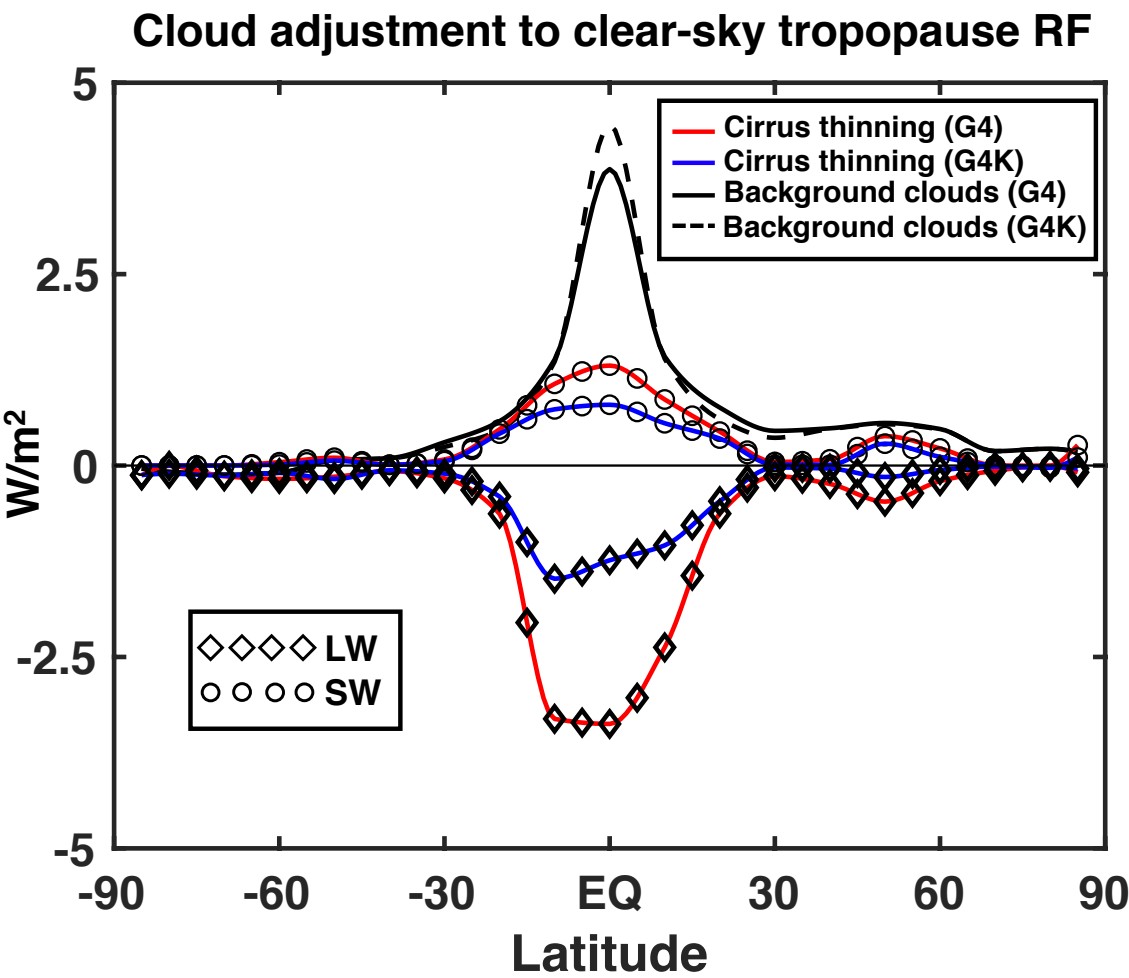

**Figure 14.** Zonally averaged cloud adjustments to the clear-sky SG aerosol RF (W/m$^2$), as a function of latitude (time average 2030-2069). See legends for line meaning. The positive adjustment due to (passive) background clouds (black solid line for G4, black dashed line for G4K) shows the net value (SW+LW), which is, however, largely controlled by the SW contribution (see Table 4 and Fig. S3).

## 4 Conclusions

Sulfate geoengineering is considered, amongst other solar radiation management (SRM) techniques, one of the most promising. One reason for this (and unlike other methods) is that we have a natural proxy for the stratospheric sulfate injection, i.e., past explosive volcanic eruptions in the tropical belt. This does not mean that SG does not still pose some scientific questions that need to be answered thoroughly, as pointed out by MacMartin et al. (2016). For instance, models still show many significant differences regarding the confinement of stratospheric sulfate aerosols in the tropical pipe Pitari et al. (2014).

In recent years, some experiments have been proposed where SG is used to meet different climate targets (MacMartin et al. (2017); Kravitz et al. (2017)). However, to properly do so, a clear understanding is needed of how multiple side effects of this technique can modify the net RF (Visioni et al., 2017a). While some of these effects produce a negligible difference in forcing, such as those from gas species perturbations ($CH_4$, $O_3$, stratospheric $H_2O$) (Visioni et al. (2017b)), this might not be the case for changes produced in the formation of thin cirrus ice clouds.

This latter indirect effect was already analysed in two previous works. Cirisan et al. (2013) looked at the potential impact of IN changes in the UT, finding a negligible positive TOA forcing (+0.02 W/m$^2$, up to 0.04 W/m$^2$) due to the number density increase of $H_2SO_4$-$H_2O$ aerosols transported down in the UT from the lower stratosphere. Kuebbeler et al. (2012), on the other hand, have studied the effects of dynamical changes caused by the aerosol-induced stratospheric warming and their consequences on UT ice formation via homogeneous freezing.They found a considerable negative TOA forcing in the longwave spectrum (-0.51 W/m$^2$), greatly attributable to the SG-induced ice optical depth reduction. In the present study, we focused on these same indirect dynamical effects, adding the potential impact of the SG aerosol-induced surface cooling (G4 experiment), which was not explicitly considered in the study of Kuebbeler et al. (2012). Their approach was also included for comparison in our study, by means of a sensitivity study (G4K) conducted with the ULAQ-CCM, where we keep the surface temperature fixed at the RCP4.5 baseline values so that we can quantify more precisely the surface cooling impact on the UT thin cirrus clouds.

A compact view of the SG effects on UT ice formation is presented in Fig. 15. On one hand, the aerosol-induced stratospheric warming and surface cooling combined together produce a further atmospheric stabilization with an even larger reduction in tropospheric updraft with respect to the G4K case. This lowers the UT probability for ice supersaturation, with less favourable conditions especially for homogeneous freezing. On the other hand, this ice formation limiting effect is partially counterbalanced by the convectively driven tropospheric cooling, which is not observed in the G4K case.

The resulting changes in ice particle number density and size distribution, when combined, translate into a globally averaged decrease of the ice optical depth ($\Delta\tau$=-0.024, at $\lambda$=0.55 $\mu$m), i.e., -6% of the baseline OD. This reduction is larger than the one in G4K relative to the Base case ($\Delta\tau$=-0.012, -3%), pointing to the dominant and controlling role of the reduced updraft velocities. According to our model results, these OD changes (coupled to increases in ice particle effective radii) translate in net tropopause RFs of -0.29 W/m$^2$ and -0.14 W/m$^2$, for G4 and G4K experiments, respectively, produced only by the cirrus

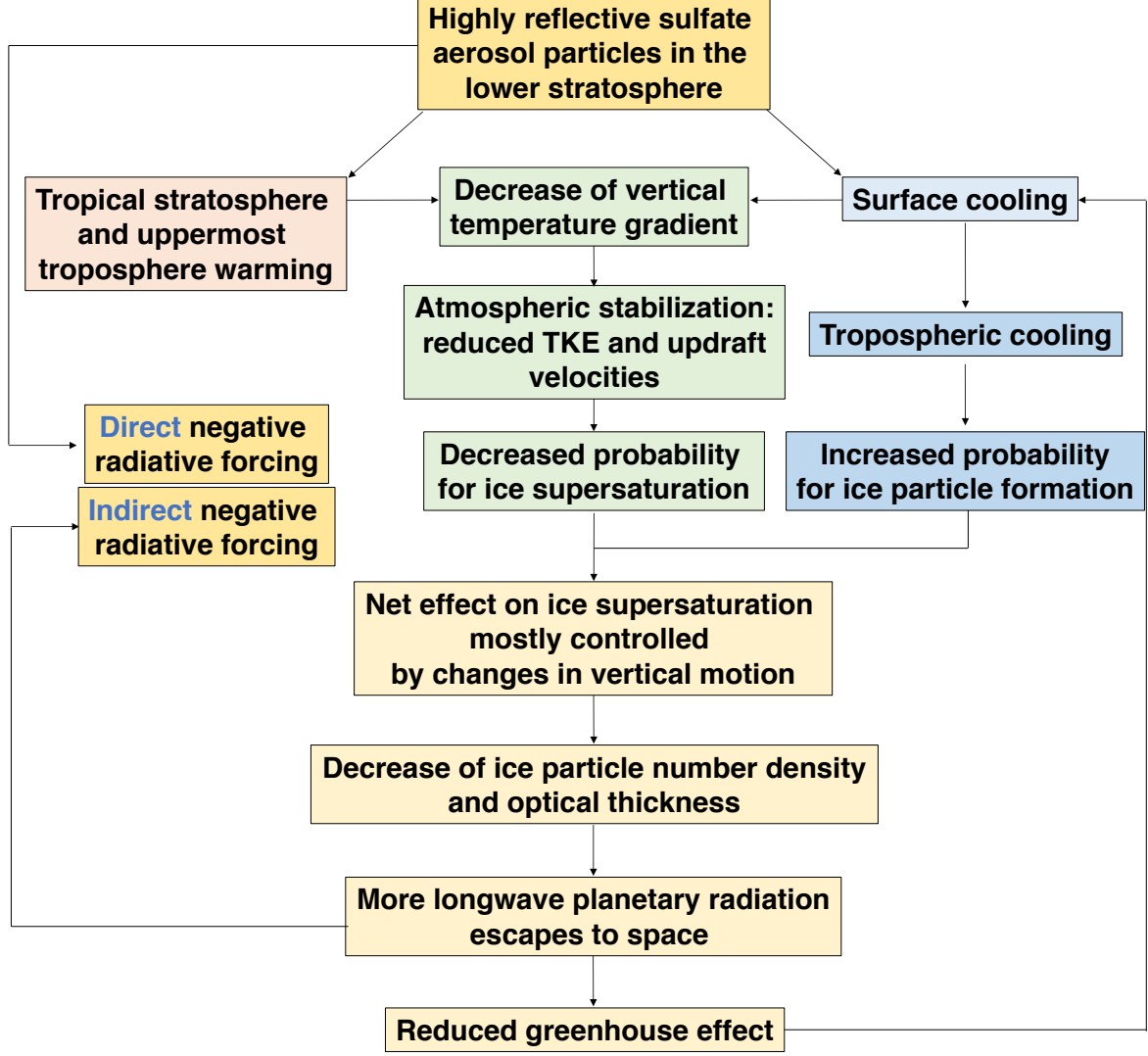

**Figure 15.** Schematic summary of the sulfate geoengineering impact on the dynamical processes driving changes of upper tropospheric ice particle formation through homogeneous freezing.

ice thinning effect of SG. These two cloud adjustments result from a combination of the SW and LW RF contributions, which account for +0.50 W/m$^2$ and +0.35 W/m$^2$ in the SW (for G4 and G4K, respectively) and -0.79 W/m$^2$ and -0.49 W/m$^2$ in the LW (again for G4 and G4K).

We can compare these ice thinning forcing contributions with the net tropopause all-sky RF produced by the stratospheric SG aerosols, i.e., of -1.17 W/m$^2$ and -1.24 W/m$^2$, for the G4 and G4K experiments, respectively. According to our model, the

net negative RF due to the cirrus ice cloud thinning is (in G4) close to 25% of the direct effect of the sulfate particles themselves. This might have consequences in the definition of the sulfate injection efficiency in terms of RF per Tg-S/yr injected, especially if such efficiency is used to determine the amount of $SO_2$ that needs to be injected into the stratosphere to achieve climate targets (MacMartin et al. (2017); Kravitz et al. (2017)).

Fig. 16 summarizes, in a schematic way, the thermo-dynamical processes leading to the changes in cirrus ice formation and the radiative response caused by these changes in the Earth's radiative balance, as analysed in detail in this paper, together with the direct radiative effect of the sulfate particles.

Furthermore, one last consideration is necessary regarding the RFs in the SG scenarios and the unperturbed atmosphere, more specifically, regarding the cloud adjustment to clear-sky RFs due to the stratospheric sulfate aerosols. In our fully interactive aerosol simulation (G4), we obtain a total cloud adjustment (from both cirrus ice thinning and passive background clouds) of +0.60 W/m$^2$ due to compensating large adjustments in the LW and SW. The SW adjustment results in part from the mere presence of (passive) background clouds and in part from the changing size distribution of UT ice particles. The increasing

particle size is more pronounced in the partially interactive aerosol simulation (G4K), thus producing a larger positive SW contribution (+0.78 W/m$^2$).

This latter value is comparable to that calculated in the similar experiment of Kuebbeler et al. (2012) (+0.60 W/m$^2$, with a 5 Tg-$SO_2$ injection). It means that the lower stratospheric warming produced by the SG aerosols acts indirectly on atmospheric dynamics with a strong feedback on the UT cirrus clouds so that a simple reduction of the incoming solar radiation is not a

20 good proxy for the eventual injection of sulfate particles into the stratosphere. When the aerosol-induced surface cooling is coupled to the lower stratospheric warming, the net cloud adjustment is significantly reduced; however, the clear-sky balance of the SW and LW RF contributions is greatly altered by the presence of background clouds coupled to the UT ice thinning.

One important caveat to the conclusions of this study, is that the physical processes behind the UT ice particle formation

are highly idealized in our parameterization. Nonetheless, the results it produces in the reference (historical) simulation are generally comparable with the MERRA-2 and ERA5 reanalysis and some satellite data. In addition, the calculated SG dynamical anomalies in the stratosphere are consistent with those from other modelling studies (Pitari et al. (2014); Niemeier and Schmidt (2017)). Finally, taking into account the consistency with the findings from the study of Kuebbeler et al. (2012), we may reasonably conclude that our results regarding the thinning of the UT ice clouds under SG conditions are sufficiently

robust. However, considering how complex is the balance between the UT ice formation changes and their radiative forcing is (Mitchell et al. (2008)), the results in the present cannot be considered conclusive and exhaustive. Additional results using different and more complete physical parametrizations (both regarding the ice formation processes and a wider range of updraft velocities), together with an on-line ocean coupling, may help clarify the net contribution of ice clouds in a sulfate geoengineering scenario.

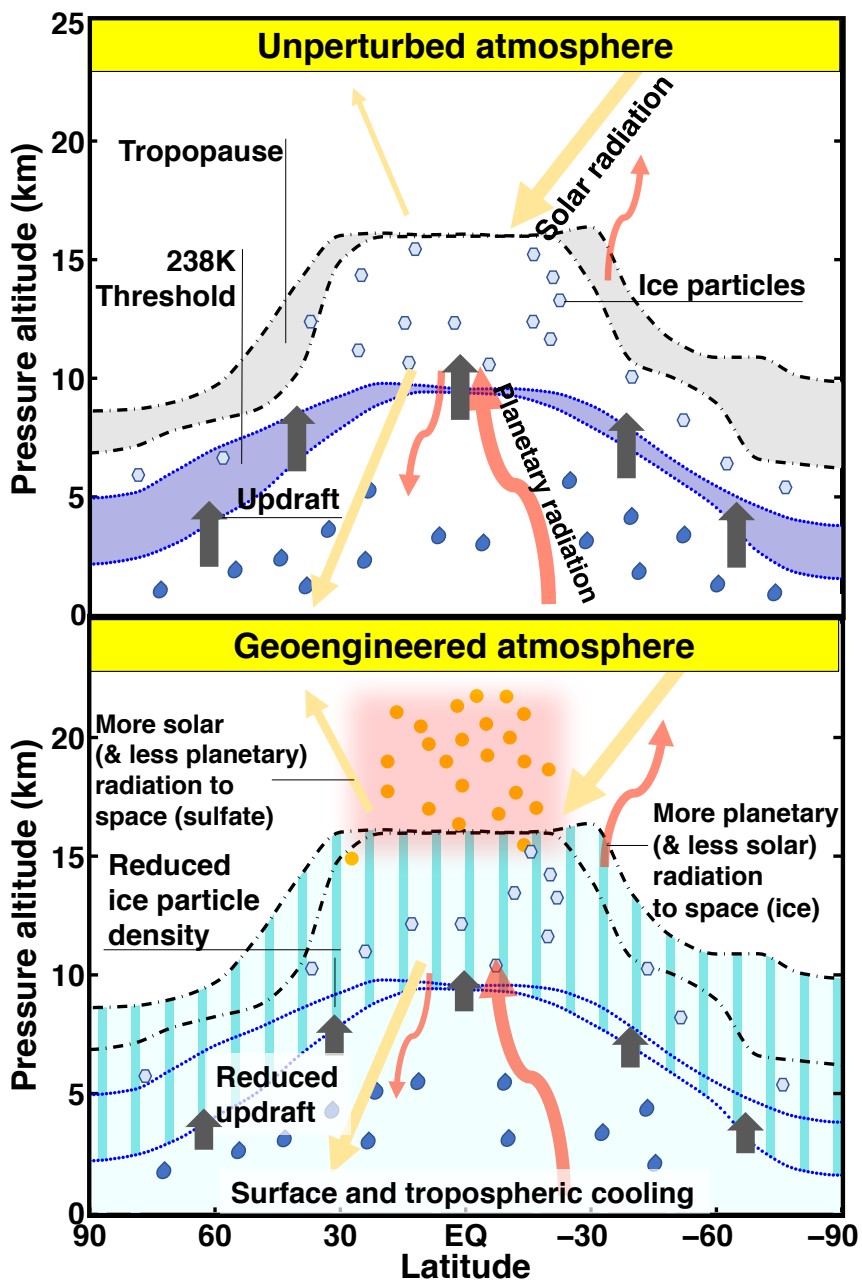

**Figure 16.** Cartoon of the sulfate geoengineering impact on cirrus ice particles formed through freezing and schematic representation of ice and aerosol changes in radiative fluxes.

*Acknowledgements.* Some of the analyses and visualizations used in this study were produced with the Giovanni online data system, developed and maintained by the NASA GES DISC. We also acknowledge the MODIS mission scientists and associated NASA personnel for the production of the data used in this research effort. One of the authors (GP) would like to thank Bernd Karcher for helpful discussions on the physical processes behind aerosol-ice interactions and for providing the heterogeneous freezing numerical code used in the ULAQ-CCM.

5   Finally, the authors are indebted with four anonymous reviewers for their helpful suggestions that improved this study in a significant way.

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
