# Peer review of "Upper tropospheric ice sensitivity to sulfate geoengineering"

_Atmospheric Chemistry and Physics, 2018_

## Referee Comment (RC1) · Anonymous Referee #1 · 15 Mar 2018

This is an interesting and timely study of how geoengineering in the form of stratospheric aerosol injection (SAI) would impact ice clouds in the upper troposphere (UT). Few papers have been published on this poorly understood aspect of SAI, so in that sense the paper is a welcome contribution to the geoengineering discussion. However, the paper is in its current form very unclear when it comes to the representation of central processes in the ULAQ model, poorly structured, and full of incorrect/poor grammar. It need a serious overhaul in all these respects. I further question the validity of the results presented, in light of the coarse resolution of the model, as well as its overly simplistic treatment of UT ice nucleation. I challenge the authors to justify why their results can be trusted despite these shortcomings. Below I've listed some additional major concerns I had about the paper, and thereafter some minor

comments (typos, questions for clarification, etc.). I would like to see all of these concerns addressed before I will consider the paper suitable for publication in ACP.

Major comments:

1) With respect to the paper structure, I found it strange that in Section 2 ("ULAQ-CCM and setup of numerical experiments") some model results in response to SG are presented (in Fig. 3-6), but not until sections 2.1 and 2.1. are descriptions of the model treatment of stratospheric aerosols and ice clouds described. I suggest moving the presentation and discussion of model results until AFTER you've described the model, and to only put content in the various sections that is consistent with the section titles.

2) There is no discussion of the effect of additional SO4 available for homogeneous nucleation in the SG cases, as evident in Fig. 7a. Why is the effect of what appears to be a tripling of SO4 particles that can nucleate ice seemingly negligible? Please explain?

Minor comments:

Abstract: "Goal of. . ." should be changed to "The goal of.."

Abstract: Don't understand what "coupled to" means in this context.

Page 2, line 4, End sentence after "documented.

Page 2, line 9: Add "optimal" before "magnitude and location".

Page 2, line 23: Add reference for the claim that homogeneous nucleation normally dominates cirrus.

formation. "Supersaturation ratio" should be "saturation ratio".

Page 2, line 31: "anyway" is not suitable here. "However" could be an alternative.

Page 2, line 32: cloud optical properties are also important here.

Page 4, line 1: This statement is confusing: sulphuric acid droplets are not ice nuclei. Please clarify.

Furthermore, this statement is not very interesting unless you explain WHY there was no effect on RF. Page 4:Vertical velocity is important for cirrus formation not primarily

because it transport water vapour to the UT, but because it controls the adiabatic cooling rate and thus supersaturation, for a given water vapour content.

Page 4, line 14-17: Catastrophic grammar.

Figure 1: This figure is confusing and not well explained. I don't find it particularly helpful at this stage of the paper, but it could be good as a final figure summarizing the findings in the paper.

Page 5, line 5: Ash  dust is not the same.

Page 6, line 1: Sassen et al. (2008) is a paper on cirrus coverage seen by CALIPSO, so I don't see how that could possibly address UT ice changes.

Page 6/Table 1: The horizontal resolution is extremely coarse - how can we have confidence in changes driven by dynamics in this context?

Page 7, line 4: Clumsy and confusing statement. Suggest writing: . . .a negative anomaly in the Arctic region that is approximately 1 K larger than that of high southern latitudes.

Page 7, line 7: What do you mean by "increasing atmospheric stabilisation"? Do you mean "increasing atmospheric stability?

Page 7, line 16-18: The Antarctic warming is not statistically significant, so I don't see the point in discussing it.

Page 8, line 2: Is the vertical velocity change mainly caused by changes to TKE, or also due to large-scale (resolved) velocity changes. If TKE is very important here, I would like to see vertical profiles of TKE for both simulations.

Page 9, lines 19-20: Discuss here the uncertainty associated with cloud ice in MERRA, which uses highly uncertain cloud parameterisations and incorporates very few ice cloud observations in its reanalysis. It would be better to use CALIPSO/CloudSat retrievals of ice cloud properties.

Page 10, line 3: Given how central UT vertical velocities are to this paper, you need to be clearer about how the calculation of vertical velocity is done, i.e. include equation for vertical velocity as a function of TKE, and clearly state if you put any upper/lower bounds in it.

Page 10, line 4: What justifies the assumption that cirrus clouds form only via homogeneous nucleation? That seems to be in stark contrast to papers that report that cirrus clouds appear to form mainly through heterogeneous nucleation (e.g., Cziczo et al., 2013).

Page 13, line 15: Remove "from".

Page 16, line 13-14: This is inaccurate - homogeneous nucleation sets in at approximately 238K, but NOT through "water vapour freezing", but rather through the spontaneous freezing of small solution droplets.

Page 16, line 18-19: This is an outdated view (and references that back this claim are not provided) - the current understanding is that a majority of cirrus clouds form via heterogeneous nucleation.

Fig. 8: Again, I do not think of MERRA as the most appropriate data set for validation of the simulated UT ice.

Page 16, line 21 (and throughout the manuscript): The standard terminology is "ice mass mixing ratio", not "ice mass fraction" which can be misleading.

Page 16, line 8 - 15: The description of how UT ice clouds form is extremely unclear. How is cloud cover determined? What probability distribution for supersaturation is used, and how does it relate to TKE. A lot of essential information is left out here.

Page 18, line 4: "each thick" is not correct English.

Page 18, line 8: What do you mean by "we are only considering sub-visible clouds"?

Fig. 10: Is the ice crystal number density calculated only when there is a cloud (i.e. an in-cloud average), or is this an average over both cloudy and cloud-free grid-boxes? The former quantity is certainly of most interest and more directly comparable to field measurements.

Page 18, line 10-11: Neglecting heterogeneous ice nucleation would lead to an overestimate of ice crystal number, because you are not able to represent the competition between heterogeneous and homogeneous nucleation that will in some cases lead to a suppression of homogeneous nucleation and therefore a reduction in ice crystal number density. In other words, that cannot explain the disagreement with

MERRA+MODIS seen in Fig. 9.

Page 26, line 7-9: How can you be confident about the radiative effect when the model consistently produces ice clouds that are optically too thin? This could bias especially the LW cooling effect of cloud thinning.

General comment: Friberg et al. (2015) seem to qualitatively support your findings based on analysis of cirrus cloud reflectance changes after volcanic eruptions, so that would be a good paper to cite.

---

## Referee Comment (RC2) · Anonymous Referee #2 · 18 Apr 2018

Review of "Upper tropospheric ice sensitivity to sulfate geoengineering" by Visioni et al

This manuscript analyzes geo-engineering simulations of sulfur injection in the ULAQ CCM. The paper is generally well written. It suffers from some minor grammar mistakes, but the scientific points are made. I am not sure how valid they are however. I think the methodology may be deeply flawed, since I am not certain that applying another model SSTs, from a model with no ice nucleation and poor upper tropospheric cirrus clouds, is a sufficiently useful method to look at perturbations. I think the resulting dynamical response could just be a model bias when the SSTs from another model is applied, and I fear that this would simply confuse the literature with another dubious single model study. This study needs some major revisions to address these points, and it may not actually be acceptable for ACP given the possible methodological flaws.

[Figure]

General Comments: 1. I know the authors' first language is not English, and English is not an easy or kind language for the article and plural mistakes they are making, but I would suggest an edit by a native English speaker.

2. As noted below, I am uncomfortable with some of the validation references. They should probably focus on papers, rather than other notes or presentations.

3. Most significantly: how does imposing SSTs from another model with an uncertain response tell us anything about the real atmosphere. You are just shocking one model with another, and you get a response. Why does the no feedback response matter, and how is it relevant? It is stated that some other models get a similar response, but I am not convinced. How would you even know if the model was self consistent?

P1,L2: incomplete sentence "The goal of the present study..."

P1,L15: Relative to the clear sky net...

P2,L10: How is this study different than previous work?

P4,L12: how much of these results are due to just individual model climatologies? Seems like the effects depend on how much homo v. Heterogenous ice a model has, and what a large scale model does to create and maintain cirrus. Why would your study be any more definitive?

P4, L14: The goal....

P4, L15: by including...

P4, L17: ULAQ model description and a reference are needed. Does the description appear later? It does. Add see below.

P4, L18: CCSM-CAM4 needs a description and the acronym spelled out. At least a reference for the simulations. Is there more model description later? Applying the cooling from another model seems problematic: presumably CCSM4 has some of the same feedbacks, operating in different ways?

P5,L1: I don't like that you have created a very arbitrary perturbation that changes vertical motion and transport in a coarse resolution GCM. The result is that I believe your perturbation is very model specific and artificial. I support attempting to understand processes in a model but this whole paper seems very dependent on a single model formulation. I'm not convinced you can or should separate all the affects this way.

P6,L20: I think you need to describe relevant features of the cloud and transport scheme of ULAQ, and the basic features of CCSM4 here. What ice nucleation mechanisms are included and how does the cloud scheme create cirrus clouds? What radiation scheme is used? How do the volcanic emissions evolve? For CCSM4: how do it's volcanic emissions evolve and how is that related.

P7, L1: the inconsistency here I think is problematic for the study. I'm not convinced you should look at this perturbation turning on and off surface temperature perturbations, and expect that the resulting impact on the model has any reference to reality since the system breaks any feedbacks that might modify the surface temperature.

P7,L14: Why should the surface temperature pattern be believed? CCSM4-CAM4 does not have interactive chemistry or a stratosphere. How are the emissions put in? Wouldn't this be different than ULAQ? Especially at high latitudes, impacts are dependent on a stratospheric circulation that I don't think CCSM4-CAM4 does correctly at all. Why not use WACCM4 Geoengineering experiments, which are at least based on a stratospheric model with interactive sulfur emissions.

P7, L25: So how is what you are doing different than Kuebbler et al 2012? Why is this novel or unique?

P7,L34: Updrafts responsible for....

P8,L3: So most of the vertical velocity is heavily and crudely parameterized by gravity waves and TKE. The TKE is probably linked strongly to the temp gradients. Does the model actually use this vertical velocity in advection? Or ice nucleation? Please

explain what is going on. It is not possible for the reader to understand whether the model formulation is realistic, though I am pretty convinced the perturbation (applying SSTs from another model) is NOT realistic for reasons described above.

P9,L5: Is a 3% change in a parameterized vertical velocity significant? Is 10% significantly different from 3%? From Figure 6, I don't think any of this is significant.

P9,L20: MODIS ice effective radius is not a reasonable product, especially for thin tropical cirrus, unless you have a validation paper that says otherwise.

P10,L3: The mention of what looks like a maximum updraft velocity here is an indication that the ULAQ ice nucleation needs to be better explained.

P12, L1: This section needs to go before all the results presented earlier.

P16, L20: It's not clear to me what fraction of ice formed in situ (T<238K) is from homogeneous and heterogeneous freezing. It would be useful to note the fraction homogenous (or heterogenous). This looks like it is in Figure 10c, but I don't think that is what I am interested in. What fraction of ice is heterogenously formed?

P21,L5: this is a decent summary that the changes are due to changes in vertical velocity and tropospheric temperatures. How model dependent do you think these quantities are?

P27, L5: Why the 5/8 scaling of the RF results?

P30, L25: How realistic is the decrease in updraft? Is it consistent with the overall circulation? I am concerned that fixing SSTs from another model will not yield a reasonable result, and it is likely to be a single model configuration, not even a general result. How can you convince me and other readers that the mechanism in ULAQ is reasonable, especially since it is imposed from another model and not-interactive, and from a model with no stratosphere.

---

## Author Comment (AC1) · 29 May 2018

Response to reviewer # 1

**Reviewer comments are in bold**. Author responses are in blue.

**This is an interesting and timely study of how geoengineering in the form of stratospheric aerosol injection (SAI) would impact ice clouds in the upper troposphere (UT). Few papers have been published on this poorly understood aspect of SAI, so in that sense the paper is a welcome contribution to the geoengineering discussion.**

We would like to thank the reviewer for his insightful comments and suggestions. We will try to address all of them below.

Overall comment:

**However, the paper is in its current form very unclear when it comes to the representation of central processes in the ULAQ model, poorly structured, and full of incorrect/poor grammar. It need a serious overhaul in all these respects. I further question the validity of the results presented, in light of the coarse resolution of the model, as well as its overly simplistic treatment of UT ice nucleation. I challenge the authors to justify why their results can be trusted despite these shortcomings. Below I've listed some additional major concerns I had about the paper, and thereafter some minor comments (typos, questions for clarification, etc.). I would like to see all of these concerns addressed before I will consider the paper suitable for publication in ACP.**

(a) A better, clearer and more complete presentation of the main processes governing ice particle formation in the ULAQ-CCM has been made in the revised manuscript.
(b) An improvement in the manuscript structure has been made, following also a specific recommendation from the reviewer (see below).
(c) Following the suggestion of both reviewers, an English technical editing of the manuscript has been done.
(d) The ULAQ-CCM version adopted in this study uses a T21 horizontal resolution, which may be (in general) defined as a rather coarse horizontal resolution. On the other hand, it has been demonstrated in many previous published works that this is a fully acceptable resolution for studies focusing on stratospheric dynamics and transport, as well as on strat-trop exchange (e.g., Pitari et al., 2016a; Pitari et al., 2016b; Visioni et al., 2018). It is obviously possible to use higher horizontal resolutions, but this is not a strict physical requirement. Many model inter-comparison campaigns prove this (see for example SPARC-CCMVal-2 or the ongoing SPARC-CCMI: Morgenstern et al., 2017; Morgenstern et al., 2018; see also other referenced papers in the manuscript, where the ULAQ model scores well compared to other models with higher horizontal resolution). This is even more true in the case of UT ice formation, which is largely driven by sub-grid vertical motions in global composition models; the latter may explicitly predict only the large-scale dynamics and need to parameterize mesoscale vertical motions, even with higher horizontal resolutions. On the other hand, we believe that the use of a high vertical resolution is necessary to properly catch the time-latitude-longitude varying altitude of the tropopause and then the upper limit for UT ice formation. A proper vertical resolution is indeed adopted in the ULAQ model (568 m in pressure altitude). In this way, the different aerosol behavior above and below the tropopause altitude is also well caught in the ULAQ-CCM.
(e) Cirrus ice formation in the ULAQ-CCM results from both homogeneous and heterogeneous freezing mechanisms and their competition. However, in our first draft of the manuscript we had decided to turn off the heterogeneous freezing mechanism, in order to focus on the SG

aerosol induced perturbation to ice formation from homogeneous freezing only. We acknowledge the specific point raised by the reviewer. The reduced updraft will affect ice formation from homogeneous freezing in a different way if ice particles may also form via heterogeneous freezing. This latter process, in fact, requires normally lower supersaturation conditions, both on mineral dust and black carbon particles. For this reason, we have performed again all our simulations with both mechanisms turned on, allowing their non-linear interaction. Results are substantially affected, with a resulting smaller indirect RF due to upper tropospheric ice changes induced by sulfate geoengineering. Looking at the revised results of our numerical experiments, we are really indebted with both reviewers for raising this specific scientific point, helping us in a more correct assessment of the ice perturbation due to SG aerosol and its indirect radiative forcing. In the manuscript, we now give a compact description of how ice particles are formed on both channels (HET-HOM), highlighting the major source of uncertainty in the parameterization of the heterogeneous freezing. A more robust scientific knowledge is present for the homogeneous freezing mechanism. In this case the numerical code adopted in the ULAQ model in the one well documented in the literature by Kärcher and Lohmann (2002) and Lohmann and Kärcher (2002), plus other subsequent studies, among which the most relevant for our present purposes is Kuebbeler et al. (2012).

Major comments:

**1) With respect to the paper structure, I found it strange that in Section 2 ("ULAQ-CCM and setup of numerical experiments") some model results in response to SG are presented (in Fig. 3-6), but not until sections 2.1 and 2.2. are descriptions of the model treatment of stratospheric aerosols and ice clouds described. I suggest moving the presentation and discussion of model results until AFTER you've described the model, and to only put content in the various sections that is consistent with the section titles.**

In the revised version of the manuscript we have shown all the model results after the sections where we describe the model, as suggested.

**2) There is no discussion of the effect of additional $SO_4$ available for homogeneous nucleation in the SG cases, as evident in Fig. 7a. Why is the effect of what appears to be a tripling of $SO_4$ particles that can nucleate ice seemingly negligible? Please explain?**

The increase of $SO_4$ in the upper troposphere due to the sulfate injection has a negligible effect on the rates of homogeneous freezing, as shown in Cirisan et al. (2013) (mainly because the number density of background $SO_4$ aerosols in the UT is already much greater than the number density of ice crystals). Furthermore, liquid supercooled sulfuric acid aerosols are inefficient IN for heterogeneous freezing; solid aerosol particles, mainly mineral dust and black carbon, may act as IN with different ice active fraction depending on aging processes and environmental conditions (e.g. Hendricks et al., 2011). For these reasons, changes in the UT population of sulfate aerosols are not expected to play a direct role in the changes in ice particle formation processes. It is actually the thermo-dynamical perturbation induced by lower stratospheric SG sulfate aerosols that may significantly impact the rate of ice formation via homogeneous freezing. This is indeed the central point of our study. In the revised manuscript, we have addressed this issue in a more in-depth way.

Minor comments:

**Abstract: "Goal of. . ." should be changed to "The goal of.."**

Changed.

**Abstract: Don't understand what "coupled to" means in this context.**

In previous experiments looking at sulfate geoengineering changes on UT ice, surface temperatures were kept fixed and only the LS warming was considered. In our case, in the experiment G4 both the surface cooling and the LS warming contribute to the modifications of the atmosphere dynamics.

**Page 2, line 4, End sentence after "documented.**

Done.

**Page 2, line 9: Add "optimal" before "magnitude and location".**

Added.

**Page 2, line 23: Add reference for the claim that homogeneous nucleation normally dominates cirrus formation. "Supersaturation ratio" should be "saturation ratio".**

Most of the literature considering geoengineering experiments (in particular, cirrus seeding) point out that most of the freezing is due to homogeneous processes, and that when including heterogeneous freezing processes, the differences are small. See for instance Gasparini et al. (2015), Gasparini et al. (2017), Storevlmo et al. (2014). We have added in the text these references and discuss better the relative weight of the two freezing mechanisms. Beyond geoengineering experiments, see also Kärcher and Lohmann (2002) for the specific point of homogeneous freezing normally dominating over the heterogeneous freezing of cirrus ice formation. Uncertainties in the latter case are discussed in Hendricks et al. (2011).

**Page 2, line 31: "anyway" is not suitable here. "However" could be an alternative.**

Corrected.

**Page 2, line 32: cloud optical properties are also important here.**

Added.

**Page 4, line 1: This statement is confusing: sulphuric acid droplets are not ice nuclei. Please clarify. Furthermore, this statement is not very interesting unless you explain WHY there was no effect on RF.**

As specified above in the response to the second major point, sulphuric acid liquid supercooled droplets are very inefficent ice nuclei. For the sake of increasing clarity, we have modified our sentence in the following way: "An upper tropospheric increase of sulfate aerosol number concentration is expected in SG conditions, due to gravitational sedimentation and large-scale transport of the particles below the tropopause from the LS. However, sulphuric acid liquid supercooled droplets are very inefficent ice nuclei (IN) for heterogeneous freezing. At the same time, the background number concentration of UT aerosols acting as nuclei for homogeneous frezing is already much higher with respect to the ice particle number density. For this reason, a negligible increase of the active IN population would be found in the UT, and the same would hold

true for the positive RF associated to a possible increase of ice particles from this effect, as Cirisan et al. (2013) conclude in their study."

**Page 4: Vertical velocity is important for cirrus formation not primarily because it transport water vapour to the UT, but because it controls the adiabatic cooling rate and thus supersaturation, for a given water vapour content.**

We have changed the sentence accordingly.

**Page 4, line 14-17: Catastrophic grammar.**

Grammar adjusted.

**Figure 1: This figure is confusing and not well explained. I don't find it particularly helpful at this stage of the paper, but it could be good as a final figure summarizing the findings in the paper.**

This figure has been moved to the final part of the manuscript, as well as Figure 2.

**Page 5, line 5: Ash dust is not the same.**

Corrected.

**Page 6, line 1: Sassen et al. (2008) is a paper on cirrus coverage seen by CALIPSO, so I don't see how that could possibly address UT ice changes.**

Yes, it was a mistake. The reference we were intending to put was Sassen et al. (1995), where possible changes in cirrus are discussed in relation to the Pinatubo eruption. We have now corrected this in the revised manuscript.

**Page 6/Table 1: The horizontal resolution is extremely coarse - how can we have confidence in changes driven by dynamics in this context?**

As explained above in the response to the overall comment of the reviewer, the ULAQ-CCM version adopted in this study uses a T21 horizontal resolution, which may be (in general) defined as a rather coarse horizontal resolution. On the other hand, it has been demonstrated in many previous published works that this is a fully acceptable resolution for studies focusing on stratospheric dynamics and transport, as well as on strat-trop exchange. Many model inter-comparison campaigns prove this (see for example SPARC-CCMVal-2 or the ongoing SPARC-CCMI). In addition, UT ice formation is largely driven by sub-grid vertical transport processes in global composition models; the latter may explicitly predict only the large-scale dynamics and need to parameterize mesoscale vertical motions, even with higher horizontal resolutions.

**Page 7, line 4: Clumsy and confusing statement. Suggest writing: ...a negative anomaly in the Arctic region that is approximately 1 K larger than that of high southern latitudes.**

We have rephrased it as the reviewer suggested.

**Page 7, line 7: What do you mean by "increasing atmospheric stabilisation"? Do you mean "increasing atmospheric stability?**

Yes, we have corrected this.

**Page 7, line 16-18: The Antarctic warming is not statistically significant, so I don't see the point in discussing it.**

Although the reviewer remark is correct, we would like to keep this short discussion as a justification for the large variability of SST changes at high latitudes. A slight modification has been made by writing: "Although not statistically significant, the SG induced warming…"

**Page 8, line 2: Is the vertical velocity change mainly caused by changes to TKE, or also due to large-scale (resolved) velocity changes. If TKE is very important here, I would like to see vertical profiles of TKE for both simulations.**

In the revised manuscript, we have explicitly included the Lohmann and Kärcher (2002) formulation for the vertical velocity (Eq. 5). In a first approximation, w in the UT is close to $TKE^{0.5}$ and the vertical velocity perturbation is dominated by changes of this latter term, so that we believe that inclusion of a new figure in the revised manuscript does not add much. We attach here the vertical profiles of w in G4 and Base experiments (Fig. R1_1), as well as the vertical profile of w changes [G4-Base], comparing the results with and without the large-scale contribution to w. It is clear that TKE changes greatly control the SG perturbation of UT updraft. The TKE vertical profile asked by the reviewer is implicit in panel (b) of the Fig. R1_1, due the $w_{TKE}$ formulation.

[Figure]

Fig. R1_1. Average upper tropospheric tropical profiles of the vertical velocity w (cm/s) in G4 and Base experiments (years 2030-39). (a) Total vertical velocity calculated as $w_{TOT}=w_{TKE}+w_{LS}$ (where $w_{TKE}$ indicates the mesoscale component calculated as a function of TKE and $w_{LS}$ indicates the large-scale model-resolved component). (b) Vertical velocity component $w_{TKE}$ alone, calculated as $w_{TKE}=0.7\times(TKE)^{1/2}$ (see Lohmann and Kärcher, 2002; see also Eq. 5 in the revised manuscript). As expected, $w_{TKE}$ dominates in $w_{TOT}$ (c) Vertical velocity changes G4-Base for $w_{TOT}$ (solid line) and $w_{TKE}$ only (dashed line) (of the order of 5% in the tropical UT). Very little changes are produced in the large-scale component under SG conditions.

**Page 9, lines 19-20: Discuss here the uncertainty associated with cloud ice in MERRA, which uses highly uncertain cloud parameterisations and incorporates very few ice cloud**

**observations in its reanalysis. It would be better to use CALIPSO/CloudSat retrievals of ice cloud properties.**

In the revised manuscript, we have added some discussion on the uncertainties of the datasets we have used for comparison with our results. We have decided to use a reanalyses dataset such as MERRA-2 also considering the large uncertainties in the satellite retrieval datasets (see for instance Zhang et al., 2010; Duncan and Eriksson, 2018) and their availability.

**Page 10, line 3: Given how central UT vertical velocities are to this paper, you need to be clearer about how the calculation of vertical velocity is done, i.e. include equation for vertical velocity as a function of TKE, and clearly state if you put any upper/lower bounds in it.**

In the revised manuscript, we have explicitly included the Lohmann and Kärcher (2002) formulation for the vertical velocity (Eq. 5) (see also Fig. R1_1 above). No imposed bounds to w are considered and its time-spatial variability is clearly shown in Figure 6 of the original manuscript.

**Page 10, line 4: What justifies the assumption that cirrus clouds form only via homogeneous nucleation? That seems to be in stark contrast to papers that report that cirrus clouds appear to form mainly through heterogeneous nucleation (e.g., Cziczo et al., 2013).**

Please refer to our reply above, regarding the homogeneous - heterogeneous freezing mechanisms, in response to the reviewer overall comment. In the revised manuscript, we discuss the major source of uncertainties and add a caveat regarding the presence of different opinions available in the literature, like the ones the reviewer suggested.
As specified above, we have taken the reviewer criticism under serious consideration and decided to redo our numerical simulations with both freezing mechanisms turned on, allowing their non-linear interaction. Results are substantially affected, with a resulting smaller indirect RF due to upper tropospheric ice changes induced by sulfate geoengineering. Looking at the revised results of our numerical experiments, we are really indebted with both reviewers for raising this specific scientific point, helping us in a more correct assessment of the ice perturbation due to SG aerosol and its indirect radiative forcing.

**Page 13, line 15: Remove "from".**

Removed.

**Page 16, line 13-14: This is inaccurate - homogeneous nucleation sets in at approximately 238K, but NOT through "water vapour freezing", but rather through the spontaneous freezing of small solution droplets.**

Corrected accordingly.

**Page 16, line 18-19: This is an outdated view (and references that back this claim are not provided) - the current understanding is that a majority of cirrus clouds form via heterogeneous nucleation.**

Please see above. Heterogeneous nucleation would dominate only if the locally available IN (mostly BC and mineral dust) would have a high ice active fraction (>~10%). These values, however, although being measured in laboratory studies for mineral dust close to the homogeneous freezing threshold (Field et al., 2006; Möhler et al., 2006; Welti et al., 2009), are most probably

highly overestimated in the real atmosphere, due to rapid aging of dust particles (as well as BC) through sulfate coating (Hendricks et al., 2011). We acknowledge (and added in the revised manuscript) the counterpoint that studies such as Cziczo et al. (2013) show that the heterogeneous freezing may dominate over the homogeneous, in the formation of UT ice particles. However, we believe there is plenty of literature showing through both modeling and in-chamber experiments the huge uncertainties relative to our understanding of UT ice formation through heterogeneous freezing, in particular regarding the available aerosol population that is actually able to form ice in the upper troposphere (ice active fraction).

For instance, regarding black carbon, laboratory measurements demonstrate that the ice active fraction (f) ranges between 0.1% and 1% (Koehler et al., 2009), which means that only a very small fraction of the available black carbon particles in the UT can act as IN. Considering the rapid BC aging in the real atmosphere (due to sulfate coating), f~0.1% may be probably considered as un upper limit for the ice active fraction, although a clear picture has not yet emerged for the factors which actually control f for a given type of atmospheric IN (Hendricks et al., 2011), thus producing a significant level of uncertainty in the present knowledge of UT ice formation via heterogeneous freezing. For mineral dust the uncertainty is even higher, with f ranging between 0.1% and 10%, although it might be even lower (Minikin et al., 2003; Cziczo et al., 2009).

Those measurements are the only one that, as modelers, we can take into account when considering which fraction is to be used in our simulations (in our experiments we chose f=0.25% for BC and 1% for mineral dust, following the best recommendations of Hendricks et al., 2011, see Eq. 1 in the revised manuscript).

**Fig. 8: Again, I do not think of MERRA as the most appropriate data set for validation of the simulated UT ice.**

See response above.

**Page 16, line 21 (and throughout the manuscript): The standard terminology is "ice mass mixing ratio", not "ice mass fraction" which can be misleading.**

Following the suggestion, we have changed it everywhere in the manuscript.

**Page 16, line 8 - 15: The description of how UT ice clouds form is extremely unclear. How is cloud cover determined? What probability distribution for supersaturation is used, and how does it relate to TKE. A lot of essential information is left out here.**

We disagree on this point. We have clearly stated our simplified probabilistic approach adopted for supersaturation, with a normal (Gaussian) distribution for the UT relative humidity: "For the ice supersaturation ratio, we adopt a simplified probabilistic approach, starting from the knowledge of climatological frequencies of the UT relative humidity ($RH_{ICE}$), from which a mean value and a standard deviation can be calculated, assuming a normal distribution". We are aware that this represents an important model simplification, and in fact we started our discussion with the above clear statement.

**Page 18, line 4: "each thick" is not correct English.**

Corrected.

**Page 18, line 8: What do you mean by "we are only considering sub-visible clouds"?**

The sentence has been modified as follows: "This should not surprise, in principle, due to the fact that vertical velocities calculated as a function of TKE do not normally exceed 30 cm/s, so that events leading to thick cirrus formation are not considered."

**Fig. 10: Is the ice crystal number density calculated only when there is a cloud (i.e. an in-cloud average), or is this an average over both cloudy and cloud-free grid-boxes? The former quantity is certainly of most interest and more directly comparable to field measurements.**

We always refer to averages weighted with the probability to have cirrus formation ($\sim P_{HOM}$). The reviewer is right in saying that an in-cloud average would be more directly comparable to field measurements. That's why we used our $P_{HOM}$ to make this type of comparison in a meaningful way. In the original manuscript, we wrote: "Using these $P_{HOM}$ values, it is possible to scale a $n_i$ value measured in the mid-latitude airborne campaign of Ström et al. (1997) during a young cirrus formation, in order to derive an average climatological value to be considered consistent with our modeling approach. They measured a mid-latitude ice concentration value $n$=0.3 cm$^{-3}$ in a young cirrus cloud at T=220 K and p=320 hPa.  If we scale this result with our corresponding $P_{HOM}$=12±3%, a "climatological-mean" value $n$=0.025±0.005 cm$^{-3}$ is obtained, close to our model predicted value of 0.031± 0.008 cm$^{-3}$."

**Page 18, line 10-11: Neglecting heterogeneous ice nucleation would lead to an overestimate of ice crystal number, because you are not able to represent the competition between heterogeneous and homogeneous nucleation that will in some cases lead to a suppression of homogeneous nucleation and therefore a reduction in ice crystal number density. In other words, that cannot explain the disagreement with MERRA+MODIS seen in Fig. 9.**

The reviewer is perfectly right. As explained above in detail, our new results confirm this. Again we thank both reviewers for raising this point and providing us a strong scientific argument to redo our numerical experiments. The following sentence has been deleted: "In addition, ice formation from heterogeneous freezing on active IN, as mineral dust particles for example, is not taken into account in our modelling approach."

**Page 26, line 7-9: How can you be confident about the radiative effect when the model consistently produces ice clouds that are optically too thin? This could bias especially the LW cooling effect of cloud thinning.**

Our confidence comes from comparing our results to previous findings (as in Kuebbeler et al., 2012, Gasparini et al., 2017). In addition, the reviewer point is rather unclear, in the sense that the ice OD for G4, G4K and Base simulations is not small in absolute values. We may then calculate the ice radiative effects both on SW and LW, once appropriate Mie scattering parameters have been derived, using a correct wavelength-dependent refractive index (Warren 1984; Warren and Brandt, 2008; Curtis et al., 2005) and the calculated particle size distribution. Results of our radiative transfer code have been successfully compared with those of Schumann et al. (2012) under similar conditions. We have added in the paper the appropriate references.

**General comment: Friberg et al. (2015) seem to qualitatively support your findings based on analysis of cirrus cloud reflectance changes after volcanic eruptions, so that would be a good paper to cite.**

We have read the paper suggested by the reviewer and had the occasion to speak with the lead author. In the revised manuscript, we now briefly discuss their conclusions and have added the appropriate reference.

References:

Cirisan, A., Spichtinger, P., Luo, B. P., Weisenstein, D. K., Wernli, H., Lohmann, U., and Peter, T.: Microphysical and radiative changes in cirrus clouds by geoengineering the stratosphere, Journal of Geophysical Research: Atmospheres, 118, 4533–4548, doi:10.1002/jgrd.50388, http://dx.doi.org/10.1002/jgrd.50388, 2013.

Curtis, D. B., Rajaram, B., Toon, O. B., and Tolbert, M. A.: Measurement of the temperature-dependent optical constants of water ice in the 15–200 μm range, Appl. Opt., 44, 4102–4118, doi:10.1364/AO.44.004102, http://ao.osa.org/abstract.cfm?URI=ao-44-19-4102, 2005.

Cziczo, D. J., Froyd, K. D., Gallavardin, S. J., Moehler, O., Benz, S., Saathoff, H., and Murphy, D. M.: Deactivation of ice nuclei due to atmospherically relevant surface coatings, Environmental Research Letters, 4, 044 013, http://stacks.iop.org/1748-9326/4/i=4/a=044013, 2009.

Cziczo, D. J., Froyd, K. D., Hoose, C., Jensen, E. J., Diao, M., Zondlo, M. A., Smith, J. B., Twohy, C. H., and Murphy, D. M.: Clarifying the Dominant Sources and Mechanisms of Cirrus Cloud Formation, Science, 340, 1320–1324, doi:10.1126/science.1234145, http://science.sciencemag.org/content/340/6138/1320, 2013.

Duncan, D. I. and Eriksson, P.: An update on global atmospheric ice estimates from satellite observations and reanalyses, Atmos. Chem. Phys. Discuss., https://doi.org/10.5194/acp-2018-275, in review, 2018.

Field, P. R., Möhler, O., Connolly, P., Krämer, M., Cotton, R., Heymsfield, A. J., Saathoff, H., and Schnaiter, M.: Some ice nucleation characteristics of Asian and Saharan desert dust, Atmos. Chem. Phys., 6, 2991-3006, https://doi.org/10.5194/acp-6-2991-2006, 2006.

Friberg J., B. G. Martinsson, M. K. Sporre, S. M. Andersson, C. A. M. Brenninkmeijer, M. Hermann, P. F. J. van Velthoven, and A. Zahn (2015), Influence of volcanic eruptions on midlatitude upper tropospheric aerosol and consequences for cirrus clouds, Earth and Space Science, 2, 285–300, doi: 10.1002/2015EA000110.

Gasparini, B. and Lohmann, U.: Why cirrus cloud seeding cannot substantially cool the planet, Journal of Geophysical Research: Atmospheres, 121, 4877–4893, doi:10.1002/2015JD024666, https://agupubs.onlinelibrary.wiley.com/doi/abs/10.1002/2015JD024666, 2016.

Gasparini, B., Munch, S., Poncet, L., Feldmann, M., and Lohmann, U.: Is increasing ice crystal sedimentation velocity in geoengineering simulations a good proxy for cirrus cloud seeding?, Atmospheric Chemistry and Physics, 17, 4871–4885, doi:10.5194/acp-17-4871-2017, https://www.atmos-chem-phys.net/17/4871/2017/, 2017.

Hendricks, J., Kärcher, B., and Lohmann, U.: Effects of ice nuclei on cirrus clouds in a global climate model, Journal of Geophysical Research: Atmospheres, 116, doi:10.1029/2010JD015302, http://dx.doi.org/10.1029/2010JD015302, d18206, 2011.

Kärcher, B. and Lohmann, U.: A parameterization of cirrus cloud formation: Homogeneous freezing of supercooled aerosols, Journal of Geophysical Research: Atmospheres, 107, doi:10.1029/2001JD000470, http://dx.doi.org/10.1029/2001JD000470, 2002.

Koehler, K. A., P. J. DeMott, S. M. Kreidenweis, O. B. Popovicheva, M. D. Petters, C. M. Carrico, E. D. Kireeva, T. D. Khokhlova, and N. K. Shonija (2009), Cloud condensation nuclei and ice nucleation activity of hydrophobic and hydrophilic soot particles, Phys.Chem.Chem.Phys., 11, 7906–7920.

Kuebbeler, M., Lohmann, U., and Feichter, J.: Effects of stratospheric sulfate aerosol geo-engineering on cirrus clouds, Geophysical Research Letters, 39, doi:10.1029/2012GL053797, http://dx.doi.org/10.1029/2012GL053797, l23803, 2012.

Lohmann, U. and Kärcher, B.: First interactive simulations of cirrus clouds formed by homogeneous freezing in the ECHAM general circulation model, Journal of Geophysical Research: Atmospheres, 107, AAC 8–1–AAC 8–13, doi:10.1029/2001JD000767, http://dx.doi.org/10.1029/2001JD000767, 2002.

Minikin, A., A. Petzold, J. Ström, R. Krejci, M. Seifert, P. van Velthoven, H. Schlager, and U. Schumann (2003), Aircraft observations of the upper tropospheric fine particle aerosol in the Northern and Southern Hemispheres at midlatitudes, *Geophys. Res. Lett.*, 30, 1503, doi: 10.1029/2002GL016458, 10.

Möhler, O., Field, P. R., Connolly, P., Benz, S., Saathoff, H., Schnaiter, M., Wagner, R., Cotton, R., Krämer, M., Mangold, A., and Heymsfield, A. J.: Efficiency of the deposition mode ice nucleation on mineral dust particles, Atmos. Chem. Phys., 6, 3007-3021, https://doi.org/10.5194/acp-6-3007-2006, 2006.

Morgenstern, O., Hegglin, M. I., Rozanov, E., O'Connor, F. M., Abraham, N. L., Akiyoshi, H., Archibald, A. T., Bekki, S., Butchart, N., Chipperfield, M. P., Deushi, M., Dhomse, S. S., Garcia, R. R., Hardiman, S. C., Horowitz, L. W., Jockel, P., Josse, B., Kinnison, D., Lin, M., Mancini, E., Manyin, M. E., Marchand, M., Marecal, V., Michou, M., Oman, L. D., Pitari, G., Plummer, D. A., Revell, L. E., Saint-Martin, D., Schofield, R., Stenke, A., Stone, K., Sudo, K., Tanaka, T. Y., Tilmes, S., Yamashita, Y., Yoshida, K., and Zeng, G.: Review of the global models used within phase 1 of the Chemistry–Climate Model Initiative (CCMI), Geoscientific Model Development, 10, 639–671, doi:10.5194/gmd-10-639-2017, https://www.geosci-model-dev.net/10/639/2017/, 2017.

Morgenstern, O., Stone, K. A., Schofield, R., Akiyoshi, H., Yamashita, Y., Kinnison, D. E., Garcia, R. R., Sudo, K., Plummer, D. A., Scinocca, J., Oman, L. D., Manyin, M. E., Zeng, G., Rozanov, E., Stenke, A., Revell, L. E., Pitari, G., Mancini, E., Di Genova, G., Visioni, D., Dhomse, S. S., and Chipperfield, M. P.: Ozone sensitivity to varying greenhouse gases and ozone-depleting substances in CCMI-1 simulations, Atmos. Chem. Phys., 18, 1091-1114, https://doi.org/10.5194/acp-18-1091-2018, 2018.

Sassen, K., D. O. Starr, G. G. Mace, M. R. Poellot, S. H. Melfi, W. L. Eberhard, J. D. Spinhirne, E. W. Eloranta, D. E. Hagen and J. Hallett, The 5-6 December 1991 FIRE II jet stream cirrus case study: Possible influences of volcanic aerosols. d. Atmos. Sci., 52, 97-123, 1995.

Schumann, U., Mayer, B., Graf, K., and Mannstein, H.: A Parametric Radiative Forcing Model for Contrail Cirrus, Journal of Applied Meteorology and Climatology, 51, 1391–1406, doi:10.1175/JAMC-D-11-0242.1, https://doi.org/10.1175/JAMC-D-11-0242.1, 2012.

Storelvmo, T. and Herger, N.: Cirrus cloud susceptibility to the injection of ice nuclei in the upper troposphere, Journal of Geo- physical Research: Atmospheres, 119, 2375–2389, doi:10.1002/2013JD020816, https://agupubs.onlinelibrary.wiley.com/doi/abs/10.1002/2013JD020816, 2014.

Strom, J., Strauss, B., Anderson, T., Schrader, F., Heintzenberg, J., and Wendling, P.: In Situ Observations of the Microphysical Properties of Young Cirrus Clouds, Journal of the Atmospheric Sciences, 54, 2542–2553, doi:10.1175/1520-0469(1997)054<2542:ISOOTM>2.0.CO;2, http://dx.doi.org/10.1175/1520-0469(1997)054<2542:ISOOTM>2.0.CO;2, 1997.

Warren, S. G.: Optical constants of ice from the ultraviolet to the microwave, Appl. Opt., 23, 1206–1225, doi:10.1364/AO.23.001206, http://ao.osa.org/abstract.cfm?URI=ao-23-8-1206, 1984.

Warren, S. G. and Brandt, R. E.: Optical constants of ice from the ultraviolet to the microwave: A revised compilation, Journal of Geophysical Research: Atmos., 113, doi:10.1029/2007JD009744, https://agupubs.onlinelibrary.wiley.com/doi/abs/10.1029/2007JD009744, 2008.

Welti, A., Lüönd, F., Stetzer, O., and Lohmann, U.: Influence of particle size on the ice nucleating ability of mineral dusts, Atmos. Chem. Phys., 9, 6705-6715, https://doi.org/10.5194/acp-9-6705-2009, 2009.

Zhang, Z., S. Platnick, P. Yang, A. K. Heidinger, and J. M. Comstock, Effects of ice particle size vertical inhomogeneity on the passive remote sensing of ice clouds, J. Geophys. Res., 115, D17203, doi:10.1029/2010JD013835, 2010.

---

## Author Comment (AC2) · 29 May 2018

Response to reviewer # 2

**Reviewer comments are in bold**. Author responses are in blue.

**This manuscript analyzes geo-engineering simulations of sulfur injection in the ULAQ CCM. The paper is generally well written. It suffers from some minor grammar mistakes, but the scientific points are made. I am not sure how valid they are however.**

We thank the reviewer for his in-depth review and for his comments. We will try to respond to all the points raised, and to show that our work is scientifically robust.

Overall comment:

**I think the methodology may be deeply flawed, since I am not certain that applying another model SSTs, from a model with no ice nucleation and poor upper tropospheric cirrus clouds, is a sufficiently useful method to look at perturbations. I think the resulting dynamical response could just be a model bias when the SSTs from another model is applied, and I fear that this would simply confuse the literature with another dubious single model study. This study needs some major revisions to address these points, and it may not actually be acceptable for ACP given the possible methodological flaws.**

We disagree on this point. At the same time, it is rather clear from the first sentence and from other remarks below, that we were not able to make the scientific structure of our work sufficiently clear to the reader. We have tried to do it much better in the revised version and in the present reply to the reviewer.

(a) Our study takes inspiration from a previous one (Kuebbeler et al, 2012), where SSTs were kept unchanged in the sulfate geoengineering perturbed G4 case (5 Tg-SO$_2$/yr injection in the tropical lower stratosphere), with respect to the control simulation without geoengineering aerosols. In that case, dynamical changes produced by the lower stratospheric aerosol heating become drivers of a significant indirect effect of sulfate geoengineering (SG) on ice particle formation in the upper troposphere via homogeneous freezing. The increasing atmospheric static stability, due to the lower stratospheric aerosol-induced warming, produces a reduction in synoptic scale vertical motions with a resulting decrease in ice particle formation.

(b) An important question (raised in the same paper of Kuebbeler et al., 2012) may obviously be to what extent the surface cooling produced by the increased planetary albedo in G4 conditions (i.e., the DIRECT effect of geoengineering aerosol, for which SG is actually designed) may contribute to dynamical changes together with the lower stratospheric aerosol heating.

(c) In order to tackle this last scientific question, different approaches are possible. The ideal procedure would be to use an ocean-atmosphere coupled model, with chemistry-aerosol-ice clouds on-line, fully interactive with radiation and dynamics and with a vertical extension covering both troposphere and stratosphere. Possibly in a multi-model configuration, to assess inter-model differences (i.e. MIP approach). At the moment, however, this is rather difficult to achieve, since all the above requirements are not easy to be found and adapted to a SG configuration. This would be what the reviewer calls (below) a more "definitive" study.

(d) An alternative approach would be to use an atmospheric climate-chemistry coupled model (CCM) with sulfur chemistry and sulfate aerosol microphysics on-line (Pitari et al., 2014; Visioni et al., 2018), in-depth process evaluation (Visioni et al., 2017b), ice cloud formation scheme and its evaluation (as attempted in the present work). This is exactly what we have

done with the ULAQ-CCM. On the other hand, it is well known that CCMs need an external specification of SSTs (due to their intrinsic formulation). This has been extensively made in previous international model campaigns, on-going since 2006 (i.e., SPARC-CCMVal-1, SPARC-CCMVal-2, SPAR-CCMI-1). What is needed for this purpose is the output of an atmosphere-ocean coupled model run with and without SG, under a given RCP scenario. The desired SG effect on SST is indeed the DIRECT aerosol effect (i.e., surface cooling due to the increasing planetary albedo), and CCSM-CAM4 does it well. Indirect effects on both chemistry and upper tropospheric ice are secondary effects not needed at this stage, contrary to what the reviewer claims. In fact, having an external SST change sensitive also to SG aerosol INDIRECT perturbations (chemistry and ice), would actually create an inconsistency in the nudging procedure. We rely of the fact that the SG aerosol radiative perturbation is the dominant one (see Visioni et al., 2017a) and we use the resulting changes on SST predicted by CCSM-CAM4 as the first-approximation dynamical driver for a CCM designed for the same SG perturbation (i.e., 8 Tg-SO$_2$/yr), but also including indirect effects on ice clouds (present work) and chemistry (see Visioni et al., 2017b). We will try to address this point in the revised version of the manuscript.

(e) We strongly believe that our "single model work", which is scientifically respectable as the "single model work" of Kuebbeler et al (2012), may be considered a good step forward, in the sense that the use of an externally specified SST sensitive to the direct SG aerosol radiative perturbation makes the CCM ice response more realistic than keeping SSTs fixed with respect to the baseline reference case.

(f) At the same time, we hope that in future a "more definitive" study will be conducted with on-line predicted SSTs, function of direct and indirect radiative changes produced by stratospheric SG aerosols.

General Comments:

**1. I know the authors' first language is not English, and English is not an easy or kind language for the article and plural mistakes they are making, but I would suggest an edit by a native English speaker.**

Following the suggestion of both reviewers, an English technical editing of the manuscript has been done.

**2. As noted below, I am uncomfortable with some of the validation references. They should probably focus on papers, rather than other notes or presentations.**

We are not sure what the reviewer is referring to. We have however reviewed the references used in the paper regarding the validation data we used. The only technical report in the original manuscript is Chou et al. (2001) regarding the longwave radiative transfer code; a journal paper was also cited for the shortwave radiative transfer code (Randles et al., 2013). Bosilovich et al. (2017) describes the MERRA-2 data and is a peer reviewed paper. If the reviewer asks for additional references regarding observational data, we have added two more, i.e. Gelaro et al. (2017) and Duncan and Eriksson (2018).

**3. Most significantly: how does imposing SSTs from another model with an uncertain response tell us anything about the real atmosphere. You are just shocking one model with another, and you get a response. Why does the no feedback response matter, and how is it relevant? It is stated that some other models get a similar response, but I am not convinced. How would you even know if the model was self-consistent?**

As explained in the response to the overall general comment, this is the normal way CCMs are used for baseline and sensitivity experiments (to RCP scenarios, solar fluxes, short lived species ground fluxes and many other components of the climate systems, along with their connection with chemistry) [see above points (d-e)]. Our results are consistent with those of Kuebbeler et al. (2012) for the upper tropospheric ice sensitivity to stratospheric SG aerosols, in the sense that the lower stratospheric aerosol longwave and solar heating rates are the major driver for circulation changes, but we go a step forward considering also the potential significance of the tropospheric cooling induced by the stratospheric aerosols [see above points (d-f)].

The reviewer often uses the argument of "self-consistency", but this does not apply in CCM experiments, because SSTs are an input parameter in this type of model. I think we have clearly explained in response to the overall comment, that we use SSTs from the CCSM-CAM4 ocean-atmosphere model for having a reliable input on "baseline" surface temperatures in a future RCP scenario and a "reliable" input for the SG aerosol perturbation to these temperatures. The latter is the "dominant direct" climate effect of SG. Indirect effects (i.e. chemistry and upper tropospheric ice) are treated consistently in the UAQ-CCM formulation, assuming SST changes produced by the SG stratospheric aerosols as a good first approximation. The CCSM-CAM4 SG stratospheric aerosol distribution used in the geoengineering simulation has been detailed in Tilmes et al. (2015).

**Incomplete sentence "The goal of the present study..."**

Corrected.

**Relative to the clear sky net...**

Changed.

**How is this study different than previous work?**

The only other study regarding the thermo-dynamical effects of sulfate geoengineering on cirrus cloud was that of Kuebbeler et al. (2012). In their case, however, sea surface temperatures where kept fixed. In our study, as the authors of the aforementioned paper asked in their conclusions, we try to analyze the difference between a sulfate injection with (G4) and without (G4K) the changes in sea surface temperatures due to the injected sulfate. We believe that, by showing the differences between G4 and G4K results in our model, we can gain further knowledge regarding this particular side effect.

**How much of these results are due to just individual model climatologies? Seems like the effects depend on how much homo v. heterogeneous ice a model has, and what a large-scale model does to create and maintain cirrus. Why would your study be any more definitive?**

We don't believe our study to be definitive in any way. We show, when comparing our G4K results with those from Kuebbeler et al. (2012), that the results from the two models are comparable in that scenario, and that further differences appear when considering changes in sea surface temperatures produced by the SG aerosol perturbation. We believe that by analyzing the differences caused by only that factor (SST changes due to the SG aerosol direct effect) we can constrain one of the possible factors that might influence the dynamical response to sulfate geoengineering. This approach was also used in a previous study related to methane changes (Visioni et al., 2017b), where we compared our results (in simulations with and without changing SSTs) against results from GEOSCCM.

**The goal....**

Corrected.

**by including...**

Changed.

**ULAQ model description and a reference are needed. Does the description appear later? It does. Add see below.**

Following also the precious suggestions of reviewer 1, in the revised manuscript we have modified the structure of the paper in order to have the model description before anything else.

**CCSM-CAM4 needs a description and the acronym spelled out. At least a reference for the simulations. Is there more model description later? Applying the cooling from another model seems problematic: presumably CCSM4 has some of the same feedbacks, operating in different ways?**

In the revised manuscript we have described CCSM-CAM4 and tried to better explain our modeling approach in the use of this model SSTs. Again, please refer to our response to the overall comment.

**I don't like that you have created a very arbitrary perturbation that changes vertical motion and transport in a coarse resolution GCM. The result is that I believe your perturbation is very model specific and artificial. I support attempting to understand processes in a model but this whole paper seems very dependent on a single model formulation. I'm not convinced you can or should separate all the affects this way.**

As previously explained, the methodology of using externally provided SSTs as input parameter in CCM experiments is intrinsic in the CCM formulation itself. This has been done in all CCMVal-1, CCMVal-2 and CCMI-1 experiments, and more recently for the ISA-MIP Project (Timmreck et al., 2018; https://www.geosci-model-dev-discuss.net/gmd-2017-308/). As for using perturbed SSTs in case of a geoengineering scenario, we can point out to previous works where our dynamical perturbations have been compared to other models (Pitari et al., 2014; Visioni et al., 2017b).
The resulting dynamical changes are not arbitrary, but consistent with the SG aerosol dynamical drivers, i.e. perturbation of lower stratospheric heating rates and SSTs. A clear discussion is made in the manuscript on how the resulting changes in vertical motion are produced and how they are sensitive to these aerosol drivers. The results are obviously valid in the limitation of "a single model formulation" (as in the case of Kuebbeler et al., 2012, by the way), but may certainly represent a step forward and could be a valuable reference point in the literature for future multi-model experiments, possibly with ocean-atmosphere coupled models.

**I think you need to describe relevant features of the cloud and transport scheme of ULAQ, and the basic features of CCSM4 here. What ice nucleation mechanisms are included and how does the cloud scheme create cirrus clouds? What radiation scheme is used? How do the volcanic emissions evolve? For CCSM4: how do its volcanic emissions evolve and how is that related.**

As already specified above, a new paragraph on the CCSM-CAM4 model has been included in the revised manuscript. A full description of the ULAQ-CCM is available in the Morgenstern et al. (2017) paper, which summarizes the major features of all global-scale model participating in the SPARC-CCMI model initiative. Details on the radiation scheme are also available in this latter

paper, as well as in Pitari et al. (2014). Evolution of volcanic clouds in the ULAQ-CCM has been fully discussed in Pitari et al. (2016a) and Pitari et al. (2016b). CCSM-CAM4 is described in Tilmes et al. (2016).

A full section in the original manuscript is devoted to explaining the cirrus cloud formation in the ULAQ-CCM, via homogeneous freezing. An additional paragraph on the ice formation via heterogeneous freezing is now included in the revised manuscript.

Regarding the volcanic emissions, the simulations are in the future under a RCP4.5 scenario, so volcanic emissions are not considered.

**The inconsistency here I think is problematic for the study. I'm not convinced you should look at this perturbation turning on and off surface temperature perturbations, and expect that the resulting impact on the model has any reference to reality since the system breaks any feedbacks that might modify the surface temperature.**

Most of the available works on sulfate geoengineering have been performed using models with prescribed SSTs (as an example, Kuebbeler et al., 2012; Niemeier and Timmreck, 2015; Niemeier and Schmidt, 2017). We believe that showing what happens when turning on and off the surface temperature perturbation might be a valuable way to understand some of the feedbacks.

In addition, we would like to remind that the primary perturbation driving dynamical changes in the atmosphere is the lower stratospheric heating due to SG aerosols (see Kuebbeler et al., 2012). We show that SST changes end up increasing the atmospheric stabilization, which is primarily produced by the lower stratospheric aerosol warming.

**Why should the surface temperature pattern be believed? CCSM-CAM4 does not have interactive chemistry or a stratosphere. How are the emissions put in? Wouldn't this be different than ULAQ? Especially at high latitudes, impacts are dependent on a stratospheric circulation that I don't think CCSM-CAM4 does correctly at all.**

We believe that the surface temperature predicted by CCSM-CAM4 in case of a sulfate geoengineering injection can be used as long as it is clear that it is a first order approximation, because it responds to the direct SG effects (i.e. aerosol increased planetary albedo), allowing the ULAQ-CCM a more realistic study of the atmospheric response to the indirect effects (chemistry, ice) with respect to a case in which SSTs were kept fixed at the RCP4.5 reference values (G4K). This is clarified in the revised manuscript. In addition, in order to be more specific regarding CCSM-CAM4, we have asked the scientist responsible for those SG G4 simulations (Simone Tilmes) to give her contribution to the manuscript by further explaining some of the aspects of the model (as was done for Visioni et al., 2017b). Regarding the modeling of the stratosphere in CCSM-CAM4, we will also reference Lamarque et al. (2012), Neale et al. (2013) and Tilmes et al. (2016) in the revised manuscript.

**Why not use WACCM4 Geoengineering experiments, which are at least based on a stratospheric model with interactive sulfur emissions.**

The available WACCM4 Geoengineering simulations have not been performed using a fixed injection from 2020 and 2070 as prescribed by the GeoMIP protocol.

**So how is what you are doing different than Kuebbler et al. (2012)? Why is this novel or unique?**

As we explained before, we believe that including the two direct effects of SG aerosols in the CCM, as primary drivers for dynamical changes, it allows a more complete assessment of the SG impact

on upper tropospheric ice formation, with respect to previous study by Kuebbeler et al. (2012) where SSTs were kept fixed at the reference RCP values.

**Updrafts responsible for....**

Corrected.

**So most of the vertical velocity is heavily and crudely parameterized by gravity waves and TKE. The TKE is probably linked strongly to the temp gradients. Does the model actually use this vertical velocity in advection? Or ice nucleation? Please explain what is going on. It is not possible for the reader to understand whether the model formulation is realistic, though I am pretty convinced the perturbation (applying SSTs from another model) is NOT realistic for reasons described above.**

Vertical advection of trace species in the model is treated using the large scale vertical velocity calculated in the dynamical core of the CCM. Ice formation via homogeneous freezing in the upper troposphere is produced by updraft on sub-grid scales (see Kärcher and Lohmann, 2002; Lohmann and Kärcher, 2002). The latter is parameterized using the TKE formulation, as explained in the same referenced studies. For what concerns the SST specification see above in response to the overall comment and in other specific comments.

**Is a 3% change in a parameterized vertical velocity significant? Is 10% significantly different from 3%? From Figure 6, I don't think any of this is significant.**

Figure 6 showed the variability of the calculated vertical velocity (large scale + f(TKE) mesoscale contribution from synoptic scale and gravity wave motions) and the time-averaged mean values. Changes in temperature and wind profiles produced by the SG aerosol forcing are related to a TOARF of the order of -1 W/m$^2$ and produce a change in TKE of the order of -120 cm$^2$/s$^2$ in the tropical upper troposphere in G4 relative to the Base case, i.e. close to -20%. Following the parameterization developed in Lohmann and Kärcher (2002), w is taken as the sum of the large-scale term (of the order of 0.2 cm/s in the tropical UT) and 0.7×TKE$^{0.5}$ (of the order of 17 cm/s in the tropical UT) (see Eq. 1 in the revised manuscript). A change in TKE of approximately -120 cm$^2$/s$^2$ translates in a change of -1.8 cm/s of w, i.e. close to -10%. The G4K vertical profile of w is intermediate between G4 and Base, because TKE changes result only from the lower stratospheric aerosol heating, with surface temperature kept fixed at the reference RCP scenario. This ends up in a w change of approximately -3% in the tropical UT. The SG perturbation of the temperature profile is obviously small relative to baseline atmospheric conditions, both in G4 and G4K, but these small changes are exactly those impacting the atmospheric static stability and vertical motions. And differences in G4K and G4 are proved to be significant from this point of view.

To better clarify, we note that the variability of w in Figure 6 is essentially due to seasonal changes and non-zonal asymmetries of the TKE. But if we isolate a given month in the time series, the vertical velocity change due to SG is more comparable to the w variability in the time series. We attach a figure below (Fig. R2_1) showing this quantity, to show the reviewer what we mean.

[Figure]

Fig. R2_1. October monthly mean of the upper tropospheric tropical profiles of vertical velocity (cm/s) in G4, G4K and Base experiments (years 2030-39). Shaded areas represent ±1σ for the ensemble over the October month in the 10 year period 2030-39.

**MODIS ice effective radius is not a reasonable product, especially for thin tropical cirrus, unless you have a validation paper that says otherwise.**

As per the reviewer request, we have tried to add some peer reviewed references to the MODIS ice effective radius, in particular Yang et al. (2007). We will also discuss some of the limitations regarding the retrieval of the ice effective radius (Delanoe and Hogan, 2008; Zhang et al.,2010).

**The mention of what looks like a maximum updraft velocity here is an indication that the ULAQ ice nucleation needs to be better explained.**

Ice nucleation is now presented in a more complete way in the revised manuscript.

**This section needs to go before all the results presented earlier.**

We have done what the reviewer suggested, also following the recommendation of reviewer 1.

**It's not clear to me what fraction of ice formed in situ (T<238K) is from homogeneous and heterogeneous freezing. It would be useful to note the fraction homogenous (or heterogenous). This looks like it is in Figure 10c, but I don't think that is what I am interested in. What fraction of ice is heterogenously formed?**

Cirrus ice formation in the ULAQ-CCM results from both homogeneous and heterogeneous freezing mechanisms and their competition. However, in our first draft of the manuscript we had decided to turn off the heterogeneous freezing mechanism, in order to focus on the SG aerosol induced perturbation to ice formation from homogeneous freezing only.

We acknowledge this specific point of the reviewer. The reduced updraft will affect ice formation from homogeneous freezing in a different way if part of the available water vapor goes to ice particles formed via heterogeneous freezing, which requires smaller supersaturation ratios, both on mineral dust and black carbon particles. Following the reviewer suggestion, we have decided to perform again our simulations with both mechanisms turned on, allowing their non-linear interaction. Results are substantially affected, with a resulting smaller indirect RF due to upper tropospheric ice changes induced by sulfate geoengineering.

Looking at the revised results of our numerical experiments, we are really indebted with the reviewer(s) for making this specific scientific point, helping us in a more correct assessment of the ice perturbation due to SG aerosol and its indirect radiative forcing.

**This is a decent summary that the changes are due to changes in vertical velocity and tropospheric temperatures. How model dependent do you think these quantities are?**

Results of our numerical experiments are obviously dependent on model features and design. However, one major point of our work was to systematically compare our results with observed ice-related quantities, on one hand, and to an independent modelling work (Kuebbeler et al., 2012) for the SG-related ice perturbation, on the other hand. It is shown that our results are consistent and that inclusion of SST changes may be significant, following a suggestion explicitly made in Kuebbeler et al. (2012).

**Why the 5/8 scaling of the RF results?**

We wanted to compare our results to their Clear Sky RF, and as a first approximation we scaled our results to their injection rate. However, we recognize that this might be confusing to the reader, and we now compare the direct results of our model with those from Kuebbeler et al. (2012).

**How realistic is the decrease in updraft? Is it consistent with the overall circulation? I am concerned that fixing SSTs from another model will not yield a reasonable result, and it is likely to be a single model configuration, not even a general result. How can you convince me and other readers that the mechanism in ULAQ is reasonable, especially since it is imposed from another model and not-interactive, and from a model with no stratosphere.**

We believe that we have widely responded above to these specific points. In particular, the use of SSTs as input parameter in CCMs is intrinsic in the CCM nature and formulation itself. A comparison of the SG aerosol optical depth and extinction from CCSM-CAM4 is presented in Fig. R2_2 and Fig. R2_3, respectively (attached below), with those predicted and fully interactive in the ULAQ-CCM (Visioni et al., 2017b). This proves that the two aerosol latitudinal and vertical distributions are consistent, so that the aerosol direct radiative forcing applied in CCSM-CAM4 and regulating SST changes due to SG is consistent with that in the ULAQ-CCM. Finally, it is not true that CCSM-CAM4 has no stratosphere.

[Figure]

Fig. R2_2. Annually and zonally averaged SG aerosol optical depth at λ=0.55 µm used in CCSM-CAM4 and calculated in our study with the ULAQ-CCM.

[Figure]

Fig. R2_3. Annually and zonally averaged SG aerosol extinction at λ=0.55 µm ($10^{-3}$ $km^{-1}$) used in CCSM-CAM4 (left panel) and calculated in our study with the ULAQ-CCM (right panel).

References:

Bosilovich, M. G., Robertson, F. R., Takacs, L., Molod, A., and Mocko, D.: Atmospheric Water Balance and Variability in the MERRA-2 Reanalysis, Journal of Climate, 30, 1177–1196, doi:10.1175/JCLI-D-16-0338.1, https://doi.org/10.1175/JCLI-D-16-0338.1, 2017.

Chou, M. M. J. S. X. L.: A thermal infrared radiation parameterization for atmospheric studies, Tech. Rep. TM-2001-104606, NASA, NASA Goddard Space Flight Cent., Greenbelt, MD, 2001.

Delanoe, J., and R. J. Hogan (2008), A variational scheme for retrieving ice cloud properties from combined radar, lidar, and infrared radiometer, J. Geophys. Res., 113, D07204, doi:10.1029/2007JD00900

Duncan, D. I. and Eriksson, P.: An update on global atmospheric ice estimates from satellite observations and reanalyses, Atmos. Chem. Phys. Discuss., https://doi.org/10.5194/acp-2018-275, in review, 2018.

Gelaro, R., McCarty, W., Suárez, M. J., Todling, R., Molod, A., Takacs, L., Randles, C. A., Darmenov, A., Bosilovich, M. G., Reichle, R., Wargan, K., Coy, L., Cullather, R., Draper, C., Akella, S., Buchard, V., Conaty, A., da Silva, A. M., Gu, W., Kim, G. K., Koster, R., Lucchesi, R., Merkova, D., Nielsen, J. E., Partyka, G., Pawson, S., Putman, W., Rienecker, M., Schubert, S. D., Sienkiewicz, M., and Zhao, B.: The modern-era retrospective analysis for research and applications, version 2 (MERRA-2), J. Climate, 30, 5419–5454, https://doi.org/10.1175/JCLI-D-16-0758.1, 2017.

Kärcher, B. and Lohmann, U.: A parameterization of cirrus cloud formation: Homogeneous freezing of supercooled aerosols, Journal of Geophysical Research: Atmospheres, 107, doi:10.1029/2001JD000470, http://dx.doi.org/10.1029/2001JD000470, 2002.

Kuebbeler, M., Lohmann, U., and Feichter, J.: Effects of stratospheric sulfate aerosol geo-engineering on cirrus clouds, Geophysical Research Letters, 39, doi:10.1029/2012GL053797, http://dx.doi.org/10.1029/2012GL053797, l23803, 2012.

Lamarque, J.-F., Emmons, L. K., Hess, P. G., Kinnison, D. E., Tilmes, S., Vitt, F., Heald, C. L., Holland, E. A., Lauritzen, P. H., Neu, J., Orlando, J. J., Rasch, P. J., and Tyndall, G. K.: CAM-chem: description and evaluation of interactive atmospheric chemistry in the Community Earth System Model, Geosci. Model Dev., 5, 369-411, https://doi.org/10.5194/gmd-5-369-2012, 2012.

Lohmann, U. and Kärcher, B.: First interactive simulations of cirrus clouds formed by homogeneous freezing in the ECHAM general circulation model, Journal of Geophysical Research: Atmospheres, 107, AAC 8–1–AAC 8–13, doi:10.1029/2001JD000767, http://dx.doi.org/10.1029/2001JD000767, 2002.

Morgenstern, O., Hegglin, M. I., Rozanov, E., O'Connor, F. M., Abraham, N. L., Akiyoshi, H., Archibald, A. T., Bekki, S., Butchart, N., Chipperfield, M. P., Deushi, M., Dhomse, S. S., Garcia, R. R., Hardiman, S. C., Horowitz, L. W., Jockel, P., Josse, B., Kinnison, D., Lin, M., Mancini, E., Manyin, M. E., Marchand, M., Marecal, V., Michou, M., Oman, L. D., Pitari, G., Plummer, D. A., Revell, L. E., Saint-Martin, D., Schofield, R., Stenke, A., Stone, K., Sudo, K., Tanaka, T. Y., Tilmes, S., Yamashita, Y., Yoshida, K., and Zeng, G.: Review of the global models used within phase 1 of the Chemistry–Climate Model Initiative (CCMI), Geoscientific Model Development, 10, 639–671, doi:10.5194/gmd-10-639-2017, https://www.geosci-model-dev.net/10/639/2017/, 2017.

Neale, R.B., J. Richter, S. Park, P.H. Lauritzen, S.J. Vavrus, P.J. Rasch, and M. Zhang: The Mean Climate of the Community Atmosphere Model (CAM4) in Forced SST and Fully Coupled Experiments. *J. Climate,* **26**, 5150–5168, https://doi.org/10.1175/JCLI-D-12-00236.1 , 2013.

Niemeier, U. and Schmidt, H.: Changing transport processes in the stratosphere by radiative heating of sulfate aerosols, Atmospheric Chemistry and Physics, 17, 14 871–14 886, doi:10.5194/acp-17-14871-2017, https://www.atmos-chem-phys.net/17/14871/2017/, 2017.

Pitari, G., Aquila, V., Kravitz, B., Robock, A., Watanabe, S., Cionni, I., De Luca, N., Di Genova, G., Mancini, E., and Tilmes, S.: Stratospheric ozone response to sulfate geoengineering: Results from the Geoengineering Model Intercomparison Project (GeoMIP), Journal of Geophysical Research: Atmospheres, 119, 2629–2653, 2014.

Pitari, G., Cionni, I., Di Genova, G., Visioni, D., Gandolfi, I., and Mancini, E.: Impact of Stratospheric Volcanic Aerosols on Age-of-Air and Transport of Long-Lived Species, Atmosphere, 7, http://www.mdpi.com/2073-4433/7/11/149, 2016a.

Pitari, G., Di Genova, G., Mancini, E., Visioni, D., Gandolfi, I., and Cionni, I.: Stratospheric Aerosols from Major Volcanic Eruptions: A Composition-Climate Model Study of the Aerosol Cloud Dispersal and e-folding Time, Atmosphere, 7, 75, doi:10.3390/atmos7060075, http://dx.doi.org/10.3390/atmos7060075, 2016b.

Randles, C. A., Kinne, S., Myhre, G., Schulz, M., Stier, P., Fischer, J., Doppler, L., Highwood, E., Ryder, C., Harris, B., Huttunen, J., Ma, Y., Pinker, R. T., Mayer, B., Neubauer, D., Hitzenberger, R., Oreopoulos, L., Lee, D., Pitari, G., Di Genova, G., Quaas, J., Rose, F. G., Kato, S., Rumbold, S. T., Vardavas, I., Hatzianastassiou, N., Matsoukas, C., Yu, H., Zhang, F., Zhang, H., and Lu, P.: Intercomparison of shortwave radiative transfer schemes in global aerosol modeling: results from the AeroCom Radiative Transfer Experiment, Atmospheric Chemistry and Physics, 13, 2347–2379, doi:10.5194/acp-13-2347-2013, http://www.atmos-chem-phys.net/13/2347/2013/, 2013.

Tilmes, S., Mills, M. J., Niemeier, U., Schmidt, H., Robock, A., Kravitz, B., Lamarque, J.-F., Pitari, G., and English, J. M.: A new Geoengineering Model Intercomparison Project (GeoMIP) experiment designed for climate and chemistry models, Geosci. Model Dev., 8, 43-49, https://doi.org/10.5194/gmd-8-43-2015, 2015.

Tilmes, S., Lamarque, J.-F., Emmons, L. K., Kinnison, D. E., Marsh, D., Garcia, R. R., Smith, A. K., Neely, R. R., Conley, A., Vitt, F., Val Martin, M., Tanimoto, H., Simpson, I., Blake, D. R., and Blake, N.: Representation of the Community Earth System Model (CESM1) CAM4-chem within the Chemistry-Climate Model Initiative (CCMI), Geosci. Model Dev., 9, 1853-1890, https://doi.org/10.5194/gmd-9-1853-2016, 2016.

Timmreck, C., Mann, G. W., Aquila, V., Hommel, R., Lee, L. A., Schmidt, A., Brühl, C., Carn, S., Chin, M., Dhomse, S. S., Diehl, T., English, J. M., Mills, M. J., Neely, R., Sheng, J., Toohey, M., and Weisenstein, D.: The Interactive Stratospheric Aerosol Model Intercomparison Project (ISA-MIP): Motivation and experimental design, Geosci. Model Dev. Discuss., https://doi.org/10.5194/gmd-2017-308, in review, 2018.

Visioni, D., Pitari, G., and Aquila, V.: Sulfate geoengineering: a review of the factors controlling the needed injection of sulfur dioxide, Atmos. Chem. Phys., 17, 3879-3889, https://doi.org/10.5194/acp-17-3879-2017, 2017a.

Visioni, D., Pitari, G., Aquila, V., Tilmes, S., Cionni, I., Di Genova, G., and Mancini, E.: Sulfate geoengineering impact on methane transport and lifetime: results from the Geoengineering Model Intercomparison Project (GeoMIP), Atmospheric Chemistry and Physics, 17, 11 209– 11 226, doi:10.5194/acp-17-11209-2017, https://www.atmos-chem-phys.net/17/11209/2017/, 2017b.

Visioni, D., Pitari, G., Tuccella, P., and Curci, G.: Sulfur deposition changes under sulfate geoengineering conditions: quasi-biennial oscillation effects on the transport and lifetime of

stratospheric aerosols, Atmos. Chem. Phys., 18, 2787-2808, https://doi.org/10.5194/acp-18-2787-2018, 2018.

Yang, P., Zhang, L., Hong, G., Nasiri, S. L., Baum, B. A., Huang, H. L., King, M. D., and Platnick, S.: Differences Between Collection 4 and 5 MODIS Ice Cloud Optical/Microphysical Products and Their Impact on Radiative Forcing Simulations, IEEE Transactions on Geoscience and Remote Sensing, 45, 2886–2899, doi:10.1109/TGRS.2007.898276, 2007.

Zhang, Z., Platnick, S., Yang, P., Heidinger, A. K., and Comstock, J. M.: Effects of ice particle size vertical inhomogeneity on the passive remote sensing of ice clouds, Journal of Geophysical Research: Atmospheres, 115, doi:10.1029/2010JD013835, https://agupubs. onlinelibrary.wiley.com/doi/abs/10.1029/2010JD013835, 2010.

---

## Referee Report (RR1)

This is a valuable contribution to the underrepresented topic of cirrus responses to stratospheric sulphur injections.  While I think the authors did a good job in explaining the main physical mechanism behind the observed changes, I am pointing out a few more issues, which would need to be addressed before the paper can be published in final form.

**Major comments**

1.) Is detrained moisture/ice water content from convection included in the cirrus formation mechanisms? How did you include it?

You mention on page 7 that upper tropospheric ice can be formed only by homogeneous or heterogeneous freezing. However, a large part of the cirrus, in particularly in the tropics, is formed by detrainment of ice crystals from deep convective cores.  Such ice crystals formed either by homogeneous nucleation of cloud droplets or in mixed phase by heterogeneous nucleation; their formation is therefore significantly different from the in-situ cirrus.

Did you include such detrained ice crystal sources in your model? I think the strength and level of maximum detrainment is probably modulating the responses of in-situ formed cirrus in the tropics, i.e. in the region where most of your cirrus cloud radiative effect comes from.

Most of your ice mass comes from heterogeneous freezing at lower elevations in the tropics. This is in a zonal average perspective not realistic, as most of it should be a result of detrainment from deep convection, at least near the location of the intertropical convergence zone.

Detrainment from deep convective clouds is an important source of cirrus clouds and therefore needs to be mentioned/commented in the manuscript.

2.) Model evaluation with MERRA2/MODIS data

I. Please state which version of MODIS data you use. You cite Yang et al., 2007, which is a reference for the V5. I assume you either use V5 or V6, please add this as the retrievals changed between several product versions. Do you use level 3 1x1° gridded data?

II. I would suggest removing the use of MODIS IC radius due to the following reasons:

- MODIS derived IC radius is valid only for cloud tops of optically thicker clouds and not representative of the whole cloud distribution. In a thick cloud, the MODIS IC effective radius would correspond to the upper

portion of the thick(er) cloud, until the optical depths of about 1.2, at least for the case of detrained anvil clouds as shown in Hong et al., 2012. The retrieval would give more weight to the radius closer to cloud top also for the case of intermediately thick cirrus (COD between 1 and 5, Zhang et al., 2010).

- MODIS cannot see the thinnest of the cirrus clouds. Its approximate detection limit is close to COD of 0.4 (Ackerman et al., 2008). I assume you include clouds of any optical depth in your analysis.
- MODIS is a passive instrument and detects cloud properties only during daytime, while I assume you take both day and night data from the model output

To summarize my point, the comparison of IC radius and the derived IC number concentration is based on too many very shaky assumptions and needs to be removed from the manuscript. If you would like to keep it, you may use the MODIS satellite simulator, which takes into account MODIS retrieval limitations and therefore ensures an apple-to-apple comparison.

III. MERRA2

MERRA2 has a very simplistic treatment of ice clouds, leading to large biases (e.g large biases in cloud radiative effect noted in Bosilovich et al., 2015). Using a reanalysis dataset is anyway not the best, but if you already went for one, ERA5 would be a more appropriate choice, as it compares better with CALIPSO-CloudSat datasets (DARDAR, 2C-ICE) as shown for instance by Duncan and Eriksson, 2018. Nevertheless, considering this is the second phase of review, I can accept the comparison used in Figure 1 as good enough due to large IWC retrieval uncertainty (as you also pointed out in the manuscript).

Yet, I think you should remove from the paper your optical depth estimates from MERRA+MODIS in figure 2, as the assumptions behind that plot are too large and you are mixing up reanalysis, satellite retrieval, and model output without making sure this is an "apple-to-apple" comparison (i.e. you don't take into account the satellite retrieval limitations and the issue of collocation of data in space and time).

**Minor comments**

- please add uncertainty estimates (e.g. +/- 1 st. dev.) to the results you show, at least in the tables. This would give the reader a better feeling for the significance of your radiative forcing anomalies.

Abstract:
After line 15 the abstract clarity becomes challenging for the reader as you are making very fast transitions from effects of cirrus clouds which cool the climate, to comparing all-sky with clear-sky forcing, and saying that the all-sky has a

positive effect on the radiative balance. I would just qualitatively mention the effect of a positive (total) cloud radiative effect -> dimming the sun that reaches the cloud tops indeed has to decrease the amount of reflected SW radiation. Moreover, do you really need to always mention 2 significant numbers after the decimal point, considering all the uncertainties?

page 2, line 21-24:
The current best knowledge of cirrus microphysics does not show much support of the predominance of homogeneous nucleation in in-situ cirrus cloud.
Your extensive answer to reviewer #1 unfortunately does not help in changing that view. I think the uncertainty in cirrus formation mechanisms is high enough to accept your modelling results related to the freezing mechanisms as plausible.

What does it mean that homogeneous processes dominate the heterogeneous? Do you refer to the relative radiative forcing difference, the ice water content, ice crystal number concentration, frequency of occurrence of nucleation events?

Moreover, I am not sure whether figures from the latest ECHAM-HAM studies (e.g. Gasparini and Lohmann 2016, Gasparini et al., 2018) confirm your homogeneous vs. heterogeneous nucleation arguments. Homogeneous freezing seems to dominate only near the tropopause and over mountains.
You could also cite Barahona et al., 2017, which shows somewhat consistent results with Gasparini and Lohmann 2016 in terms of homogeneous vs. heterogeneous freezing importance.

page 2, line 33/34 (and on page 33):
I don't think Sanderson et al., 2008 is looking at radiative balance of upper tropospheric clouds, but rather at the sensitivity of climate feedback to tuning parameters.
Also, Mitchell et al., 2008 look at differences in simulated climate by changing the particle shape distributions, affecting the fall velocities, and finally the radiative effects of clouds.

page 3, line 5
Liquid (or more precisely aqueous) sulphuric acid droplets CANNOT act as ice nucleating particles for heterogeneous freezing.
The increase in IC number concentration in Cirisan et al., 2013 is related to the presence of large sulphuric acid particles, which makes homogeneous freezing more favourable. Sulphuric particles at $r<0.1$ μm only hardly nucleate ice crystals homogenously due to the strong Kelvin effect. Stratospheric perturbations shift this distribution closer to sulphuric aerosol radii between 0.6 and 0.8, which were shown to be most susceptible for homogeneous freezing (see paragraph 2.3 of Cirisan et al.).

page 4, lines 5-10
The discussion seems to clearly highlight the thinning of cirrus in presence of a volcanic forcing. I think that by our current best knowledge we cannot give a conclusive answer on the influence of volcanic eruptions on cirrus clouds

frequency, microphysics, or radiative properties (e.g. Meyer et al., 2015 has a different conclusion from the study you cited).

page 7, line 18
Homogeneous freezing threshold is not constant, but should have some temperature (or, more precisely, water activity) dependence. Many parameterizations follow the Koop et al., 2000 results/formula. You have to therefore mention that important shortcoming, which might lead in most places to some overestimation of your homogeneous freezing probability, and the opposite at temperatures close to the homogeneous freezing temperature of water.

page 7, line 32-33:
Again, I don't think there is much evidence for the dominant role of homogeneous freezing. At most, you can mention that the relative importance of homogeneous vs. heterogeneous freezing is currently still very uncertain. The cited study with the message: "beware of the coating of dust, which decreases the ability to nucleate ice of several ice nucleating particles" (i.e. Cziczo et al., 2009) is not a proof of your statement!

page 10, line 8-10:
That's surprising; I would expect that the IWC at the lowest levels is dominated by detrained sources. Indeed, you might be just looking at $IWC_{het}/IWC_{hom}$, which is OK, but you need to mention in this case the missing and probably large convective IWC source in the tropics below about 12 km.

page 11, line 9-15:
I do not see much value in the comparison of your globally averaged ice crystal number concentration with a randomly picked study from a field campaign (which is, moreover, likely affected by the pre early 2000s retrieval problems due to ice crystal shattering, see Cziczo et al., 2014).
Again, I also do not see any reason to trust the MERRA+MODIS derived IC number concentrations on Figure 4.

Figure 12 and related text:
Again, the derived extinction from MERRA+MODIS does not add much of scientific value.
Same for Figure 13 b.

page 30, line 3-5:
Background clouds have a positive cloud radiative effect. That means they reflect less (if we assume all comes from SW), and not more!
If the solar radiation reaching top of the clouds decreases by, say, 1%, the amount of reflected SW radiation has to also decrease by the same relative value to first order (1% in this example).

Figure 14:
Why is the background cloud effect plotted only once? I guess it does change between the two cases.

Figure 15:
You never show that there is reduced water vapour transport to the upper troposphere? Prove it or remove it!
Also, I would like to see some evidence for the "convectively driven tropospheric cooling" before putting that in your summary sketch!
In summary, your schematic is a bit too complicated to be easily digested by an average reader. I think you can drop a few of the points, unless you prove them to be crucial in delivering your message.

page 31, lines 27-28:
Or maybe simply the water cycle slows down due to decrease of surface temperature, following Clausius-Clapeyron?

page 33, lines 14-16:
This is not really a good explanation of the SW adjustment. It is rather confusing to the reader. I thought you do not include cirrus in the "background clouds" effects based on your Figure 14, which shows the background effect separately from the effect on ice clouds.

References

Ackerman et al., 2008: Cloud detection with MODIS. Part II: Validation
Bosilovich et al., 2015: MERRA-2: Initial Evaluation of the Climate
Cziczo et al., 2014: Sampling the composition of cirrus residuals
Duncan and Eriksson, 2018: An update on global atmospheric ice estimates from satellite observations and reanalyses
Hong et al., 2012: Estimating effective particle size of tropical deep convective clouds with a look-up table method using satellite measurements of brightness temperature differences
Koop et al., 2010: Water activity as the determinant for homogeneous ice nucleation in aqueous solutions
Meyer et al., 2015: Did the 2011 Nabro eruption affect the optical properties of ice clouds?
Yang et al., 2007: Differences Between Collection 4 and 5 MODIS Ice Cloud Optical/Microphysical Products and Their Impact on Radiative Forcing Simulations
Zhang et al., 2010: Effects of ice particle size vertical inhomogeneity on the passive remote sensing of ice clouds

---

## Author Response (AR2)

Dear Editor,

first of all, we would like to thank you for giving us the opportunity to further prove the scientific robustness of our study.

As you have requested, we have performed new simulations, where changes in key atmospheric components (namely: chemical species, tropospheric ice and differences between our sulfate distribution and that of CCSM-CAM4) and their radiative effects are coupled on-line with surface temperature perturbations. In the responses to Reviewers 2 and 3 and in the revised manuscript we explain in detail how this was made.

Furthermore, as you requested, we have responded to all the objections posed by the new reviewers. Considering a specific suggestion of the Reviewers, we have further expanded the evaluation of our model's output with ERA5 and MLS data.

A point by point response to the reviews is attached below, together with a marked-up copy of the manuscript with all the changes tracked.

Daniele Visioni on behalf of all authors

**Response to Reviewer #2.**

Referee comments are in black, author responses are in blue.

I appreciate the authors' time in a careful revision of the manuscript, but I am unable to recommend publication in ACP for this paper.

I am still not convinced that the method the authors have chosen is appropriate, and the authors attempts to convince me of this are insufficient.

I do not think that you can impose SST perturbations from one model on another model, and expect that they will not have an impact. Because the models are different, the CCSM state applied to ULAQ is going to create imbalances that are not realistic, and would not be seen in CCSM4. Furthermore, CCSM4 itself does not have a realistic stratospheric circulation, so that any SST response to sulfate geoengineering is approximate. This would be fine, except there is a more sophisticated version of that model available with a stratosphere, and several papers have been written using fully coupled (to an ocean) versions of this model.

I think you are attempting to attribute changes that may be affected by the method and have nothing to do with a response to geoengineering.

**I do not see how the present study adds to the literature, except to confuse the issues, and do not see how the authors would be able to get around this point. I am sorry, but I think this method is ill posed.**

We understand the Reviewer's point and decided to explicitly calculate SG-driven changes in surface temperatures in the ULAQ-CCM at any grid point and time step. This is done by superimposing to the background RCP4.5 surface temperatures calculated in the CCSM-CAM4 model (and used as prescribed boundary condition in the ULAQ-CCM itself) the SG-driven surface temperature perturbation associated with the ULAQ-CCM radiative flux changes produced by SG with an injection of 8 Tg-SO2/yr. These radiative flux changes are those produced by stratospheric sulfate aerosols and upper tropospheric ice particles, as well as those produced by changes of greenhouse gases directly and indirectly affected by stratospheric geoengineering aerosols (i.e., O3, H2O, CH4 and CO2 from changing methane oxidation). The ULAQ-CCM calculated SG effects on these greenhouse gases were documented in Pitari et al. (2014) and Visioni et al. (2017). Details on the ULAQ-CCM calculation of SG aerosol microphysics, size distribution, optical thickness, transport, strat-trop exchange and radiative impact are given in Visioni et al. (2018), as well as in Pitari et al. (2014).

In order to minimize the approximation introduced from a missing explicit ocean module in the ULAQ-CCM, the procedure adopted in the calculation of the SG perturbation on surface temperatures is described below.

(a) As a first approximation, we use in our G4 experiment the CCSM-CAM4 predicted surface temperatures with inclusion of the radiative impact of geoengineering aerosols (with injection of 8 Tg-SO2/yr); this ocean-atmosphere coupled simulation does not include chemistry and upper tropospheric ice particle changes induced by SG. As discussed in Tilmes et al. (2016), the stratospheric aerosol distribution used in CCSM-CAM4 is sufficiently robust and validated. As clearly shown in Table 3 of the revised

manuscript, the net tropopause RF from these aerosols represent by far the largest contribution to the net SG RF.

- (b) However, as the Reviewer points out, the aerosol distribution calculated on-line in the ULAQ-CCM (which is based on a well-tested microphysics scheme and detailed stratosphere) may be different from the one in CCSM-CAM4, thus introducing a potentially significant inconsistency in the modeling scheme. To correct for this undesired effect, the ULAQ-CCM radiative-climate module has been modified for calculating on-line (in a fully coupled approach) the surface temperature perturbation produced by radiative flux changes due to the sulfate aerosol imbalance with respect to the CCSM-CAM4 distribution. In addition, we also include in the radiative balance the SG-driven indirect perturbation of greenhouse gases (see above), as well as upper tropospheric ice particles (which are the focus of the present study). Table S1, Fig. S2 and Fig. 6 of the revised manuscript document the effects of these radiative flux changes on the calculated surface temperatures.
- (c) Surface temperature changes due to the above discussed indirect SG effects are calculated from the instantaneous perturbation of radiative fluxes, which is of course an exact procedure over continents and polar ice caps, whereas is only approximate over the oceans. On the other hand, as well explained above and clearly visible in Table S1, Fig. S2 and Fig. 6, the radiative perturbation additive to the dominant one (that is the one produced by stratospheric sulfate aerosols in the CCSM-CAM4 simulation) is normally small, both globally and locally. Only the ice induced changes of surface temperatures may be comparable in magnitude to those from the stratospheric aerosols, but limited to tropical continental surfaces, where UT ice may have a significant optical depth. On the other hand, the SST calculated changes due to chemistry and ice indirect effects of SG are usually smaller, so that the impact of our approximation may be expected to be negligible.

**References:**

Pitari, G., Aquila, V., Kravitz, B., Robock, A., Watanabe, S., Cionni, I., De Luca, N., Di Genova, G., Mancini, E., and Tilmes, S.: Strato- spheric ozone response to sulfate geoengineering: Results from the Geoengineering Model Intercomparison Project (GeoMIP), Journal of Geophysical Research: Atmospheres, 119, 2629–2653, 2014.

Tilmes, S., Lamarque, J.-F., Emmons, L. K., Kinnison, D. E., Marsh, D., Garcia, R. R., Smith, A. K., Neely, R. R., Conley, A., Vitt, F., Val Martin, M., Tanimoto, H., Simpson, I., Blake, D. R., and Blake, N.: Representation of the Community Earth System Model (CESM1) CAM4-chem within the Chemistry-Climate Model Initiative (CCMI), Geoscientific Model Development, 9, 1853–1890, doi:10.5194/gmd- 9-1853-2016, https://www.geosci-model-dev.net/9/1853/2016/, 2016.

Visioni, D., Pitari, G., Aquila, V., Tilmes, S., Cionni, I., Di Genova, G., and Mancini, E.: Sulfate geoengineering impact on methane transport and lifetime: results from the Geoengineering Model Intercomparison Project (GeoMIP), Atmospheric Chemistry and Physics, 17, 11 209– 11 226, doi:10.5194/acp-17-11209-2017, https://www.atmos-chem-phys.net/17/11209/2017/, 2017.

Visioni, D., Pitari, G., Tuccella, P., and Curci, G.: Sulfur deposition changes under sulfate geoengineering conditions: quasi-biennial oscilla- tion effects on the transport and lifetime of stratospheric aerosols, Atmospheric Chemistry and Physics, 18, 2787–2808, doi:10.5194/acp- 18-2787-2018, https://www.atmos-chem-phys.net/18/2787/2018/, 2018.

**Response to Reviewer #3.**

**Reviewer comments are in black, author responses are in blue.**

The manuscript by Visioni et al. aims at investigating the impact of geoengineering by stratospheric sulfur injections on upper tropospheric cirrus formations using the ULAQ-CCM. My comments below refer only to the revised version of the manuscript. I was not involved in the review process of the first manuscript version, but from the authors' response to the Reviewer comments I understand that they reran the model simulations due to an inadequate model setup in the first round.

**Overall comment:**

My overall impression is that the study indeed targets an interesting and important side effect of sulfate geoengineering, namely changes in upper tropospheric cirrus clouds, but that the chosen methodology and the performed analysis of the model results are flawed. I am afraid that the ULAQ-CCM is simply not the right tool for such an investigation (details see below). At least from what is written in the paper I am not convinced that the results are valid and provide substantial new insights. In the present form I cannot recommend this paper for publication in ACP.

**General comments:**

1) Model approach: Although I agree with the authors that it is a common approach to use sea surface temperatures and sea ice coverage from coupled ocean-atmosphere models in chemistry-climate models, I think that this approach is not appropriate for all kind of research questions. In the present case I have strong doubts that the change in sea surface temperatures as simulated by CCSM-CAM4 for G4 (no chemistry, simplified ice cloud scheme) is consistent with ULAQ-CCM's G4 aerosol distributions and the change in ozone, clouds, etc. From Fig. R2\_2 and R2\_3 it is obvious that both models show substantial differences in the SG aerosol distribution and AOD. Therefore, using the SSTs from CCSM-CAM4 is as consistent as applying an artificial negative SST anomaly. In my opinion the only meaningful approach would be to use a coupled ocean-atmosphere model with aerosol scheme and interactive chemistry. Such models are in the meantime available, although probably quite expensive. Even with such models I would expect a large spread in the upper tropospheric cirrus response to SG due to uncertainties in parameterized processes.

We understand the Reviewer's point and decided to explicitly calculate SG-driven changes in surface temperatures in the ULAQ-CCM at any grid point and time step. This is done by superimposing to the background RCP4.5 surface temperatures calculated in the CCSM-CAM4 model (and used as prescribed boundary condition in the ULAQ-CCM itself) the SG-driven surface temperature perturbation associated with the ULAQ-CCM radiative flux changes produced by SG with an injection of 8 Tg-SO2/yr. These radiative flux changes are those produced by stratospheric sulfate aerosols and upper tropospheric ice particles, as well as those produced by changes of greenhouse gases directly and indirectly affected by stratospheric

geoengineering aerosols (i.e.,  $O_3$ ,  $H_2O$ ,  $CH_4$  and  $CO_2$  from changing methane oxidation). The ULAQ-CCM calculated SG effects on these greenhouse gases were documented in Pitari et al. (2014) and Visioni et al. (2017b). Details on the ULAQ-CCM calculation of SG aerosol microphysics, size distribution, optical thickness, transport, strat-trop exchange and radiative impact are given in Visioni et al. (2018), as well as in Pitari et al. (2014).

In order to minimize the approximation introduced from a missing explicit ocean module in the ULAQ-CCM, the procedure adopted in the calculation of the SG perturbation on surface temperatures is described below.

- (a) As a first approximation, we use in our G4 experiment the CCSM-CAM4 predicted surface temperatures with inclusion of the radiative impact of geoengineering aerosols (with injection of 8 Tg-SO2/yr); this ocean-atmosphere coupled simulation does not include chemistry and upper tropospheric ice particle changes induced by SG. As discussed in Tilmes et al. (2016), the stratospheric aerosol distribution used in CCSM-CAM4 is sufficiently robust and validated. As clearly shown in Table 3 of the revised manuscript, the net tropopause RF from these aerosols represent by far the largest contribution to the net SG RF.
- (b) However, as the Reviewer points out, the aerosol distribution calculated on-line in the ULAQ-CCM (which is based on a well-tested microphysics scheme and detailed stratosphere) may be different from the one in CCSM-CAM4, thus introducing a potentially significant inconsistency in the modeling scheme. To correct for this undesired effect, the ULAQ-CCM radiative-climate module has been modified for calculating on-line (in a fully coupled approach) the surface temperature perturbation produced by radiative flux changes due to the sulfate aerosol imbalance with respect to the CCSM-CAM4 distribution. In addition, we also include in the radiative balance the SG-driven indirect perturbation of greenhouse gases (see above), as well as upper tropospheric ice particles (which are the focus of the present study). Table S1, Fig. S2 and Fig. 6 of the revised manuscript document the effects of these radiative flux changes on the calculated surface temperatures.
- (c) Surface temperature changes due to the above discussed indirect SG effects are calculated from the instantaneous perturbation of radiative fluxes, which is of course an exact procedure over continents and polar ice caps, whereas is only approximate over the oceans. On the other hand, as well explained above and clearly visible in Table S1, Fig. S2 and Fig. 6, the radiative perturbation additive to the dominant one (that is the one produced by stratospheric sulfate aerosols in the CCSM-CAM4 simulation) is normally small, both globally and locally. Only the ice induced changes of surface temperatures may be comparable in magnitude to those from the stratospheric aerosols, but limited to tropical continental surfaces, where UT ice may have a significant optical depth. On the other hand, the SST calculated changes due to chemistry and ice indirect effects of SG are usually smaller, so that the impact of our approximation may be expected to be negligible.

2) Sect. 2.3: The description of the experimental set up is very confusing and imprecise. Do you use sea surface temperatures and sea ice coverage only or also land surface temperatures from CCSM-CAM? Is it a nudging approach or a prescribed boundary condition? There are fundamental differences. Nudging is a Newtonian relaxation technique which adds non-physical terms to the models' equations to "pull" certain variables like temperatures towards observed values. Scientific inaccuracy is a general problem throughout the whole manuscript.

We acknowledge that our description of this part of the experimental set-up is not clear enough. The right definition, as the Reviewer suggests, is that of "prescribed boundary conditions". CCSM-CAM4 surface temperatures are used as boundary conditions in the baseline RCP4.5 and G4 simulations (2020-2069) of the ULAQ-CCM on both sea surface and land. The historical reference case (1960-2015) has been run following the SPARC-CCMI specifications for the REF-C1 experiment (Eyring et al., 2013), i.e., sea surface temperatures and sea ice coverage from available observations and on-line explicitly calculated land temperatures. We have adjusted the manuscript to reflect a greater accuracy on this aspect. Please note the footnotes of Table 1, which clearly explain how surface temperatures are treated in Reference, Base, G4 and G4K experiments.

3) ULAQ-CCM performance wrt ice clouds: p10, l 2-4: "...we are considering thin ice clouds..." is this because ULAQ-CCM does not consider thick ice clouds or is this because the authors did a subsampling of the model output, i.e. selected only cases with thin cirrus? In the first case I have (again) severe doubts that ULAQ-CCM is the appropriate model for this study. In the second case the evaluation does not make sense, because the authors compare apples (thin cirrus) with oranges (all ice clouds) (apart from uncertainties in the MERRA and MODIS derived quantities).

The ULAQ-CCM does not consider thick ice clouds, as the Reviewer states, simply because the updraft velocities are calculated as a function of TKE, with typical values less than 30 cm/s, so that convective events of much stronger intensity, and leading to the formation of thicker ice clouds, are not present (see also Kärcher and Lohmann, 2002). We believe that the comparison of our results with MERRA2 and ERA5 data regarding ice water mixing ratio (see Fig. 1) is indeed meaningful, in the sense it may highlight both similarities and differences. Most of the latter may be attributed to this missing driver mechanism of thick cirrus formation (which is clearly discussed in the manuscript). Purpose of our study is to show that the largest fraction of UT ice originates from updraft in the range of approx. 10-30 cm/s and could be significantly affected by the atmospheric stabilization induced by SG.

4) The interpretation of the model results is very much focused on the "vertical temperature gradient – homogeneous ice formation" relationship. I am not sure whether this is a remnant of the first set of simulations in which heterogeneous ice formation had been erroneously switched off, but I miss an open discussion of other potential feedback effects. For example, changes in background cloudiness or large-scale circulation changes.

Other feedback effects have already been discussed in previous works (and referenced in the text), for example regarding circulation changes and stratospheric chemistry (Pitari et al., 2014; Visioni et al., 2017a). The background cloud cover is fixed in our model simulation. This is now clearly stated in the manuscript (P. 31, lines 4-7 in the revised manuscript). We also address the point that potential changes in background clouds may originate from the SG induced dynamical perturbation as noted by the Reviewer (thanks!), thus potentially affecting the all-sky TOA forcing of cirrus ice particles.

**Specific comments:**

**- p1, l3: Why only "homogeneous freezing"? How can you exclude SG effects on heterogeneous freezing? Or is this a remnant from the first draft, for which heterogeneous freezing was switched off in the simulations?**

Our model simulations (as well as others in the literature) show that the homogeneous freezing dominates over heterogeneous freezing (which results to produce less than 10% of the overall ice OD). Therefore, changes in heterogeneous freezing impact the overall changes in a limited way. To better pose the problem, however, we have removed "homogeneous": "The goal of the present study is to better understand the SG thermodynamical effects on the freezing mechanisms leading to ice particle formation."

**- p1, l6: How do you define "longwave" radiation? Aerosols also absorb incoming radiation in the near IR.**

The "longwave" term (LW as short name, throughout the manuscript) is used for the whole spectrum of the terrestrial blackbody radiation. The shortwave term (SW as short name, throughout the manuscript) is used for the whole spectrum of the incoming solar radiation. We are well aware that aerosols absorb both in the terrestrial spectrum and in the solar NIR. To avoid misunderstanding we now write "terrestrial and solar near infrared radiation".

**- p3, l24: Again - why only "homogeneous freezing"?**

Changed, see above.

**- p4, l6: ... help to explain...**

**Corrected.**

**- p4, l9-11: To design SG experiments which meet certain climate targets it is, at least in my view, crucial to consider all aspects and feedback processes in a self-consistent manner, which is not done in the present study. So this argument is counterproductive.**

Our goal was not to design a SG experiment such as that, and we agree with the comments of the Reviewer regarding the self-consistency. We believe however (and we make this clear in the conclusions of this study) that highly idealized experiments, such as ours, can still shed light on the physical processes that might be affected by SG, and this was our goal all along.

**- Sect. 2.1: Are the CCSM-CAM4 simulations ensemble runs or only one realization for each scenario? And what is the climate sensitivity of the model?**

The ensemble size for both CCSM-CAM4 scenarios is 2 (see Visioni et al., 2017b, Table 2). The equilibrium climate sensitivity of the model is 2.9 °C (Bitz et al., 2012), considering an idealized 2 x  $CO_2$  scenario. Further specifications are given in the above-mentioned study.

**- Table 1: equatorual -> equatorial; footnote 3 not used; do you use surface temperatures or SEA surface temperatures from CCSM-CAM4 in Base, G4 and G4K? That's an important difference and needs clarification!**

The Reviewer is correct. The third footnote is meant to go together with the second one in row 6, column 3. We have specified better in the manuscript that we always refer to surface temperatures and not just sea surface temperatures. The footnotes have now been changed to better explain our updated modeling strategy.

**- p7, l17: This represents...**

Corrected.

**- p8, I5: What is an ice mass fraction? I guess you mean ice mass mixing ratio?**

Yes, we have corrected accordingly.

**- p8, I23: What is Qext? Extinction efficiency coefficient?**

Yes, we now specify this in the manuscript.

**- p8, l24: How is upper troposphere defined? Which altitude range?**

The sum is over all vertical layers (but cirrus ice is only found in the upper troposphere, and this is why we specified it). We have therefore removed UT from the phrase.

**- p8, l25: remove ij after rij**

Done.

- p8, Fig.1: Do you use MERRA or MERRA-2? I assume MERRA-2. Please use a consistent nomenclature throughout the manuscript.

Yes, we use MERRA-2 and have corrected this everywhere.

**- p9, I5: Again - ice mass fraction?**

**Corrected.**

- Fig. 3: What's the purpose of showing this figure? I do not see the link to the present study, neither for model evaluation nor for the SG effects.

The previous review (referee #1) deemed important to know the fraction of heterogeneous freezing, so we added this figure to his suggestion.

**- Fig. 3: Add explanation of dashed and dash-dotted lines in panel b) to the caption.**

Done.

**- Fig. 4c): Again - what's the purpose of showing this figure?**

The probability of ice formation is crucial in the scheme we use to determine when and where freezing may occur for a given ice supersaturation threshold.

**- Fig. 3, 4 and 7: Which seasons are shown (tropopause)? My year has 4 seasons, but only there are only two lines.**

By seasonal variability, we mean 1 standard deviation from the average height. We have better clarified this in the new caption, the first time it appears (Fig. 3).

**- Table 2: caption: homogenous -> homogeneous, row 2: (ULAQ-CCM) missing bracket; row 5: (HET) missing**

**Corrected.**

**- Sect. 2.3: As mentioned above Sect. 2.3 needs clarification, especially with respect to the treatment of surface or sea surface temperatures, nudging or prescribed lower boundary condition.**

We have further clarified the aspects pointed out by the Reviewer (from P. 16, line 34 to P. 18 line 6 plus the revised Fig. 6, Table S1 and Fig. S2).

**- p14, l32: Does G4 assume 5 Tg(SO2)/yr or 8 Tg(SO2)/yr, as stated 2 lines above?**

G4 normally assumes 5 Tg-SO2/yr, at  $^{\circ}$ O. We have clarified this.

**- p15, l3: (Fig. 4-5) should read (Fig. 5-6)**

**Corrected.**

- Fig 5: Wouldn't it make more sense to show temperature anomalies from ULAQ-CCM instead from CCSM-CAM4 as this is the basis for the study? As far as I understand surface temperatures over land are calculated by the ULAQ-CCM?

We better clarify that we use for the reference case CCSM-CAM4 surface temperature, and not just SST temperatures. Considering the changes we have made to the simulations, we have however added to Fig. 6 the

changes in the surface temperatures calculated by ULAQ-CCM, due to online calculated indirect effects of SG aerosols.

**- Fig. 6 and following: Why do you show averages for 2030-2039 when the simulations cover 2020-2069 (at least according to Table 1)?**

We simply wanted to show changes in a specific selected decade, and because the injection starts at 2020, by 2030 the SG perturbation may have reached a quasi-steady state condition. Analyzing 10 years, allows us to remove time-dependent effects in the distribution of the stratospheric aerosols (such as the QBO), but results do not change significantly by averaging one or more decades. To follow the Reviewer suggestion, anyway, we now present all results as an average over 2030-69, i.e., simply skipping the first 10 years of SG when the stratospheric aerosols (and surface temperature anomalies G4-Base) may have not reached equilibrium.

**- Fig. 7a,b: A $\Delta T = 0$ contour line would be helpful to clearly identify regions with positive and negative temperature changes. Alternatively, a better color scale. And I would prefer to see the temperature changes starting at the surface.**

In the revised figure, we added a contour line for  $\Delta T = 0$ , as suggested by the Reviewer. We attach here the figure with altitude starting 0 km instead of 3 km, but in the manuscript we have decided to keep it at 3 km in order to focus on changes happening in the troposphere. Now, however, Fig. 6 focuses more on surface temperature changes, so that the reader can see both tropospheric and surface changes in the two figures.

**Fig. R3\_1.** Zonally and time-averaged changes of temperature (panels a,b) and vertical velocity (panels c,d) in experiments G4 (panels a,c) and G4K (panels b,d) with respect to the Base case (years 2030–69). The dashed lines show the mean tropopause height (with seasonal variability). The dash-dotted lines show the mean height (with seasonal variability) at which the temperature reaches 238 K, thus enabling homogeneous freezing. The dotted white line highlights where  $\Delta T=0$  K.

**- Discussion of Fig. 7a,b: The difference in surface temperatures between G4 and G4K has effect on the outgoing longwave energy and therefore the IR absorption by the SG aerosols. Furthermore I would expect general changes in cloudiness which also affect emission of terrestrial radiation. These aspects are not at all mentioned in the study.**

The Reviewer is right, in fact the results in terms of aerosol optical depth and RFs are not the same between simulations G4 and G4K (a brief discussion is now made in the manuscript). For what concerns the background cloudiness, this is kept fixed in the ULAQ model at climatological values and no effect of SG is present on clouds. This is beyond the purposes of the present study and could be considered in future studies. This is now specified in the manuscript (P. 31, lines 4-7 in the revised manuscript).

**- p18, l16ff: The authors state here that vertical motions caused by synoptic scale disturbances and gravity waves dominate the updraft velocities. Furthermore, they state that in G4 the vertical updrafts are reduced due to a reduced vertical temperature gradient, but how about the impact of changes in the meridional temperature gradient and subsequent changes in zonal winds and gravity waves?**

As clearly discussed in Kärcher and Lohmann (2002) the updraft velocities in the UT may span of 2-3 orders of magnitude, due to different dynamical drivers (synoptic scale motions, gravity waves, convection). Our calculation is based on the TKE approach and w values do not normally exceed ~30 cm/s. TKE is a function of the vertical temperature gradient and mean zonal wind shear. Dynamical changes produced by the SG aerosol radiative perturbation produce changes (including those cited by the Reviewer) that end up modifying the temperature gradient and zonal wind shear and then the TKE. This is the reason why we write: "The vertical velocity is reduced in G4 with respect to the Base case...due to the atmospheric stabilization caused by a reduction in the temperature vertical gradient". It is a short summary of what we explain above and which also appears in various parts in the manuscript discussion. For example, few lines below, we write (referring to Fig. 9): "They help explain how the SG sulfate changes act as drivers for dynamical changes in the UT, with significant effects on ice particle formation". To follow the Reviewer suggestion, we have modified this last sentence as follows: "They help explain how the SG sulfate changes act as drivers for dynamical changes in the UT (vertical and meridional temperature gradients, as well as vertical, horizontal winds and wave amplitude), with significant effects on ice particle formation" (P. 20, line 24-26 in the revised manuscript).

**- p18, l23: I assume the authors mean SO4 in the particulate phase, not in the gas phase.**

**Yes, we have better clarified this.**

- Discussion of Fig. 9: First of all, I do not understand why the LW heating rates in Fig. 9b) have been

calculated with temperatures fixed at Base values as written in the caption? Furthermore, the authors do not mention potential changes in adiabatic heating rates due to a change in the Brewer-Dobson-Circulation ("decreased wave activity and a consequent decrease in poleward mass fluxes") and their effects on lower stratospheric temperature changes. What is meant by "tropospheric convective cooling"? And how do the authors explain the neg. LW anomalies in G4 above ~26 km?

In order: (a) LW heating rates are shown as "instantaneous" heating rates (as it is normally done in the middle atmosphere) because the stratosphere is in quasi-radiative-equilibrium, so that the results of a non-adiabatic warming (or cooling) is a quick adjustment (positive or negative, respectively) of the temperature field. Except of course for the small departure from the radiative equilibrium, which produces changes in the residual vertical motion. In order to quantify the non-adiabatic radiative forcing of the SG aerosols is then necessary to show (in addition to the solar heating rates) the instantaneous LW heating rates (i.e., with fixed temperature in the calculation of the longwave radiative fluxes). As specified above, we are talking here of LW heating rates in the planetary blackbody spectrum, while SW heating rates result from both the solar NIR absorption by the aerosols and from UV absorption by ozone (including the effects of both SG-produced ozone changes and increased radiation scattering by the aerosols). (b): adiabatic heating rates are an indirect effect of the changing tropical upwelling. We are focusing on the direct radiative forcing by the aerosols, i.e. the diabatic heating rates. The latter will then perturb the atmospheric radiative budget and induce circulation changes and finally an adiabatic response of the atmosphere. Circulation changes (tropical upwelling, wave activity, poleward mass fluxes, strat-trop exchange have been fully covered and discussed in Pitari et al. (2014), Visioni et al. (2017), Visioni et al. (2018). (c) Tropospheric convective cooling refers to the decreased latent heat exchange due to less intense deep convection in a cooler SG atmosphere (the rather confusing statement in the original manuscript has been changed in a clearer way). (d) The discussion relative to the negative LW anomalies of heating rates above ~25 km was already given in the revised manuscript: "All features of the SW and LW heating rate anomalies in Fig. 9b can be fully explained taking into account the aerosol- $O_3$  coupled effects (Pitari et al. (2014)). The sign of tropical ozone changes under the SG conditions depends on altitude. The  $O_3$  decreases below ~25 km and increases above this height; this helps explain the positive/negative heating anomalies in SW and LW components above 25 km altitude". In other words: O3 increases above ~25 km due to increasing NOx and increasing tropical upwelling, so that the LW cooling rates in the 9.6  $\mu$ m O3 band increases as well.

**- Fig. 7/8: Are the displayed changes in vertical velocities statistically significant? The $\pm 1\sigma$ range (for which scenario? Base?) in Fig. 8 seems to be pretty wide compare to the differences between Base and G4.**

We report here part of our previous response to Reviewer #2, who expressed similar doubts. We note that the variability of w in Figure 8 is essentially due to seasonal changes and non-zonal asymmetries of the TKE. But if we isolate a given month in the time series, the vertical velocity change due to SG is more comparable to the w variability in the time series. We attach a figure below showing this quantity, to show what we mean.